# Context-Aware Testing: A New Paradigm for Model Testing with Large Language Models

**Paulius Rauba**\*
University of Cambridge
pr501@cam.ac.uk

**Nabeel Seedat**\*
University of Cambridge
ns741@cam.ac.uk

**Max Ruiz Luyten**
University of Cambridge
mr971@cam.ac.uk

**Mihaela van der Schaar**
University of Cambridge
mv472@cam.ac.uk

## Abstract

The predominant *de facto* paradigm of testing ML models relies on either using only held-out data to compute aggregate evaluation metrics or by assessing the performance on different subgroups. However, such *data-only testing* methods operate under the restrictive assumption that the available empirical data is the sole input for testing ML models, disregarding valuable contextual information that could guide model testing. In this paper, we challenge the go-to approach of *data-only testing* and introduce *context-aware testing* (CAT) which uses context as an inductive bias to guide the search for meaningful model failures. We instantiate the first CAT system, *SMART Testing*, which employs large language models to hypothesize relevant and likely failures, which are evaluated on data using a *self-falsification mechanism*. Through empirical evaluations in diverse settings, we show that SMART automatically identifies more relevant and impactful failures than alternatives, demonstrating the potential of CAT as a testing paradigm.

## 1 Introduction

The ability to rigorously test and validate machine learning (ML) models is crucial for their reliable deployment in real-world applications. Despite solid aggregate performance, ML models are unreliable in a variety of real-world scenarios, such as on different subgroups [1–6], or encountering data deviating from its training distribution [7–10], often resulting in significant financial or societal consequences. These failures point to deficiencies in the way we test such ML models. To understand this in greater detail, let's address two questions: how are we currently testing ML models and can we do better?

**How are we testing?** The predominant *de facto* paradigm of testing ML models relies on using only held-out data to evaluate the model either on average or by assessing performance on different subgroups within the dataset. Such *data-only methods* optimize a given objective function to find subgroups (or slices of data) within the dataset where a trained model underperforms relative to aggregate performance. We refer to this relative underperformance as a *model failure*. However, data-only methods operate under the restrictive assumption that the available empirical data is the sole input for testing ML model performance. In practice, this is almost always violated. It is common to have *a priori* knowledge of where models are likely to fail, given the problem context.

The restrictive assumption of data-only methods comes at a significant cost. Specifically, data-only methods, which have been the go-to approach for ML testing, are required to iterate and test the

---
\*Equal Contribution

38th Conference on Neural Information Processing Systems (NeurIPS 2024).

performance across a large number of possible subgroups. Each subgroup test is equivalent to evaluating a separate hypothesis about the model's performance on that specific slice of data. This raises a subtle but important problem—from the perspective of hypothesis testing, such methods *implicitly test multiple hypotheses* (Sec. 3.1). The more subgroups we test, the more hypotheses we implicitly evaluate. Consequently, this results in critical problems associated with multiple testing: (i) *high false positive* and (ii) *high false negative rates* (Sec. 3.2). A third complementary challenge is that each data slice tested is (iii) *not necessarily practically meaningful*. The practical implications of these drawbacks are quite concerning—they hinder our ability to accurately identify where the model performs poorly, thereby undermining the reliability of our model evaluations.

**Can we do better?** To address these three challenges, we propose to *loosen the restrictive assumption of reliance only on data* and propose a new paradigm of testing called Context-Aware Testing (CAT) to offer an alternative to the dominant data-only view (Sec. 3.3). CAT provides a principled approach to incorporate external knowledge—or context— to the ML testing process. With the view that each evaluated data slice corresponds to an implicit hypothesis test, we propose that ML evaluation on observational datasets can be achieved via a *context-guided sampling mechanism* (Definition 1). This mechanism is a sampling procedure that uses context as an inductive bias to prioritize specific data slices to test for which have a higher chance of surfacing meaningful model failures. Therefore, CAT fundamentally helps to answer the question of "what should we test for?".

Let's consider an example of building an ML model to predict prostate cancer [11]. Data-only methods employ a search procedure over the dataset to find divergence across a large number of possible feature combinations which may lead to (i) *high false positive rates* by identifying spurious underperforming subgroups (e.g. based on eye color or patient ID); (ii) *high false negative rates* by failing to identify true underperforming subgroups due to the large number of combinations tested and applied testing corrections; and (iii) *testing subgroups which are not practically meaningful* (e.g. interaction between eye color and height). In contrast, a CAT-based approach would define and target task-relevant subgroups, limiting the number of tests conducted with better false positive control and greater statistical power. As we empirically show in Sec. 5, obtaining many false positives and false negatives is overwhelmingly common in current testing practices.

In bringing CAT to reality, we develop the framework called **SMART** [2] **Testing**, which performs automated ML model evaluation by actively identifying potential failure cases (Sec. 4). SMART uses large language models (LLMs) to generate contextually relevant failure hypotheses to test. We further introduce a *self-falsification mechanism*, to automatically validate the generated failure hypotheses using the available data, allowing efficient pruning of spurious hypotheses. Finally, SMART generates comprehensive model reports that provide insights into the identified failure modes, their impact, and potential root causes, enabling stakeholders to make informed decisions.

**Contributions.** ① We identify critical gaps in predominant data-only ML testing, illustrating they miss important dimensions (Sec. 3). ② We formalize the *Context-Aware Testing* paradigm, providing a principled framework to incorporate context in addition to data into the testing process, which is then used to guide the generation of targeted tests (Sec. 3). ③ We build the first context-aware testing system, *SMART* Testing, which employs LLMs to hypothesize likely and relevant model failures and empirically refutes them with data using a novel *self-falsification mechanism* (Sec. 4). ④ We demonstrate the value of context for effective testing, challenging the de facto data-only paradigm by showing how SMART identifies impactful model failures while avoiding false positives across diverse settings, when compared to data-only testing. Additionally, SMART identifies failures on important societal groups and generates comprehensive model reports (Sec. 5).

## 2 Related work

To highlight the need for SMART Testing, we contrast it with other ML testing paradigms — specifically Data-only testing methods which address the same testing problem as SMART. We provide an overview in Table 6 and an extended discussion in Appendix A.

*Data-only testing methods* [12–15]: Address the question: "what should we test?". Data-only methods search the data to find "slices" where the model's predictions underperform compared to average performance, deeming those slices as model failures. Although automated, data-only

---

[2]systematic, modular, automated, requirements-responsive, transferable

approaches operate *only* on raw data without accounting for the problem context. Consequently, they must search across a large space of potential failures, usually covering all subsets of features and their distinct values. While an exhaustive search may seem beneficial, as we show in Sec. 3, the reality is that performing many tests on a finite dataset risks discovering slices where model failure is irrelevant or due to random variability, i.e. the multiple testing problem. SMART Testing addresses this challenge by prioritizing relevant and likely model failures through contextual awareness.

***Orthogonal testing dimensions***: While SMART addresses what to test, several other dimensions of model testing exist that are orthogonal to our approach (detailed in Appendix A). (i) Behavioral Testing [16, 17, 5] evaluates model behaviors (i.e. responses to data) by operationalizing tests along pre-defined dimensions (often defined by humans) — rather than discovering the test dimensions. SMART fundamentally differs by addressing the core issue of "what to test". (ii) Software functional testing [18, 19], aims to primarily test functional correctness (e.g. input-output functionality such as monotonicity), rather than testing for failures. In addition, test cases are either pre-specified or specified with an approximation of the underlying model to probe for functional correctness.

# 3 A context-aware testing framework for ML

The prevailing paradigm for testing ML models relies on *data-only* methods which exclusively use data to surface model failures. In this section, we explore the limitations of data-only methods by viewing ML testing as a multiple hypothesis testing problem. We explore why data-only methods are uniquely prone to finding *false positive* and *false negative* model failures. To address this, we introduce a new paradigm of testing called **context-aware testing** which relies on external knowledge, or *context*, as an inductive bias to better identify where models fail.

## 3.1 A multiple hypothesis testing view of ML evaluation

**Preliminaries.** Denote the feature space by $\mathcal{X}$ and the label space by $\mathcal{Y}$, and $\mathcal{P}$ the joint probability distribution over $\mathcal{X} \times \mathcal{Y}$. We wish to test a fixed, trained black-box model $f : \mathcal{X} \to \mathcal{Y}$, using a finite dataset $\mathcal{D} \subset \mathcal{X} \times \mathcal{Y}$ usually split into $\mathcal{D}_{train} = \{(x_i, y_i)\}_{i=1}^{N_{train}}$ and $\mathcal{D}_{test} = \{(x_i, y_i)\}_{i=1}^{N_{test}}$. We assume the existence of a loss function $\ell : \mathcal{Y} \times \mathcal{Y} \to \mathbb{R}$ which measures the discrepancy between the model's prediction and the true labels point-wise.

The primary goal of ML testing is to identify *meaningful failure modes*—subgroups (data slices) of the data distribution where the model's performance is significantly worse than its average behavior. Formally, let $\mathcal{S} \subseteq \mathcal{X} \times \mathcal{Y}$ denote a *data slice* and let $\mathcal{P}_{\mathcal{S}} = \mathcal{P}(\cdot | (x, y) \in \mathcal{S})$ be the conditional distribution induced by $\mathcal{S}$. We aim to identify slices where the slice-specific expected loss $\mu_{\mathcal{S}} = \mathbb{E}_{(x,y) \sim \mathcal{P}_{\mathcal{S}}}[\ell(f(x), y)]$ significantly exceeds the population-level expected loss $\mu_{\mathcal{P}} = \mathbb{E}_{(x,y) \sim \mathcal{P}}[\ell(f(x), y)]$[3].

**Testing ML models is a multiple hypothesis testing problem**. We are interested in identifying failure modes that generalize beyond the training dataset. We can interpret the empirical dataset $\mathcal{D}$ as a sample from a broader distribution $\mathcal{P}$, and our goal is to make an *inferential* claim on the performance on the data slices with respect to $\mathcal{P}$. Suppose we have a candidate data slice $\hat{\mathcal{S}} \subseteq \mathcal{D}$. We can evaluate the empirical slice-specific loss as $\hat{\mu}_{\mathcal{S}} = |\hat{\mathcal{S}}|^{-1} \sum_{(x,y) \in \hat{\mathcal{S}}} \ell(f(x), y)$ and compare it to the empirical loss over the entire dataset $\hat{\mu}_{\mathcal{D}} = |\mathcal{D}|^{-1} \sum_{(x,y) \in \mathcal{D}} \ell(f(x), y)$. To make an inferential claim about the model's performance on the data slice $\mathcal{S}$ w.r.t. $\mathcal{P}$, we can follow the frequentist testing paradigm and formulate a hypothesis test where we evaluate whether the performance is significantly different. Therefore, $H_0 : \mu_{\mathcal{S}} = \mu_{\mathcal{D}}$ and alternative hypothesis $H_1 : \mu_{\mathcal{S}} \neq \mu_{\mathcal{D}}$, where $\mu_{\mathcal{S}} = \mathbb{E}_{(x,y) \sim \mathcal{P}_{\mathcal{S}}}[\ell(f(x), y)]$ and $\mu_{\mathcal{D}} = \mathbb{E}_{(x,y) \sim \mathcal{P}}[\ell(f(x), y)]$ denote the *true* slice-specific and population-level losses. In practice, this evaluation could be done by running an appropriate frequentist statistical test and evaluating whether $p < \alpha$ for each slice, given some pre-defined $\alpha$.

However, in realistic testing scenarios, we evaluate the model's performance not just on a single slice but on a large collection of candidates $\{\mathcal{S}_j\}_{j=1}^{m}$. This amounts to conducting many simultaneous hypothesis tests. Accounting for multiple testing is important. A naive testing procedure that does not adjust for multiplicity could surface a large number of spurious failure modes simply by chance

---

[3]while technically any subset of the dataset can be considered a data slice, we are practically interested in *meaningful* failures, such as model failures on vulnerable socio-economic groups.

(Type I error). Conversely, controlling the false discovery rate [20] may involve adjusting the per-test significance threshold to $\alpha' \ll \alpha$, potentially sacrificing power to detect true failures (Type II error).

> ♀ The multiple hypothesis testing viewpoint reveals a key challenge in ML model evaluation: To reliably surface meaningful failures, we require a principled procedure for generating a relatively small number of promising hypotheses (candidate slices) to test.

### 3.2 The failures of data-only testing

Existing ML testing methodologies are *data-only* in that they only use the available empirical data to test for model failures and vary in their optimization objective [14, 12, 13, 21, 15]. However, data-only methods operate under the restrictive assumption that the available empirical data is the sole input for testing ML model performance. In practice, this is almost never the case. It is common to have an *a priori* understanding of where models are likely to fail given the data distribution, model class, training algorithm, and deployment context. This restrictive assumption results in three challenges: ▶ **(i) High false positive rate**: data-only methods search over a large space of data slices and each evaluation amounts to an implicit hypothesis test (Sec. 3.1). Therefore, the probability of observing a false failure increases with every test performed. ▶ **(ii) High false negative rate**: The naive testing procedure can be made robust by correcting for the number of tests performed which reduces the statistical power to detect true failures. ▶ **(iii) Lack of meaningful failures**: Data-only methods are fundamentally limited by the fact that not all statistically significant slices are practically meaningful. We empirically validate these claims in Sec. 5.

> ♀ The core limitation of data-only testing is the lack of a principled failure mode *discovery* procedure that can incorporate prior knowledge to guide the search for meaningful errors.

### 3.3 Formulating context-aware testing

To address the limitations of data-only testing, we introduce **context-aware testing** (CAT), a principled framework for identifying meaningful model failures using context. This could be the context implicitly encoded in the dataset (i.e. via meaningful feature names) or available external input (i.e. external contextual knowledge, such as a string of input information from a human). Let $\mathcal{C}$ denote the space of all possible contextual information and $c \in \mathcal{C}$ be specific external input. Our core insight is that we can use $\mathcal{C}$ as an inductive bias to select which slices to test for.

**Definition 1** (Context-Aware Testing). *Let $\mathcal{X}$, $\mathcal{Y}$, $\mathcal{P}$, $f$, $\mathcal{D} = \{(x_i, y_i)\}_{i=1}^{N}$, and $\ell$ be defined as in the standard supervised learning setup. Let $\mathcal{C}$ be a space of contexts.*

*Context-aware testing is defined by two procedures:*

1. *A context-guided slice sampling mechanism $\pi : \mathcal{C} \times (\mathcal{X} \times \mathcal{Y}) \times \mathbb{N} \to 2^{\mathcal{X} \times \mathcal{Y}}$ such that $\pi(c, \mathcal{D}, m) = \{\mathcal{S}_1, \ldots, \mathcal{S}_m\}$, where $c$ is used as an inductive bias for function $\pi$ to prioritize slices likely to contain meaningful failures, and $m \in \mathbb{N}$ are the number of slices to evaluate.*

2. *A multiple hypothesis testing procedure: $\forall \mathcal{S}_i \in \pi(c, \mathcal{D}, m)$, test $H_0 : \mu_{\mathcal{S}_i} = \mu_{\mathcal{D}}$ vs. $H_1 : \mu_{\mathcal{S}_i} \neq \mu_{\mathcal{D}}$, where $\mu_{\mathcal{S}_i} = \mathbb{E}_{(x,y) \sim \mathcal{P}_{\mathcal{S}_i}}[\ell(f(x), y)]$, $\mu_{\mathcal{D}} = \mathbb{E}_{(x,y) \sim \mathcal{P}}[\ell(f(x), y)]$*

*A meaningful failure is characterized by statistical significance and practical relevance.*

The targeted sampling mechanism $\pi$ uses the context $\mathcal{C}$ as an inductive bias to prioritize testing of slices that are (i) *more relevant* to the deployment context, and (ii) *more likely* to exhibit significant performance gaps.

This principled slice selection offers several key advantages over data-only methods: ▶ *(i) Improved false positive control*: by limiting the number of tests conducted to $m$, CAT controls the risk of spurious discoveries that arise when naively testing all possible slices. ▶ *(ii) Improved true positive rate*: The targeted selection of slices likely to contain failures maintains test power by avoiding the need to aggressively correct for multiple testing. ▶ *(iii) Meaningful failures*: context-guided sampling identifies failures that are both statistically significant and practically relevant.

💡 Context-aware testing overcomes the limitations of data-only methods by employing a principled, context-guided slice sampling mechanism $\pi$ to prioritize the discovery of meaningful model failures.

The core technical challenge in realizing CAT is the development of an effective context-conditional sampling algorithm $\pi$. In the following section, we propose a concrete instantiation of $\pi$ using a large language model to generate plausible failure modes and guide testing of ML models.

## 4  SMART Testing

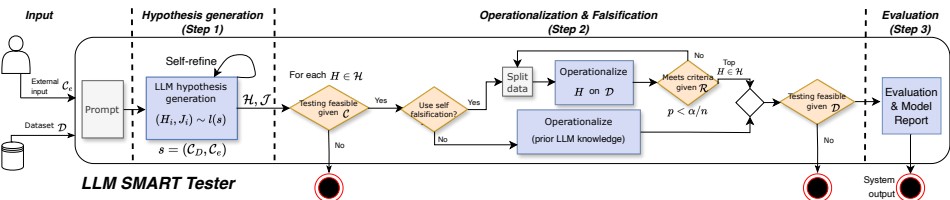

Figure 1: Overview of the SMART Testing Framework showing the four steps. All steps are automatically executed by an LLM.

We now instantiate the CAT framework outlined in Sec. 3.3 with a method called **SMART** testing (**s**ystematic, **m**odular, **a**utomated, **r**equirements-responsive, and **t**ransferable). SMART generates relevant and likely hypotheses about potential model failures and empirically evaluates these hypotheses on available data. SMART follows a four-step procedure: (1) Hypothesis generation; (2) Operationalization; (3) Self-falsification; (4) Reporting. Table 1 provides an early illustration of what it practically means to generate hypotheses and test them with SMART. The procedure below details how the four steps are implemented.

Table 1: Example hypotheses on model failure, justifications, and operationalizations generated by the SMART framework on a healthcare dataset. The p-values show whether the model's performance significantly differs from average performance with $|\Delta Acc|$ measuring the effect size.

| Hypothesis | Justification | Operationalization | p-value | $|\Delta Acc|$ | Evidence |
|---|---|---|---|---|---|
| $f$ underperforms on elderly patients. | Elderly patients may have more complex health situations due to age-related comorbidities, which could make predictions less accurate. | age $>= 72$ | 0.000 | 0.194 | Supported |
| $f$ underperforms on patients with multiple comorbidities. | The presence of multiple comorbidities could complicate the prediction model due to interactions between different health conditions. | comorbidities $>= 2$ | 0.900 | 0.0270 | Not supported |
| $f$ underperforms on patients undergoing conservative management. | Conservative management might be chosen for patients with more complex or less predictable cases, which could lead to worse predictive performance. | treatment_conservative _management $== 1$ | 0.000 | 0.200 | Supported |

**Step 1: Hypothesis Generation.** Recall from Sec. 3.3, we wish to define a sampling mechanism $\pi$ to sample slices $\mathcal{S}$ which are both relevant and have a high relative likelihood of failure — where $\pi$ should be both contextually-aware and able to integrate requirements to guide sampling.

We posit that LLMs have the potential to satisfy these properties due to the following capabilities: ▶ *Contextual understanding*: LLMs have been pretrained with a vast corpus of information and hence have extensive prior knowledge around different contexts and settings [22–25]. ▶ *Integrate requirements*: LLMs are adept at integrating requirements or additional information about the problem via natural language [26, 24]. ▶ *Hypothesis proposers*: In proposing likely failure modes, LLMs have also been shown to be "phenomenal hypothesis proposers" [27].

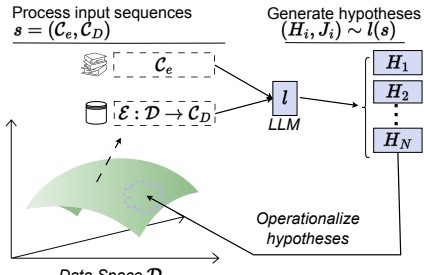

Figure 2: SMART uses an LLM to integrate context $\mathcal{C}$, and data context $\mathcal{D}_c$. Relevant and likely failure hypotheses are then generated by LLM (i.e. sampling mechanism). The hypotheses are then operationalized in $\mathcal{D}$ and evaluated. In contrast, data-only methods are not guided by context and requirements, searching more exhaustively in $\mathcal{D}$ for divergent slices.

An LLM is defined as a probabilistic mapping $l : \Sigma^* \to P(\Sigma)$, where $\Sigma$ denotes the vocabulary space, and $P(\Sigma)$ represents the probability distributions over $\Sigma$. The model processes input sequences $s \in \Sigma^*$, each a concatenation of tokens representing external (contextual) input $\mathcal{C}_e$ (which can be null) and dataset contextualization $\mathcal{C}_D$ is formalized as $s = (\mathcal{C}_e, \mathcal{C}_D)$. We extract the contextualized description $\mathcal{C}_D$ from the dataset $\mathcal{D}$ using

an extractor function $\mathcal{E} : \mathcal{D} \to \mathcal{C}_D$, which captures essential dataset characteristics (e.g. feature relationships and high-level dataset information). Additionally, we highlight the the LLM will implicitly extract context based on the context encoded in the dataset (e.g. via meaningful feature names). Based on input $s$, the LLM predicts a distribution over $\Sigma$ from which hypotheses of model failure and corresponding justifications are sampled.

As depicted in Fig. 2, given $\mathcal{C}_e$ and $\mathcal{C}_D$, we sample the $N$ most likely hypotheses of failures, $\mathcal{H} = \{H_1, H_2, \ldots, H_N\}$, and corresponding justifications $\mathcal{J} = \{J_1, J_2, \ldots, J_N\}$, to provide explainability. This process is formalized using the LLM's mapping $l$ as follows:

$$(H_i, J_i) \sim l(s), \text{where } s = (\mathcal{C}_e, \mathcal{C}_D), \ \forall i \in \{1, 2, \ldots, N\}.$$

**Step 2: Operationalization.** The process of operationalizing each hypothesis $H_i \in \mathcal{H}$ involves translating its natural language expression into a form that can directly operate on the training dataset $\mathcal{D}_{train}$ (an example is provided in Table 1). To achieve this, we define an interpreter function $I : \mathcal{H} \to \{0, 1\}^{\mathcal{X}}$ that maps each natural language hypothesis $H_i$ to a corresponding binary-valued function $g_i : \mathcal{X} \to \{0, 1\}$ on the feature space $\mathcal{X}$, where $g_i(x) = 1$ if $x$ satisfies the criteria and $g_i(x) = 0$ otherwise. Each function $g_i$ induces a data slice $\mathcal{S}_i \subseteq \mathcal{D}_{\text{train}}$ consisting of data points that satisfy the criteria of hypothesis $H_i$, such that $\mathcal{S}_i = \{(x, y) \in \mathcal{D}_{\text{train}} : g_i(x) = 1\}$. Therefore, each hypothesis, after operationalization, corresponds to a specific slice that is being tested on. Steps 1 and 2 serve to practically instantiate $\pi$ from Sec. 3.3.

**Step 3: Self-falsification** We introduce a novel self-falsification mechanism to empirically evaluate (or *refute*) the generated hypotheses[4]. Specifically, for each feasible hypothesis $H_i \in \mathcal{H}$, we attempt to falsify the hypothesis with observed empirical data [5]. This involves evaluating the model $f$ over the data slice $\mathcal{S}_i$ operationalized from $H_i$. We then assess whether the slice performance on $f$ has a significant deviation from the model's overall performance. For instance, in Table 1, this is done by computing $|\Delta \text{Acc}|$ and the p-value. The significance of this deviation is determined through frequentist statistical testing, i.e. when $p < \alpha$ for any $\alpha$ which might also be adjusted for multiple hypothesis testing. This step effectively "reshuffles/reranks" the hypotheses based on their likelihood on the observed data. For

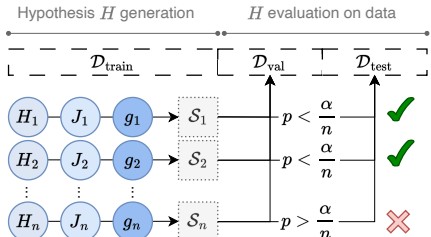

Figure 3: A self-falsification module within the SMART framework. A hypothesis generator $l$ generates plausible hypotheses and justifications for when the model might fail. This is operationalized with $\phi_i$ and tested against the empirical data.

example, when benchmarking we select the top $n$ hypotheses based on statistical significance: $\mathcal{H}^* = \arg\min_n_{h_i \in \mathcal{H}_{\mathcal{T}}} \{p_i < \alpha\}$.

***Remark***: As shown in Fig. 1, we can exclude the self-falsification from SMART in cases of small-sample sizes. We denote this ablation of SMART as $SMART_{NSF}$.

**Step 4. Model Performance Evaluation and Reporting.** Finally, SMART automatically generates a report of the overall performance of the model under varying conditions generated by the LLM, including a summary report, a complete summary of the tests carried out, intermediate and final results, and potential failure modes of the ML model.

> 💡 SMART is a tabular CAT method which (i) directly *searches for model failures*, sampling targeted tests (Sec. 3.3) and (ii) *incorporates data, and context* into ML testing.

**Practical use of SMART.**

We highlight the practical use of SMART testing and emphasize SMART's ease of use and minimal input requirements needed. In particular, as shown in the example below, users do not need any prior knowledge to use SMART. Rather, we make use of the context inherently encoded in the dataset, feature names and task.

---

[4]self-falsification is rooted in scientific philosophy and the seminal ideas of Popper [28] who proposed the principle of falsification—the ability to refute hypotheses with empirical observations—as driving science.

[5]We assume data for self-falsification is available. If not, discovered slices could guide new data collection.

```
1  import SMART
2  # Instantiate SMART
3  model_tester = SMART('gpt-4')
4  # Give desired context (this can be left as an empty string)
5  context = "Find where a model fails for the cancer prediction task."
6  # Load ML model
7  model = XGBoost()
8  dataset_description = X.describe()
9  # Test the model
10 model_tester.fit(X, context, dataset_description, model)
```

Code Listing 1: Use of SMART Testing

## 5 Illustrating SMART use cases

We now quantitatively evaluate SMART Testing [6] and demonstrate the viability of this new ML testing paradigm (i.e. CAT), in contrast to data-only testing. Table 2 summarizes our experiments and takeaways.

**Baselines.** We compare SMART with a variety of data-only testing baselines namely: Autostrat [14], Slicefinder [12], Sliceline [13], Pysubgroup [21] and Divexplorer [15]. We also evaluate $SMART_{NSF}$ (i.e. no self-falsification) as an ablation. Given space limits, we exemplify the LLM in SMART by GPT-4 [29] and our ML model to audit is logistic regression. We investigate other LLMs in the Appendix D.3 and other tabular models in Appendix D.4, observing similar results. We provide additional experimental details in Appendix C.

Table 2: Summary of experiments and takeaways.

| Sec. | Experiment | Takeaway |
|---|---|---|
| Figure 4 | Assess robustness to irrelevant features | SMART consistently avoids irrelevant features, outperforming data-only methods. |
| Table 3 | Assess ability to identify significant model failures | SMART discovers slices with larger performance discrepancies across models. |
| Table 4 | Measure FNR in identifying underperforming subgroups | SMART achieves the lowest false negative rates in all settings. |
| Table 5 | Assess robustness to potential LLM biases | SMART effectively mitigates biases in identifying underperforming subgroups. |
| Table 10 | Evaluate ability to satisfy testing requirements | SMART satisfies most requirements while maintaining statistical significance. |
| Figure 8 | Assess how sample size affects irrelevant feature detection | SMART consistently avoids irrelevant features regardless of sample size. |
| Table 11 | Evaluate performance in different deployment environments | SMART identifies more significant failure slices in new environments. |
| Figure 9 | Assess impact of sample size on performance | SMART consistently outperforms data-only methods across all sample sizes. |
| Table 12 | Evaluate tendency to flag non-existent failures | SMART avoids spurious failures, unlike data-only methods. |
| App. D.3 | Compare performance of GPT-3.5 and GPT-4 in SMART | SMART using both GPT versions outperform benchmark methods. |
| Table 13, 14 | Evaluate SMART across different tabular ML models | SMART identifies larger performance discrepancies across various models. |
| Table 20 | Evaluate SMART the cost of using SMART | SMART is cost efficient to generate hypothesis less than 0.5 USD. |
| Table 21 | Compare the overlap of hypotheses w/ open-weight models | SMART using open-weight models has a significant overlap of hypotheses. |
| Table 22 | Evaluate SMART' | SMART identifies larger performance discrepancies across various models. |
| App. D.10 | Showcase SMART's practical output | SMART generates comprehensive model reports, providing clear justifications for each hypothesis. |

### 5.1 Robustness to False Positives

**Goal.** We aim to underscore the role of contextual awareness in preventing false positives in model testing. In particular, we consider the scenario when dealing with tabular data that may contain many irrelevant or uninformative features [30], persisting even post-feature selection [31, 32]. We contrast SMART which explicitly accounts for context, to data-only approaches which are context-unaware and can only operate on the numerical data.

**Setup.** We fit a predictive model to the training dataset, varying the number of irrelevant, synthetically generated features contained in the dataset. The irrelevant features are drawn from different distributions. We then quantify the proportion of conditions in the identified slices that *falsely* include the irrelevant synthetic features. We evaluate using five real-world tabular datasets spanning diverse domains, namely finance, healthcare, criminal justice and education: loan, breast cancer, diabetes, COMPAS recidivism [33] and OULAD eduction [34]. These datasets have varying characteristics, from sample size to number of features and are representative of different contexts pertinent to tabular ML, to demonstrate the effectiveness of SMART across various real-world contexts

**Analysis.** Fig. 4 shows the proportions of irrelevant features included in slices for increasing numbers of irrelevant features. Data-only methods are unaware of context and are shown to spuriously include high proportions of irrelevant features in their slices; i.e. *false positives (FPs)*. Additionally, these false discoveries increase for data-only methods as the number of irrelevant features increases.

---

[6]Code can be found at: https://github.com/pauliusrauba/SMART_Testing or https://github.com/vanderschaarlab/SMART_Testing

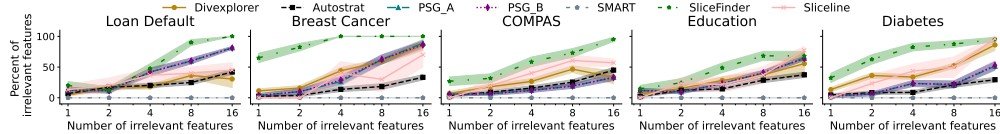

Figure 4: Contextual-awareness in SMART reduces FPs, i.e. reducing the proportion of irrelevant features in slices. SMART is not sensitive to the number of irrelevant features, unlike data-only methods. ↓ is better

Consequently, the sensitivity to FPs of data-only methods risks that slices identified are neither empirically relevant nor meaningful. In contrast, SMART, by virtue of contexual-awareness when generating hypotheses is *not* sensitive to extraneous and contextually irrelevant features. Importantly, SMART also maintains its robustness to FPs even with an increasing number of irrelevant features.

***Remark.*** We also assess sensitivity to the number of data samples. Fig. 9, Appendix D.6 shows that SMART remains robust to variations irrespective of sample size. In contrast, data-only methods have variable performance with false discoveries sensitive to the sample size.

**Avoiding non-existent failure slices.** It is important that model testing does not flag failures when there are none. In Appendix D.5, we demonstrate that SMART's contextual awareness means that it is resistant to spurious failure slices that have no underlying relationships. In contrast, data-only approaches implicitly assume the existence of problematic slices, which we empirically show makes them prone to spuriously flagging non-existent failure slices.

**Takeaway 1.** SMART's contextual awareness ensures testing relevance, thereby reducing false positives across different scenarios, in contrast to data-only methods.

## 5.2 Targeting model failures

**Goal.** We assess whether the identified failure slices persist when evaluated on new, unseen data across different tabular models. This evaluates the generalization of the identified model failures across multiple different ML models.

**Setup.** We use the prostate cancer dataset [35] and aim to discover slices indicative of underperformance. Thereafter, we assess generalizability of the identified failures across four different ML models (logistic regression, SVM, XGBoost and MLP). Specifically, we compare the top identified slice from each method and compute the absolute difference between the accuracies of that slice and the remainder of the dataset.

Table 3: Identifying slices with the highest performance discrepancies. We show differences in accuracies ($|\Delta Acc|$) between the top identified divergent slice and average performance across four classifiers (over 5 runs). ↑ is better.

| | LogisticRegression | SVM | XGBClassifier | MLPClassifier |
|---|---|---|---|---|
| Autostrat | $0.24 \pm 0.02$ | $0.24 \pm 0.02$ | $0.09 \pm 0.09$ | $0.24 \pm 0.02$ |
| $PSG\_B$ | $0.23 \pm 0.01$ | $0.23 \pm 0.01$ | $0.11 \pm 0.07$ | $0.23 \pm 0.01$ |
| $PSG\_A$ | $0.23 \pm 0.01$ | $0.23 \pm 0.01$ | $0.11 \pm 0.07$ | $0.23 \pm 0.01$ |
| Divexplorer | $0.05 \pm 0.11$ | $0.09 \pm 0.13$ | $0.14 \pm 0.15$ | $0.02 \pm 0.05$ |
| Slicefinder | $0.01 \pm 0.00$ | $0.01 \pm 0.00$ | $0.00 \pm 0.00$ | $0.01 \pm 0.01$ |
| Sliceline | $0.26 \pm 0.06$ | $0.26 \pm 0.06$ | $0.18 \pm 0.09$ | $0.26 \pm 0.06$ |
| SMART$_{NSF}$ | $0.17 \pm 0.01$ | $0.17 \pm 0.01$ | $0.09 \pm 0.05$ | $0.17 \pm 0.01$ |
| **SMART** | $\mathbf{0.37 \pm 0.03}$ | $\mathbf{0.37 \pm 0.03}$ | $\mathbf{0.26 \pm 0.06}$ | $\mathbf{0.37 \pm 0.03}$ |

We posit that a testing framework should be able to identify slices with high-performance discrepancies on unseen data across multiple ML models.

**Analysis.** Table 3 shows that SMART slices exhibit the greatest discrepancies in model accuracies on unseen test data across different ML models — indicating the discovered failure modes are generalizable. We find that SMART surfaces slices with greater performance differences compared to $SMART_{NSF}$ (ablation without self-falsification), highlighting the importance of the introduced self-falsification mechanism. In contrast, data-only methods fail to identify slices where the performance discrepancy is as large as SMART. This limitation can be attributed to the tendency of data-only methods to overfit the training data, leading to high false discovery rates.

**Takeaway 2.** SMART discovers failure slices where the model substantially underperforms, generalizing to unseen test data across different ML models. In contrast, data-only approaches fail to find slices where the difference in accuracies is as large, highlighting the lack of generalizability and reliability of their findings.

## 5.3 Robustness to False Negatives

In our setup, false negatives (FNs) are directly tied to true positives (TPs). i.e. the more true positives we find, the fewer FNs we miss. Across multiple experiments (Fig. 4, Table 10, Table 3, Table 16), we consistently show that SMART identifies TPs at substantially higher rates than data-only. For example, in Table 16, SMART identifies an average of 9.6 out of 10 subgroups where the ML

model significantly underperforms. The fact that data-driven methods discover fewer such subgroups implies that they are missing the ones SMART uncovers.

**Goal.** That said, we conduct an additional experiment to directly assess the false negative rate, wherein we can control issues (as FNs are naturally unspecified in real data).

**Setup.** We simulate a dataset to predict recidivism ($Y$) based on five covariates: gender, race, age, income, and education. $\log(\frac{P(Y_i|X_i=j)}{P(Y_i\neq j)}) = \alpha_j - (\delta_1 X_{\text{age}} + \delta_2 C_{\text{income}} + \delta_3 C_{\text{education}}) + \epsilon$, where $\epsilon \sim \mathcal{N}(\mu_0, \sigma_0)$. We then train a predictor function $\hat{f}$ on the data and synthetically introduce underperformance on certain corrupted subgroups. For an individual $i$, if they belong to corrupted subgroup $j$, the prediction $\hat{Y}_i$ is equal to $\hat{f}(X_i)$ with probability $1 - p$, and a random prediction sampled from a Bernoulli distribution with probability $p$. If the individual does not belong to subgroup $j$, the prediction is simply $\hat{f}(X_i)$. Finally, we measure how often each testing method identifies that the model $\hat{f}$ underperforms on a subgroup.

**Analysis.** Table 4 demonstrates results where we synthetically manipulate/corrupt the performance of an ML model on a single subgroup (n=1), two subgroups (n=2), and three subgroups (n=3) out of a total of five. The results show the average of 20 runs, where the corrupted groups are randomly selected within each run. We find

Table 4: False Negative Rate (FNR) for different methods at various settings. ↓ is better.

|  | FNR (n=1) | FNR (n=2) | FNR (n=3) |
|---|---|---|---|
| Autostrat | $0.75 \pm 0.44$ | $0.88 \pm 0.22$ | $0.92 \pm 0.15$ |
| pysubgroup_beam | $0.65 \pm 0.49$ | $0.75 \pm 0.26$ | $0.68 \pm 0.17$ |
| pysubgroup_apriori | $0.65 \pm 0.49$ | $0.75 \pm 0.26$ | $0.68 \pm 0.17$ |
| Divexplorer | $0.05 \pm 0.22$ | $0.40 \pm 0.35$ | $0.72 \pm 0.31$ |
| Sliceline | $0.25 \pm 0.44$ | $0.50 \pm 0.36$ | $0.70 \pm 0.28$ |
| **SMART** | $\mathbf{0.00 \pm 0.00}$ | $\mathbf{0.05 \pm 0.15}$ | $\mathbf{0.38 \pm 0.27}$ |

that SMART consistently is least susceptible to false negatives across all corrupted variables, when compared to data-only methods which struggle especially once more than one variable is corrupted. This serves to corroborate our earlier results.

**Takeaway 3.** SMART is less prone to FNs, compared to data-only methods across all settings.

### 5.4 Assessing and mitigating potential LLM challenges and biases.

**Background.** In many settings, we want SMART to be part of a human-in-the-loop model evaluation, particularly to address challenges with LLMs, such as biases or missing dimensions. Let us first discuss how SMART addresses some issues by design and then perform an experimental assessment.
▶ **Using data to mitigate LLM challenges**: We use data in two ways *(i) Data usage in the generation of hypotheses:* Before generating explicit hypotheses of where the model is likely to fail, we provide the LLM with additional information about the data description and model failures of the training dataset. The hypotheses sampled are therefore reflective of the inductive bias of the LLM as well as being conditioned on the data itself; *(ii) Data usage in falsifying hypotheses:* core to SMART is the self-falsification mechanism where we iteratively generate hypotheses and test them on a validation dataset. Data is therefore used to filter out hypotheses which are not supported by the data. Hence, even if "incorrect" hypotheses about group failures are proposed, this step ensures they are discarded.
▶ **SMART provides clear and transparent testing**: This is done in two ways: *(i) SMART's model reports:* that document specific failure cases with natural language justifications (see Appendix D.10 for examples). Automatically generated reports can be a useful tool for humans-in-the-loop experts to audit and validate the testing process, such as evaluating whether tests should be added or removed. For example, a domain expert (e.g. a clinician) could review the report to assess whether the identified failure modes are truly relevant and concerning in that specific context. *(ii) Tests include justifications:* The justifications for the tests allow human users to inspect the model's testing procedures, understand the reasons, and audit for biases or missed dimensions.

**Goal.** Going beyond the mitigation strategies by design, we also assess SMART's robustness to prior biases. Specifically, we assess common ethnicity related biases of LLMs.

**Setup.** We use the same data generating process as Sec. 5.3.

We train a predictor function $\hat{f}$ and simulate a scenario where we intentionally corrupt a model's predictions for a proportion $\tau$ of a minority subgroup ("white" or "black" ethnicity).

Table 5: Proportion of times the corrupted minority subgroup is correctly identified as the top underperforming subgroup.

|  | Corrupted White | | Corrupted black | |
|---|---|---|---|---|
| $\tau$ | $P_{white}$ | $P_{black}$ | $P_{white}$ | $P_{black}$ |
| 0.01 | $0.78 \pm 0.04$ | $0.15 \pm 0.04$ | $0.16 \pm 0.04$ | $0.78 \pm 0.04$ |
| 0.02 | $0.88 \pm 0.03$ | $0.08 \pm 0.03$ | $0.10 \pm 0.03$ | $0.87 \pm 0.03$ |
| 0.05 | $0.98 \pm 0.01$ | $0.00 \pm 0.00$ | $0.00 \pm 0.00$ | $0.98 \pm 0.01$ |
| 0.10 | $0.98 \pm 0.01$ | $0.00 \pm 0.00$ | $0.00 \pm 0.00$ | $0.99 \pm 0.01$ |
| 0.20 | $0.98 \pm 0.01$ | $0.00 \pm 0.00$ | $0.00 \pm 0.00$ | $1.00 \pm 0.00$ |
| 0.30 | $1.00 \pm 0.00$ | $0.00 \pm 0.00$ | $0.00 \pm 0.00$ | $1.00 \pm 0.00$ |
| 0.50 | $1.00 \pm 0.00$ | $0.00 \pm 0.00$ | $0.00 \pm 0.00$ | $1.00 \pm 0.00$ |

$\tau$ ranges from 0 (no corruption) to 1 (completely random predictions for the subgroup). SMART aims to identify the top underperforming subgroup without bias based on historical patterns. We measure the proportion of times the top subgroup contains the "white" ($P_{\text{white}}$) or "black" ($P_{\text{black}}$) minority, averaged over 20 runs and 5 seeds, separately corrupting "white" and "black" ethnicities.

**Analysis.** We show in Table 5 that SMART is able to identify where models underperform even in scenarios such as ethnicity bias, where LLMs exhibit prior biases from the training dataset. This links to the above discussion that SMART mitigates such biases by design both using the training dataset to guide hypothesis generation and using the self-falsification mechanism to empirically evaluate hypotheses (and discard those that aren't reflective of the data).

**Takeaway 4.** SMART mitigates potential biases in the LLM, both by using the real data to guide hypothesis generation, as well as using the self-falsification mechanism to filter spurious hypotheses.

## 6 Discussion and limitations

Responding to recent calls for better model testing [36, 6], we formalize **Context-Aware Testing**; a new testing paradigm, actively seeking out relevant and likely model failures based on contextual awareness — going beyond data alone. We develop **SMART Testing**, using LLMs to hypothesize likely and relevant model failures providing *improved* and *automated* testing of tabular ML models, compared to data-only methods in various scenarios.

**Model reports.** SMART produces comprehensive and automated model reports documenting failure cases and justifications, thereby providing data scientists, ML engineers and stakeholders increased visibility into model failures. We provide an example SMART report in Appendix D.10.

**Practical considerations**. Given the potential utility of SMART we highlight the following five practical considerations: ▶*Hypothesis generation.* While CAT offers a principled framework for context-guided testing, LLMs present challenges in hypothesis generation. Although SMART has mechanisms to address these (see Sec. 5.4), it cannot guarantee the absence of biases. ▶*Use with small datasets.* SMART may be limited in some cases by insufficient real data to operationalize and test hypotheses, suggesting future work could explore targeted data collection or synthetic data generation to enhance testing. ▶ *Extensions to other modalities.* SMART is formalized to test tabular ML models, due to the interpretable and structured features in tabular data to guide hypothesis generation. While extensions to other modalities such as image and text is beyond the scope of this work, this is a promising future research direction that would require addressing the lack of explicitly interpretable features possibly via external metadata and developing new ways to operationalize hypotheses on unstructured data. That said, tabular data is ubiquitous in real-world applications [37, 38] with approximately 79% of data scientists working on tabular problems daily, vastly surpassing other modalities [39, 40]. This highlights the immediate impact and relevance of SMART. ▶ *Need for interpretable/meaningful feature names.* Feature names play an important role in finding model failures (see Appendix Appendix D.9). The need for interpretable/meaningful feature names (e.g. column labels such as sex, age, race etc) as a source of context is similar to human requirements of interpretable feature names to understand what the data refers to. While feature names are typically the de facto in research and industry datasets, in the rare occasions they are not, this will affect the performance of SMART. ▶ *Cost of SMART.* SMART is extremely accessible and cheap to use, approximately <0.10 USD for 5 hypotheses and <0.5 USD for 100 hypotheses for state-of-the-art models (see Appendix D.7).

**Broader impact**. Better testing practices can help ensure models are reliable, safe, and beneficial before being deployed in real-world applications. We hope that our work can help mitigate model testing risks in real-world applications as well as spur new testing regimes which are *context aware*.

## Acknowledgments and Disclosure of Funding

We would like to thank the reviewers, Fergus Imrie, Nicolas Astorga, Kasia Kobalczyk, Tennison Liu and Andrew Rashbass for their helpful feedback. PR is supported by GSK, ML is supported by AstraZeneca, NS by the Cystic Fibrosis Trust. This work was supported by Microsoft's Accelerate Foundation Models Academic Research initiative.

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

# Appendix: Context-aware testing: a new paradigm for testing with large language models

## Table of Contents

# A  Extended related work

## A.1  Enhanced overview of relevant literature

This work primarily engages with works on ML model testing. Consequently, we detail different paradigms of ML model testing next, which we summarize in Table 6 and Figure 5 showing that none of the existing paradigms satisfy all the properties for ML testing neither in terms of automation nor from the perspective of relevance as they are unable to incorporate context and/or requirements.

Table 6: Comparison of ML testing paradigms in terms of how tests are defined, the type of search space (i.e. relevance based on context and requirements), sensitivity to multiple testing and automation to enable scalability.

| | | | Search space | | | | |
|---|---|---|---|---|---|---|---|
| Paradigm | Objective | Test definition | Context-aware | Integrate Requirements | Automated testing | Outputs | Examples |
| Average testing | Overall performance | Data split | N.A. | N.A. . | ✔ | Single score | |
| Behavioral | Test expected behavior | Pre-defined (Human) | N.A. | N.A. | ✗ | Multi-dimensional score | [16, 17, 5] |
| Data-only | Identify divergent groups | Search | ✗ | ✗ | ✔ | Multi-dimensional score | [14, 21, 15, 12, 13] |
| **SMART (Ours)** | Probe for model failure | Search | ✔ | ✔ | ✔ | Multi-dimensional score justifications, report | |

**Existing paradigms in ML model testing.** The ML community has predominatly approached ML model evaluation/testing via the use of held-out test datasets. On the basis of the test dataset a single performance metric (accuracy, AUC etc) is computed. This single average evaluation may mask nuances of the model's performance along various dimensions. One approach to address this is to create better benchmark datasets when evaluating models on common benchmarking tasks. For example, manual corruptions like Imagenet-C [41] or by collecting additional real data such as the Wilds benchmark [8]. Benchmark datasets are labor-intensive to collect and their utility is limited to the specific benchmark tasks.

What if we want to evaluate models not confined to benchmarking tasks? To test in cases beyond benchmark tasks, the community has proposed trying to find slices or regions wherein the model fails (i.e. via stress tests). As mentioned in Sec. 1, this could be ► **Behavioral testing**: which requires human expertise and intuition to define the test scenarios (e.g. Checklist [16], HateCheck [17] or 3S-Testing [5]) — which is not automated and does not scale. Moreover, it runs a high risk of overlooking critical weaknesses due to human cognitive biases. ► **Data-only testing**: which does not account for context or requirements and searches exhaustively (e.g. Autostrat [14], SliceFinder [12] or DivExplorer [15]). This may slices focus on arbitrary, less important, or unrealistic/implausible scenarios that are unlikely to be seen in reality. Moreover, we run the risk of the multiple testing problem, where by virtue of the large number of tests evaluated, we might discover a divergent group by chance.

**Hypothesis-driven ML model testing.** In contrast, a hypothesis-driven approach to ML model testing brings about the falsifiability approach widely adopted in science. It begins with the formulation of specific, testable hypotheses based on theoretical understanding, context/domain knowledge, and the intended application of the model (i.e. requirements). The concept of hypothesis-driven ML model testing is deeply rooted in work by Popper [28] who proposed the principle of falsifiability as a driver of scientific progress. The progression of knowledge hinges on the formulation and rigorous testing of hypotheses which can either be falsified or supported by empirical evidence. In the context of hypotheses in ML model testing, testable statements about the model's performance under various conditions, fairness, and robustness can improve our understanding of the model's performance. By rigorously testing these hypotheses, we can uncover the strengths and limitations of ML models.

**Contrast to data-only methods on unstructured data.**

In this paper, we have discussed data-only methods for slice discovery or blindspot discovery applicable to structured data. Specifically, we focus on tabular data where metadata in the form of column names is explicitly encoded into the data. The data-only approaches covered in this work directly search over the raw feature space to identify slices with similar attributes wherein the model would exhibit underperforming predictions.

For completeness, we contrast to data-only approaches often applicable to data without explicit structure and metadata (images, text, audio). There have been numerous methods including [42–45]

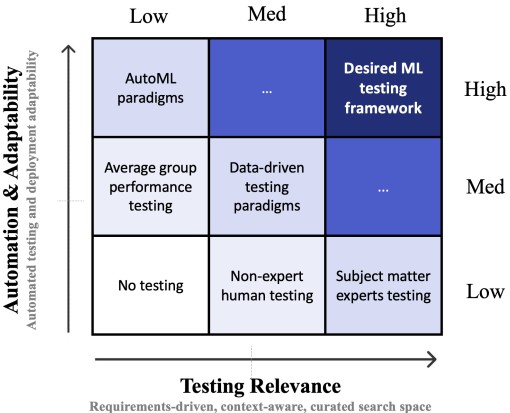

Figure 5: Contrasting paradigms of ML model testing along (i) Testing relevance which accounts for requirements and/or context and (ii) Degree of automation and adaptability when carrying out testing. We desire a new paradigm of ML testing to address both.

to identify slices of unstructured data with systematic failures. They all follow a similar pattern: (1) the data is embedded (often with a pre-trained) model into a representation space and (2) the underperforming slices are identified by clustering on the raw embedding space or after dimensionality reduction. We then need to post-hoc interpret the clusters in order to understand what they represent. This contrasts the tabular data setting where we find cohesive groups of features that provide an explicit interpretation. Moreover, in tabular regimes, we do not have access to pre-trained models in the same way as for domains such as images or text.

Specifically, to context-aware testing we also note that tabular data inherently includes context through interpretable/meaningful feature names and metadata. This context is not naturally present in images (which are tensors of pixel intensities). Hence, extending CAT to other domains like images or text would require incorporating metadata to provide necessary context — which is often unavailable. In addition, one would need to develop new ways to operationalize hypotheses on unstructured data.

**Contrast to software testing.**

Going beyond ML model testing, the idea of testing is also prevalent in software systems. For example, unit tests of functions in a codebase. We highlight that software testing of functional input-output correctness is also an area proposed in software testing, which could also theoretically be applied to ML systems. That said, we contrast between SMART and these paradigms in Table 7 below, demonstrating that we tackle a different testing problem.

Table 7: Comparison of SMART testing with software testing works

| Criteria | SMART | Christakis et al. [18] | Sharma et al [19] | How is SMART different? |
|---|---|---|---|---|
| Test case generation | Automatic LLM-based test cases | Pre-Specified dimensions of functional correctness (k-safety properties) | Approximates the black-box model to test with a white-box decision tree model. | SMART uses the LLM to automatically hypothesize the test cases & execute them |
| What is the focus of testing | Identify model failures | Assesses I/O functional correctness | SMT solvers find violations to monotonicity properties . i.e. input increase!=output increase | SMART failures encompass a broader range of possibilities |
| Are tests context-aware? | Yes | No | No | SMART uniquely incorporates context awareness into testing, leading to more realistic assessments. |

Additionally, a further orthogonal area to ML model testing is that of data validation [46, 47]. In contrast, to testing an ML model for failures, data validation aims to test input data pipelines for data quality problems or drift.

## A.2 Components of the ML Testing pipeline

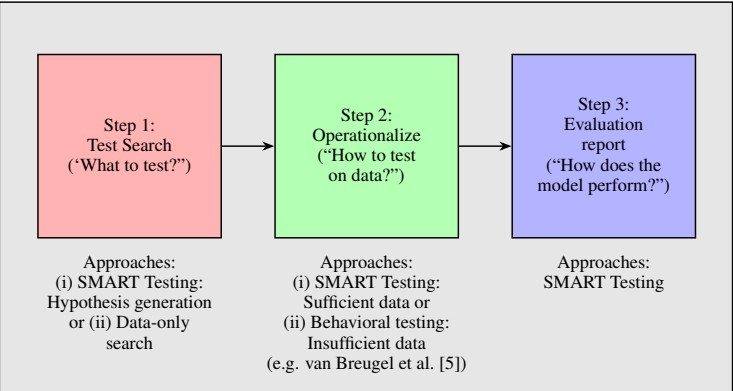

Figure 6: Components of ML Testing — (i) Test search, (ii) Operationalize, (iii) Evaluation report, with example approaches for each component.

- Test search: First, we need to decide "what to test". This is a search problem to identify test dimensions. This could either be done via SMART which is targeted (via context and requirements) or data-only methods (where the space is larger). Alternatively, if human experts are available, humans could define the test dimensions.

- Operationalize: Second, we need to operationalize the test and address the challenge of "how to carry out this test on data". If we have sufficient data — SMART could operationalize the tests on the data via an interpreter. Alternatively, if we don't have sufficient data to run the test — then SMART could be augmented by behavioral testing approaches such as van Breugel et al. [5] — which once the test has been defined use synthetic data to augment small subgroups/slices.

- Evaluation report: Third, we carry out the test and evaluate the model to answer the question, "how does the model perform'. SMART can be used to produce a comprehensive report of failures and justifications in an automated manner.

## A.3 Comparison of the features of slice discovery methods

Even as we zoom into the task of discovering slices where the model might underperform, we observe that SMART has features which are not supported by most of other discovery methods. We exemplify some of these features in Table 8

Table 8: Comparison of slice discovery methods

| Criteria | SliceFinder | Pysubgroup | DivExplorer | Autostrat | SMART Testing |
|---|---|---|---|---|---|
| Integrates custom domain knowledge | ✗ | ✗ | ✗ | ✗ | ✔ |
| Constant slice discovery time | ✗ | ✗ | ✗ | ✗ | ✔ |
| Performance is resistant to irrelevant data | ✗ | ✗ | ✗ | ✗ | ✔ |
| Can capture rare slices | ✗ | ✗ | ✗ | ✗ | ✔ |
| Inherently supports logical ORs | ✗ | ✗ | ✗ | ✔ | ✔ |
| Resistant to overfitting the training set | ✗ | ✗ | ✗ | ✗ | ✔ |

# B SMART Details

We present a block diagram of the key components of SMART Testing in Figure 7. In addition, for each component we provide additional motivations and technical details not covered in the main paper. We further provide more technical information on certain implementation details.

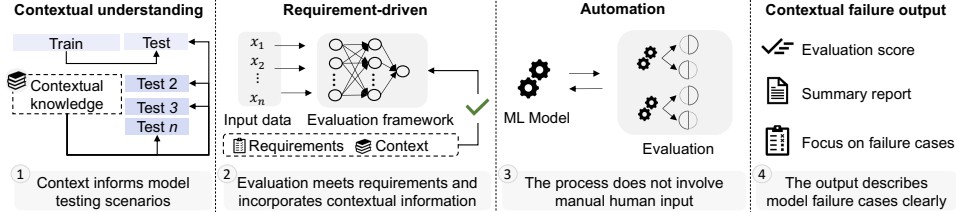

Figure 7: A strong machine learning testing framework should incorporate textual understanding during testing, should meet requirements, should be automated, and provide contextual reports with an emphasis on model failures.

## B.1 Contextual understanding

A core component of SMART is leveraging LLMs to strategically identify *what to test for* in tabular ML models. This process is anchored on the premise that LLMs can effectively navigate the space of potential model failure slices, harnessing their contextual understanding to pinpoint slices where the model is most susceptible to failure. The operation of LLMs within the SMART framework involves three critical inputs:

1. **Context** ($\mathcal{C}$): A string describing the overarching scenario and the task at hand.
2. **Dataset Information** ($\mathcal{D}_c$): Extracted from the training data, this includes a description of the observations where the model did not fail ($\hat{y} = y$), or where it failed $\hat{y} \neq y$). This could be characterized as a string with a description of the covariates of each group (e.g., mean, median, mode, and a textual description of the distribution).

Utilizing these inputs, the LLM generates a set of hypotheses ($\mathcal{H}$) and corresponding justifications ($\mathcal{J}$) regarding potential model failures. The framework also incorporates a *self-refine mechanism* to enhance hypothesis generation. This mechanism iteratively refines hypotheses based on the observed or provided context ($\mathcal{C}$) and data ($\mathcal{D}_c$), re-ranking them by their likelihood. The self-refine mechanism is introduced in order to generate hypotheses that are more likely to target specific model failures.

## B.2 Operationalizing variables

Once the hypotheses have been proposed, it is important to operationalize them. This operationalization can be achieved through one of two methods:

1. **LLM-based Operationalization**: Hypothesizing possible operationalizations (e.g., $age > 70$ as a way to operationalize "elderly people"). In this case, the previous interactions and information is provided to the LLM, together with the relevant description of the data. It is then asked to provide possible ways to operationalize a specific hypothesis. This is done by passing an "operationalization prompt" which contains the aforementioned information.

The following is an example of an operationalization prompt in Python:

```python
f"""
The following are hypotheses about which people within
a dataset the model might underperform on.
Propose specific ranges for each hypothesis. Hypotheses
: {hypotheses}.

Dataset information: {context}. {context_target}

The dataset contains {len(unique_values)} columns. The
columns are {', '.join(unique_values.keys())}. The values
are {str(unique_values.items())}

TASK: Propose specific variable ranges for each
hypothesis such that they are clearly operationalizable and
defined. Use this format: Hypothesis: <>; Operationalization
: <>.
"""
```

Code Listing 2: Operationalization (LLM knowledge): General template

This is then converted into a specific operationalization for each hypothesis using an external compiler which maps the strings to a function.

2. **Data-driven Operationalization**: Utilizing training data to identify optimal splits for given covariates (e.g., $age >= 82.32$ as a condition to split data based on the "age" covariate). This method receives the covariate (or set of covariates) as an input and returns the optimal split. The optimal split is defined as the split which can identify two groups which have the largest absolute difference in accuracies given some requirements (e.g. a minimum or maximum group size in each group). This splitting can be done by *any black box splitting function* which takes in a set of inputs and returns a splitting mechanism.

**Black-box splitting function for data-driven operationalization**. While any black-box splitting function could perform on the data, we provide details on the specific function used in SMART.

The function implemented is based on a decision tree model, which can be either a regressor or a classifier, depending on the nature of the outcome variable. In all our experiments, we perform classification-based tasks.

The function operates by fitting a decision tree to the data and recursively traversing the tree to find the split that yields the largest absolute difference in the outcome variable's mean value between two slices. The split must also satisfy group size constraints, specified as minimum and maximum group sizes. The traversal process evaluates each potential split, calculating the mean outcome for each slice and the discrepancy between these means. The optimal split is the one that maximizes this discrepancy while adhering to the group size requirements.

The function returns a query string that represents this optimal split. This string can then be used to segment the dataset into the identified slices for further analysis or testing. The following is pseudocode for the splitting algorithm.

---

**Algorithm 1** Optimal splitting mechanism within the SMART framework

---

**Data:** $dataframe$, $features$, $outcome$, $min\_group\_size$, $max\_group\_size$

**Result:** Optimal query string for data split

**1** **Function** GetOptimalSplitQuery($dataframe$, $features$, $outcome$, $min\_group\_size$, $max\_group\_size$):

**2**     Validate input features and outcome in $dataframe$   Determine the type of decision tree model based on $outcome$ type   Fit the decision tree model to $dataframe$ using $features$ and $outcome$   Initialize an empty list $conditions$ for tracking split conditions   **return** TraverseTree($root$, 0, $conditions$)

**3** **Function** TraverseTree($node, depth, conditions$):

**4**     **if** $node\ is\ a\ leaf$ **then**

**5**        Calculate the discrepancy in $outcome$ between the two slices   **return** the condition and discrepancy if group size constraints are met

**6**     **end**

**7**     Determine left and right conditions based on the threshold at $node$   **return** TraverseTree($node.left$, $depth + 1$, $conditions \cup \{left\_condition\}$)    // or the right conditions, depending on which has the greater discrepancy

---

This splitting mechanism is used for continuous features. A separate black-box splitting mechanism is developed for categorical features based on iterating on different permutations of these features and evaluating them that helps with variable selection.

In practice, we employ the LLM-based operationalization for the SMART ablation and the data-driven operationalization within the original SMART framework. However, we highlight that this is a design choice that we have found works in practice; there is nothing stopping from using any operationalization framework within the main SMART framework.

## B.3   Feasibility checks

SMART is built on many modules which can be toggled on or off. One of such modules is a module called "feasibility check" which evaluates whether there are any hypotheses which should be tested in the first place. The experiments presented in Sec. 5.1 highlight the importance of being able to identify when no relationship between covariates exist.

The feasibility check contains three steps. First, an LLM is queried to evaluate whether any relationships could exist between the covariates and an outcome variable. For instance, this could be whether a relationship could exist between loan default and the annual rainfall in a given region. Second, the answer is self-refined. This helps to evaluate feasibility because, in practice, initial responses tend to be over-optimistic (such as hypothesizing that annual rainfall is associated with loan default via a geographic proxy). This self-refinement helps to critically evaluate the previous answer. The number of steps in the self-refinement process is a hyperparameter. Lastly, the answer (whether or not a relationship could exist between the two variables and, hence, should be inspected) is converted to a boolean value via an external function.

The following is pseudocode which implements the feasibility module.

---

**Algorithm 2** Feasibility for evaluating slices

---

**Input:** $unique\_values$, $context$, $context\_target$, $system\_message$, $n\_refine$
**Output:** $feasibility\_boolean\_response$

8 **Function** `FeasibilityCheck`($unique\_values$, $context$, $context\_target$, $system\_message$, $n\_refine$):

    `// Construct the feasibility task prompt`

9      $task \leftarrow$ "Evaluate subgroups for model performance" $task \leftarrow task \cup$ "Context: " $\cup\ context\ \cup$ "Target: " $\cup\ context\_target\ \ task \leftarrow task \cup$ "Columns: " $\cup$ `join`($unique\_values.keys()$)

    `// Get initial feasibility response`

10    $feasibility\_response \leftarrow$ `GetLLMResponse`($task$, $system\_message$)

    `// Refine the answer`

11    $feasibility\_response \leftarrow$ `SelfRefine`($unique\_values$, $context$, $context\_target$, $feasibility\_response$, $system\_message$, $n\_refine$)

    `// Convert to boolean`

12    $boolean\_task \leftarrow$ "Based on analysis, provide yes/no answer" $boolean\_task \leftarrow boolean\_task \cup$ "Analysis: " $\cup\ feasibility\_response$

13    $feasibility\_boolean\_response \leftarrow$ `GetLLMResponse`($boolean\_task$)

---

## B.4 Data adjustment queries

Given that SMART operates with an LLM, sometimes the framework outputs proposals which do not operationalize on the data well. For instance, even if a column "age" is a categorical variable, SMART might propose to operationalize the hypothesis "elderly people" as $age > 72$ which would cause an error.

To avoid this, we implement an additional data adjustment module which can handle such cases. It catches the error and re-prompts the LLM to find a group which could be operationalized given the data structure, and does so iteratively until such a group is found.

The following is a pseudocode function that explains how data adjustment is performed.

---

**Algorithm 3** Pseudocode for adjusting subgroup/slice queries

---

**Data:** X (DataFrame), n_subgroups (integer)
**Result:** Adjusted subgroup queries in the dataset

14 **Function** `AdjustSubgroupQuery`($X$, $n\_subgroups$):

15    **for** *each subgroup condition* **do**

16        **if** `CheckQueryExistence`($X$, *condition*) **then**

17            `UpdateSubgroupCondition`(*condition*)

18        **else**

            `// Condition yields no rows, adjust it`

19            $adjusted\_condition \leftarrow$ `GetLLMResponse`(*condition*) `// Update the condition in the subgroup`

20            `UpdateSubgroupCondition`(*adjusted_condition*)

21        **end**

22    **end**

23 **return**

---

## B.5 Requirements, automation, and outputs

**Requirements**. SMART can natively integrate user requirements into its framework. This is done by inputting requirements as a string and concatenating it together with the context. Some requirements are directly integrated into the framework itself (e.g. functionalities for determining the minimum or maximum sample size of a data split).

**Automation**. Fig. 1 showcases the pipeline of SMART and which components are automated. SMART is developed using an "sklearn" style fit, predict framework. The fit method automatically performs a feasibility check, generates hypotheses, justifications, operationalizes them, performs *self-falsification* using empirical data, re-ranks the hypotheses, and saves all intermediate results. The

method can then automatically be used on any piece of data to evaluate whether it underperforms on the groups that have been found to underperform.

**Outputs**. In addition to SMART outputting subgroups/slices or a scalar number, it can output a model report. An example report is provided in D.10. We note that this report is simply an example which employs both the findings of the fitting procedure and an additional LLM to summarize the outputs. More fine-grained outputs can be constructed.

### B.6 Moving outside of IID data

In the paper, we propose that SMART can move outside of IID data and generalize better in the presence of covariate shift (refer to Sec. D.2). We highlight that this is done by performing an analysis on the original data and using an LLM propose possible hypotheses and splits for a different target domain where no data is present (hence, operationalizing only using the LLM with access to previous operationalizations). The following is an example prompt which is designed to do this.

```
f"""
        You have access to the following information.

        Dataset information: {context}. {context_target}

        The dataset contains {len(unique_values)} columns. The columns
    are {', '.join(unique_values.keys())}. The values are {str(
    unique_values.items())}

        However, you are no longer working with the same data as just
    described. Rather, this is the context: {new_context}.

        These are the hypotheses: {self._updated_hypotheses}.

        TASK: Propose specific variable ranges for each hypothesis
    such that they are clearly operationalizable and defined. Use this
     format: Hypothesis: <>; Operationalization: <>.
        """
```

Code Listing 3: Operationalization (LLM knowledge): General template

### B.7 SMART and multiple testing

The reason why SMART performs testing via hypothesis generation is because testing for all slices is equivalent to generating and testing a hypothesis on the data. Here, we outline in greater detail how hypothesis generation is connected to multiple testing.

As outlined in the main manuscript, searching for $f$ failures may bring up the challenge of multiple hypothesis testing. Specifically, when we evaluate the failure rate of model $f$ across different slices $\mathcal{S}_i \subseteq \mathcal{D}$, we are testing the null hypothesis $H_0^{(i)} : \mu_{\mathcal{S}_i} = \mu_{\mathcal{D}}$ against the alternative $H_1^{(i)} : \mu_{\mathcal{S}_i} \neq \mu_{\mathcal{D}}$. Then, the probability of making a Type I error increases with each test. This drastically inflates the family-wise error rate (FWER). For instance, assuming that each slice is independent, the probability of making one or more Type I errors across all tests is given by $1 - (1 - \alpha)^m$, where $m$ is the total number of slices tested. While this can be addressed by adjusting for multiple testing, we run into the trade-off between the FWER control and statistical power. As we employ statistical correction methods to account for Type I errors, we increase the probability of Type II errors.

# C  Benchmarks & Experimental Details

We summarize all experimental details, datasets and benchmarks.

Code can be found at: `https://github.com/pauliusrauba/SMART_Testing` or `https://github.com/vanderschaarlab/SMART_Testing`

## C.1  Datasets

We summarize the different datasets we use in this paper in Table 9. The datasets vary in number of samples, number of features and domain.

Table 9: Summary of the datasets used.

| Name | $n$ samples | $n$ features | Domain | Task |
|---|---|---|---|---|
| Loan | 5000 | 8 | Finance | Loan default |
| Breast | 116 | 8 | Healthcare/Medicine | Breast cancer |
| Compas [33] | 4743 | 8 | Criminal justice | Criminal recidivism |
| OULAD [34] | 5000 | 8 | Education | Exam score |
| Diabetes | 768 | 8 | Healthcare/Medicine | Diabetes label |
| SEER [35] | 2000 | 12 | Healthcare/Medicine | Prostate cancer morality |
| CUTRACT [35] | 20000 | 12 | Healthcare/Medicine | Prostate cancer morality |

Note that, depending on the experiment, we might vary the sample size or features used. This can be because of the added synthetic variables or certain slice discovery algorithms taking too long to run.

## C.2  Baselines

We compare SMART testing with the following data-only baselines in Section 5.

- Autostrat [14]. We use the implementation from [7]
- Pysubgroup [21]. We use the implementation from [8]
- Divexplorer [15]. We use the implementation from [9]
- Slicefinder [12]. We use the implementation from [10]
- Sliceline [13]. We use the implementation from [11]

## C.3  LLM prompts

### C.3.1  Generic templates.

```
1 """Constructs the task description for the LLM."""
2 task = f"""Your task is to propose possible hypotheses as to which
     subgroups within the dataset might have worse predictive
     performance than on average because of societal bias in the
     dataset, insufficient data, other relationships, or others. The
     subgroups might be based on any of the provided characteristics,
     as well as on any combination of such characteristics.
3
4 Dataset information: {context}. {context_target}
5
6 The dataset contains {len(unique_values)} columns. The columns are {',
     '.join(unique_values.keys())}.
7
```

---

[7] `https://github.com/Adebayo-Oshingbesan/BPS_Autostrat_Code/`
[8] `https://github.com/flemmerich/pysubgroup`
[9] `https://github.com/elianap/divexplorer`
[10] `https://github.com/yeounoh/slicefinder`
[11] `https://github.com/DataDome/sliceline`

```
8  Task: Create {n} hypotheses as to which subgroups within the dataset
       the model will perform worse than on average because of societal
       biases or other reasons. Important: Your hypothesis can contain
       either one variable or two variables in the condition. Therefore,
       your goal is to find discrepancies in the model's performance, not
        the underlying data outcomes. Justify why you think that for each
        of the {n} hypotheses. Format the output: Hypothesis: <>;
       Justification: <>.
9
10 """
```

Code Listing 4: Generic hypothesis prompt

```
1  """
2  The following are hypotheses about which people within a dataset the
       model might underperform on.
3  Propose specific ranges for each hypothesis. Hypotheses: {hypotheses}.
4
5  TASK: return a dictionary that contains an index number as the key and
        the column value as the value. If there are multiple columns in
       that hypothesis, return them in a list. There are the column names
       : {', '.join(unique_values.keys())}.
6  """
```

Code Listing 5: Generic operationalization prompt

### C.3.2 Example prompts: OULAD Education.

```
1  """
2  ----------INPUT TEXT --------------
3  Your task is to propose possible hypotheses as to which subgroups
       within the dataset might have worse predictive performance than on
        average because of societal bias in the dataset, insufficient
       data, other relationships, or others. The subgroups might be based
        on any of the provided characteristics, as well as on any
       combination of such characteristics.
4
5  Dataset information:
6  Open University Learning Analytics Dataset (OULAD) contains data about
        courses, students and their interactions with Virtual Learning
       Environment (VLE) for seven selected courses (called modules).
       Presentations of courses start in February and October - they are
       marked by B and J respectively. The dataset consists of tables
       connected using unique identifiers. All tables are stored in the
       csv format.
7
8  The task is to predict the students who are likely to fail or withdraw
        from a course.
9
10
11 The dataset contains 14 columns. The columns are gender, region,
       highest_education, imd_band, age_band, num_of_prev_attempts,
       studied_credits, disability, test, group_0, group_1, group_2,
       group_3, group_4. The values are dict_items([('gender', ['M', 'F
       ']), ('region', ['South Region', 'Scotland', 'Yorkshire Region', '
       West Midlands Region', 'East Anglian Region', 'East Midlands
       Region', 'South East Region', 'Wales', 'North Western Region', '
       London Region', 'South West Region', 'Ireland', 'North Region']),
       ('highest_education', ['Lower Than A Level', 'A Level or
       Equivalent', 'HE Qualification', 'No Formal quals', 'Post Graduate
        Qualification']), ('imd_band', ['70-80%', '40-50%', '0-10%',
       '10-20', '50-60%', '20-30%', '60-70%', '80-90%', '90-100%',
       '30-40%', nan]), ('age_band', ['0-35', '35-55', '55<=']), ('
       num_of_prev_attempts', [0, 1, 2, 3, 5, 4, 6]), ('studied_credits',
        {'min': 30, 'mean': 79.77122804326149, 'max': 655}), ('disability
```

```
     ', ['N', 'Y']), ('test', [False, True]), ('group_0', [False, True
         ]), ('group_1', [True, False]), ('group_2', [True, False]), ('
         group_3', [False, True]), ('group_4', [False, True])])

12
13  Task: Create 5 hypotheses as to which subgroups within the dataset the
         model will perform worse than on average because of societal
         biases or other reasons. Therefore, your goal is to find
         discrepancies in the model's performance, not the underlying data
         outcomes. Justify why you think that. Format of the output:
         Hypothesis: <>; Justification: <>.
14  """
15
16  ----------OUTPUT TEXT --------------
17  Hypothesis 1: Students with disabilities will have worse predictive
         performance than average.
18  Justification: Students with disabilities may face additional
         challenges in their learning process, which could affect their
         performance in the course. They may require additional support or
         accommodations that are not adequately captured in the dataset,
         leading to lower predictive performance for this subgroup.
19
20  Hypothesis 2: Students from lower socioeconomic backgrounds (indicated
         by imd_band) will have worse predictive performance than average.
21  Justification: Students from lower socioeconomic backgrounds may have
         limited access to resources and support systems that are necessary
          for successful completion of the course. This lack of resources
         and support may not be adequately captured in the dataset, leading
          to lower predictive performance for this subgroup.
22
23  Hypothesis 3: Students with lower levels of education (indicated by
         highest_education) will have worse predictive performance than
         average.
24  Justification: Students with lower levels of education may have less
         prior knowledge and skills that are necessary for success in the
         course. This lack of foundational knowledge may not be adequately
         captured in the dataset, leading to lower predictive performance
         for this subgroup.
25
26  Hypothesis 4: Students who have attempted the course multiple times (
         indicated by num_of_prev_attempts) will have worse predictive
         performance than average.
27  Justification: Students who have attempted the course multiple times
         may have encountered difficulties or challenges in their previous
         attempts that hindered their success. These challenges may not be
         adequately captured in the dataset, leading to lower predictive
         performance for this subgroup.
28
29  Hypothesis 5: Students in certain regions (indicated by region) will
         have worse predictive performance than average.
30  Justification: Students in certain regions may have different
         educational systems, resources, and support networks that can
         impact their performance in the course. These regional differences
          may not be adequately captured in the dataset, leading to lower
         predictive performance for subgroups based on region.
```

Code Listing 6: Hypothesis generation: OULAD Dataset

```
1  """
2  ----------INPUT TEXT --------------
3
4  The following are hypotheses about which people within a dataset the
         model might underperform on.
5  Propose specific ranges for each hypothesis. Hypotheses: Hypothesis 1:
         Students with disabilities will have worse predictive performance
         than average.
```

```
 6
 7  Justification: Students with disabilities may face additional
        challenges in their learning process, which could affect their
        performance in the course. They may require additional support or
        accommodations that are not adequately captured in the dataset,
        leading to lower predictive performance for this subgroup.
 8
 9  Hypothesis 2: Students from lower socioeconomic backgrounds (indicated
         by imd_band) will have worse predictive performance than average.
10  Justification: Students from lower socioeconomic backgrounds may have
        limited access to resources and support systems that are necessary
         for successful completion of the course. This lack of resources
        and support may not be adequately captured in the dataset, leading
         to lower predictive performance for this subgroup.
11
12  Hypothesis 3: Students with lower levels of education (indicated by
        highest_education) will have worse predictive performance than
        average.
13  Justification: Students with lower levels of education may have less
        prior knowledge and skills that are necessary for success in the
        course. This lack of foundational knowledge may not be adequately
        captured in the dataset, leading to lower predictive performance
        for this subgroup.
14
15  Hypothesis 4: Students who have attempted the course multiple times (
        indicated by num_of_prev_attempts) will have worse predictive
        performance than average.
16  Justification: Students who have attempted the course multiple times
        may have encountered difficulties or challenges in their previous
        attempts that hindered their success. These challenges may not be
        adequately captured in the dataset, leading to lower predictive
        performance for this subgroup.
17
18  Hypothesis 5: Students in certain regions (indicated by region) will
        have worse predictive performance than average.
19  Justification: Students in certain regions may have different
        educational systems, resources, and support networks that can
        impact their performance in the course. These regional differences
         may not be adequately captured in the dataset, leading to lower
        predictive performance for subgroups based on region..
20
21  Dataset information:
22  Open University Learning Analytics Dataset (OULAD) contains data about
         courses, students and their interactions with Virtual Learning
        Environment (VLE) for seven selected courses (called modules).
        Presentations of courses start in February and October - they are
        marked by B and J respectively. The dataset consists of tables
        connected using unique identifiers. All tables are stored in the
        csv format.
23  .
24  The task is to predict the students who are likely to fail or withdraw
         from a course.
25
26
27  The dataset contains 14 columns. The columns are gender, region,
        highest_education, imd_band, age_band, num_of_prev_attempts,
        studied_credits, disability, test, group_0, group_1, group_2,
        group_3, group_4. The values are dict_items([('gender', ['M', 'F
        ']), ('region', ['South Region', 'Scotland', 'Yorkshire Region', '
        West Midlands Region', 'East Anglian Region', 'East Midlands
        Region', 'South East Region', 'Wales', 'North Western Region', '
        London Region', 'South West Region', 'Ireland', 'North Region']),
        ('highest_education', ['Lower Than A Level', 'A Level or
        Equivalent', 'HE Qualification', 'No Formal quals', 'Post Graduate
         Qualification']), ('imd_band', ['70-80%', '40-50%', '0-10%',
```

```
        '10-20', '50-60%', '20-30%', '60-70%', '80-90%', '90-100%',
        '30-40%', nan]), ('age_band', ['0-35', '35-55', '55<=']), ('
        num_of_prev_attempts', [0, 1, 2, 3, 5, 4, 6]), ('studied_credits',
         {'min': 30, 'mean': 79.77122804326149, 'max': 655}), ('disability
        ', ['N', 'Y']), ('test', [False, True]), ('group_0', [False, True
        ]), ('group_1', [True, False]), ('group_2', [True, False]), ('
        group_3', [False, True]), ('group_4', [False, True])])
28
29  TASK: Propose specific variable ranges for each hypothesis such that
        they are clearly operationalizable and defined. Use this format:
        Hypothesis: <>; Operationalization: <>.
30  """
31
32  ----------OUTPUT TEXT--------------
33  Hypothesis 1: Students with disabilities will have worse predictive
        performance than average.
34  Operationalization: Students with disabilities can be identified by
        the "disability" variable in the dataset. To test this hypothesis,
         we can define the range for students with disabilities as those
        who have a value of "Y" in the "disability" column.
35
36  Hypothesis 2: Students from lower socioeconomic backgrounds (indicated
         by imd_band) will have worse predictive performance than average.
37  Operationalization: Socioeconomic background can be measured using the
         "imd_band" variable in the dataset. To test this hypothesis, we
        can define the range for students from lower socioeconomic
        backgrounds as those who have an "imd_band" value in the range of
        "0-30%".
38
39  Hypothesis 3: Students with lower levels of education (indicated by
        highest_education) will have worse predictive performance than
        average.
40  Operationalization: Education level can be determined using the "
        highest_education" variable in the dataset. To test this
        hypothesis, we can define the range for students with lower levels
         of education as those who have a value of "Lower Than A Level" or
        "No Formal quals" in the "highest_education" column.
41
42  Hypothesis 4: Students who have attempted the course multiple times (
        indicated by num_of_prev_attempts) will have worse predictive
        performance than average.
43  Operationalization: Previous attempts can be measured using the "
        num_of_prev_attempts" variable in the dataset. To test this
        hypothesis, we can define the range for students who have
        attempted the course multiple times as those who have a value
        greater than 1 in the "num_of_prev_attempts" column.
44
45  Hypothesis 5: Students in certain regions (indicated by region) will
        have worse predictive performance than average.
46  Operationalization: Region can be determined using the "region"
        variable in the dataset. To test this hypothesis, we can define
        the range for students in certain regions as those who belong to
        the regions of "North Region" or "Wales".
```

Code Listing 7: Operationalization (LLM knowledge): OULAD Dataset

```
1  """
2  ----------INPUT TEXT --------------
3
4  The following are groups that are defined based on the dataset.
        Convert them into a Python dictionary format. Each group should be
         represented as a key-value pair in the dictionary, where the key
        is an index (0 to 4), and the value is a string representing the
        group using Python syntax and logical operators. For multiple
```

```
        conditions, use Python's logical 'and' ('&&') or 'or' ('||').
        Ensure the format is a valid Python dictionary.
5
6  Examples:
7  - Single Condition: {0: 'X > 45'}
8  - Multiple Conditions: {1: '(X > 45) and (Y < 20)'}
9
10 Groups to summarize: Hypothesis 1: Students with disabilities will
        have worse predictive performance than average.
11
12 Operationalization: Students with disabilities can be identified by
        the "disability" variable in the dataset. To test this hypothesis,
         we can define the range for students with disabilities as those
        who have a value of "Y" in the "disability" column.
13
14 Hypothesis 2: Students from lower socioeconomic backgrounds (indicated
         by imd_band) will have worse predictive performance than average.
15 Operationalization: Socioeconomic background can be measured using the
         "imd_band" variable in the dataset. To test this hypothesis, we
        can define the range for students from lower socioeconomic
        backgrounds as those who have an "imd_band" value in the range of
        "0-30%".
16
17 Hypothesis 3: Students with lower levels of education (indicated by
        highest_education) will have worse predictive performance than
        average.
18 Operationalization: Education level can be determined using the "
        highest_education" variable in the dataset. To test this
        hypothesis, we can define the range for students with lower levels
         of education as those who have a value of "Lower Than A Level" or
         "No Formal quals" in the "highest_education" column.
19
20 Hypothesis 4: Students who have attempted the course multiple times (
        indicated by num_of_prev_attempts) will have worse predictive
        performance than average.
21 Operationalization: Previous attempts can be measured using the "
        num_of_prev_attempts" variable in the dataset. To test this
        hypothesis, we can define the range for students who have
        attempted the course multiple times as those who have a value
        greater than 1 in the "num_of_prev_attempts" column.
22
23 Hypothesis 5: Students in certain regions (indicated by region) will
        have worse predictive performance than average.
24 Operationalization: Region can be determined using the "region"
        variable in the dataset. To test this hypothesis, we can define
        the range for students in certain regions as those who belong to
        the regions of "North Region" or "Wales".
25 Column names: gender, region, highest_education, imd_band, age_band,
        num_of_prev_attempts, studied_credits, disability, test, group_0,
        group_1, group_2, group_3, group_4
26 Column values: dict_items([('gender', ['M', 'F']), ('region', ['South
        Region', 'Scotland', 'Yorkshire Region', 'West Midlands Region', '
        East Anglian Region', 'East Midlands Region', 'South East Region',
         'Wales', 'North Western Region', 'London Region', 'South West
        Region', 'Ireland', 'North Region']), ('highest_education', ['
        Lower Than A Level', 'A Level or Equivalent', 'HE Qualification',
        'No Formal quals', 'Post Graduate Qualification']), ('imd_band',
        ['70-80%', '40-50%', '0-10%', '10-20', '50-60%', '20-30%',
        '60-70%', '80-90%', '90-100%', '30-40%', nan]), ('age_band',
        ['0-35', '35-55', '55<=']), ('num_of_prev_attempts', [0, 1, 2, 3,
        5, 4, 6]), ('studied_credits', {'min': 30, 'mean':
        79.77122804326149, 'max': 655}), ('disability', ['N', 'Y']), ('
        test', [False, True]), ('group_0', [False, True]), ('group_1', [
        True, False]), ('group_2', [True, False]), ('group_3', [False,
        True]), ('group_4', [False, True])])
```

```
27  """
28
29  ----------OUTPUT TEXT--------------
30  {
31      0: 'disability == "Y"',
32      1: 'imd_band in ["0-30%"]',
33      2: 'highest_education in ["Lower Than A Level", "No Formal quals"]
        ',
34      3: 'num_of_prev_attempts > 1',
35      4: 'region in ["North Region", "Wales"]'
36  }
```

Code Listing 8: Interpreter: OULAD Dataset

## C.4 Experimental details

We note that all experiments were performed using two compute resources: a server with NVIDIA RTX A4000 GPU and 18-Core Intel Core i9-10980XE, as well as an Apple M1 Pro 32GB RAM. We exemplify SMART Testing using GPT-4 [29] as the LLM but run further experiments to test the sensitivity to the type of language model in Appendix D.3.

### C.4.1 Context-aware testing (Sec. 5.1.)

**Goal.** We aim to underscore the role of context in ML model testing to prevent false positives, especially when dealing with tabular data where data may contain many irrelevant or uninformative features [30], persisting even post-feature selection [31, 32]. We contrast SMART which explicitly accounts for context, in contrast to data-only approaches which are context-unaware only operating on the data.

**Setup.** We fit a predictive model to the training dataset, varying the number of irrelevant, synthetically generated features contained in the dataset — where irrelevant features are drawn from different distributions. We then quantify the proportion of conditions in the identified slices that falsely include the irrelevant synthetic synthetically features.

Because different methods are sensitive to different types of irrelevant features, we developed a data generating processes that encompasses many types of variables. Over many runs, different data-only methods pick up on some of these variables, showcasing that all methods are susceptible to randomly sampled irrelevant features in the dataset.

**Sampling mechanism.** To evaluate the impact of irrelevant features, we enrich the dataset by adding synthetic categorical variables. The number of new variables is equal to the number of existing features in the dataset. For each new variable $x_i$, we determine its type by sampling from a Bernoulli distribution with probability 0.5. If the sampled value is 0, $x_i$ is a Bernoulli variable with success probability 0.1; otherwise, $x_i$ is a categorical variable with four categories, following a predefined probability distribution (e.g., {0.1, 0.3, 0.4, 0.2}):

$$\text{Type}(x_i) \sim \text{Bernoulli}(0.5),$$
$$x_i \sim \begin{cases} \text{Bernoulli}(0.1) & \text{if Type}(x_i) = 0, \\ \text{Categorical}(0.1, 0.3, 0.4, 0.2) & \text{if Type}(x_i) = 1. \end{cases}$$

We note that there are synthetic data generating processes that completely break other data-only methods. As an example, creating a unique ID column for each sample breaks the Autostrat algorithm, as all the subgroups/slices identified are the unique IDs. The data generating process employed in our experiment reflects a broad variety of commonly encountered DGPs.

### C.4.2 Requirements-constrained testing (Sec. 5.2.)

We test whether each of the methods can fulfil three requirements.

The first requirement involved the use of the variable "age" in each detected slice. This was passed as an input to SMART. The other methods do not accept context as input and, therefore, it was not

possible to fulfil these requirements. The numbers provided for other methods are simply how often they fulfilled the requirements by chance.

The second and third requirements involved obtaining a minimum and maximum sample size. This was passed as an input to SMART and the ablated SMART version. Based on this, SMART changed its hyperparameter within its function which asks to indicate a minimum and maximum sample size for the discovered slices. As with the previous experiments, this was not adjusted for the other groups because they do not take textual input.

### C.4.3 Targeting model failures (Sec. 5.3.)

In order to evaluate the targeting of model failures, we try four different tabular models with pre-specified hyperparameters. We find discrepant slices on the training dataset and evaluate them on the testing dataset.

### C.4.4 Adaptive testing for a deployment environment (Sec. 5.4.)

The goal of the adaptive testing experiment is to understand the extent to which SMART, as well as other data-only methods, can use data in a source domain to generalize to a new, target domain where a covariate shift has been detected. To this end, the datasets provided have a known covariate shift and can be evaluated.

Each method was trained on the UK dataset and the discovered slices were evaluated on the US dataset. No additional context was provided to data-only methods since they do not accept any text or context as inputs.

In contrast, we have provided SMART with the previously discovered slices and hypotheses, and have asked to re-evaluate these hypotheses in the context of the US dataset. Specifically, SMART re-hypothesized possible model failures for the US market but used the UK data to operationalize the variables. The ablated version, $SMART_{NSF}$, achieved the best overall performance on the US market. The ablated version (i) did not have access to the UK data and the failures of the models; and (ii) operationalized each covariate using the LLM alone (refer to Sec. B.2 for a discussion on operationalizations with different SMART versions). This provides evidence that, in the presence of covariate shift, using inductive knowledge or domain expertise might be more useful to finding meaningful model failures.

### C.4.5 Discovery of societally important groups and failure understanding (Sec. 5.5.)

As discussed in the main paper, SMART provides both possible hypotheses and justifications for model failures which can be evaluated using a simple "fit" method. Furthermore, SMART prioritizes meaningful data slices which are of societal importance. Such slices can be inspected for any data input.

# D Additional experiments

## D.1 Requirements-constrained testing

**Goal.** Requirements are a crucial, yet neglected part of ML model testing, such as verifying performance on societally relevant dimensions or verifying specific aspects to meet compliance requirements. However, *no previous testing framework* has incorporated the notion of satisfying requirements when defining the test slices. This experiment illustrates how SMART integrates requirements provided in natural language, which then influences the hypotheses generated to satisfy testing requirements.

**Setup**. We cover three real-world requirements that end-users might have: one based on demographics and two based on sample size. *Requirement 1:* Each of the top 10 identified unique slices should involve the age of a person. *Requirement 2:* The sample size of the top 10 identified unique slices must have at least 150 observations. *Requirement 3:* The sample size of the top 10 identified unique slices has to be small, within 10 and 150 observations. We exemplify the experiment using a real-world prostate cancer dataset from the UK [35], as healthcare often mandates certain testing requirements (e.g. Collins et al. [48]).

**Analysis**. Table 10 shows that SMART which directly integrates requirements (via natural language), satisfies the requirements a greater number of times compared to data-only methods, which only satisfies requirements by chance (hence the low number of times). Beyond satisfying requirements, the SMART slices also represent model failures that almost always have statistically significant performance differences from average when evaluated on test data. Finally, while $SMART_{NSF}$ (ablation without self-falsification) can satisfy requirements, the number of statistically significant slices is lower than SMART, thus underscoring the value of our self-falsification mechanism. That said, $SMART_{NSF}$ still outperforms data-only baselines.

Table 10: Requirement satisfaction showing how many times the top 10 generated slices satisfied the requirements (Req) and how many of these slices had statistically significantly different performance from average (Sig ) on a testing dataset. Maximum is 10. $\uparrow$ is better.

| | $R_1$: Age | | $R_2$: Min sample size | | $R_3$: Max sample size | |
| --- | --- | --- | --- | --- | --- | --- |
| | Req | Sig | Req | Sig | Req | Sig |
| Autostrat | $0.00 \pm 0.00$ | $0.00 \pm 0.00$ | $1.00 \pm 0.00$ | $1.00 \pm 0.00$ | $0.00 \pm 0.00$ | $0.00 \pm 0.00$ |
| PSG_B | $2.00 \pm 0.00$ | $2.00 \pm 0.00$ | $2.00 \pm 0.00$ | $2.00 \pm 0.00$ | $0.00 \pm 0.00$ | $0.00 \pm 0.00$ |
| PSG_A | $2.00 \pm 0.00$ | $2.00 \pm 0.00$ | $2.00 \pm 0.00$ | $2.00 \pm 0.00$ | $0.00 \pm 0.00$ | $0.00 \pm 0.00$ |
| Divexplorer | $3.20 \pm 2.11$ | $0.95 \pm 1.56$ | $0.00 \pm 0.00$ | $0.00 \pm 0.00$ | $2.70 \pm 2.57$ | $1.05 \pm 1.75$ |
| Slicefinder | $4.50 \pm 1.72$ | $1.75 \pm 0.99$ | $4.55 \pm 1.28$ | $3.05 \pm 0.80$ | $3.20 \pm 1.33$ | $1.10 \pm 0.94$ |
| Sliceline | $1.20 \pm 0.40$ | $1.20 \pm 0.40$ | $1.35 \pm 0.48$ | $1.35 \pm 0.48$ | $0.15 \pm 0.48$ | $0.15 \pm 0.48$ |
| **SMART**_NSF | $\mathbf{9.90 \pm 0.30}$ | $5.65 \pm 0.65$ | $6.00 \pm 0.00$ | $6.00 \pm 0.00$ | $4.15 \pm 0.65$ | $2.60 \pm 0.86$ |
| **SMART** | $9.70 \pm 0.56$ | $\mathbf{9.50 \pm 0.59}$ | $\mathbf{9.85 \pm 0.36}$ | $\mathbf{9.80 \pm 0.51}$ | $\mathbf{8.15 \pm 1.11}$ | $\mathbf{6.10 \pm 1.70}$ |

**Takeaway 2.** SMART, unlike data-only methods, identifies slices that have significant performance differences, whilst also satisfying requirements — an important dimension not even considered by previous testing methods.

## D.2 Adaptive testing for a deployment environment

**Goal**. Deploying an ML model often entails going beyond IID, such as a different deployment environment. We consider the case of deploying a model to a different country where there is a covariate shift [12] and evaluate the capabilities of testing frameworks to adapt across the different environment and identify model failures.

**Setup**. We use real-world prostate cancer datasets from different country's cancer registries with known distribution shifts: SEER (US) [11] and CUTRACT (UK) [35]. We train predictor $f$ on UK data, while our target deployment environment is the US.

**Analysis.** ▶ *Identifying model failures.* Table 11 shows that SMART better tests models *at deployment time* using the information provided. SMART identifies a much greater number of statistically significant model failures (*almost all possible*), both within the same environment (UK) and when shifting to a different one (US), even after adjusting for multiple comparisons using Bonferroni correction.

▶ *Sample size sensitivity.* We also assess *sensitivity to sample size*, see Fig. 8. Both variants of SMART are shown to consistently outperform data-only counterparts in identifying a much greater number of significant model failures. *Within domain (UK)*: as expected, we find that for lower sample sizes, $SMART_{NSF}$ (without the self-falsification mechanism) is superior, however, given enough data we then find that SMART benefits from the self-falsification.

Table 11: Number of slices identified (out of a maximum of 10) that had significantly divergent performance from average (higher is better). $S\_\alpha$ counts the number of significantly divergent groups at $\alpha = 0.05$; $S\_\alpha/n$ applies the Bonferroni correction. ↑ is better.

| | $\mathcal{D}^{UK}_{train}$ | | $\mathcal{D}^{UK}_{test}$ | | $\mathcal{D}^{US}_{test}$ | |
|---|---|---|---|---|---|---|
| | $S_\alpha$ | $S_{\alpha/n}$ | $S_\alpha$ | $S_{\alpha/n}$ | $S_\alpha$ | $S_{\alpha/n}$ |
| Autostrat | $1.00 \pm 0.00$ | $1.00 \pm 0.00$ | $1.00 \pm 0.00$ | $1.00 \pm 0.00$ | $0.95 \pm 0.22$ | $0.95 \pm 0.22$ |
| $PSG_B$ | $2.00 \pm 0.00$ | $2.00 \pm 0.00$ | $2.00 \pm 0.00$ | $2.00 \pm 0.00$ | $1.75 \pm 0.43$ | $1.50 \pm 0.50$ |
| $PSG_A$ | $2.00 \pm 0.00$ | $2.00 \pm 0.00$ | $2.00 \pm 0.00$ | $2.00 \pm 0.00$ | $1.75 \pm 0.43$ | $1.45 \pm 0.50$ |
| Divexplorer | $1.65 \pm 1.88$ | $0.45 \pm 0.92$ | $2.00 \pm 2.28$ | $0.85 \pm 1.53$ | $3.90 \pm 2.57$ | $2.75 \pm 2.21$ |
| Slicefinder | $3.65 \pm 0.96$ | $2.75 \pm 0.62$ | $3.85 \pm 1.11$ | $2.70 \pm 0.46$ | $6.80 \pm 1.03$ | $5.95 \pm 1.02$ |
| Sliceline | $1.00 \pm 0.00$ | $1.00 \pm 0.00$ | $1.35 \pm 0.48$ | $1.35 \pm 0.48$ | $1.35 \pm 0.48$ | $1.35 \pm 0.48$ |
| **SMART_NSF** | $8.30 \pm 0.46$ | $8.00 \pm 0.00$ | $8.45 \pm 0.50$ | $8.00 \pm 0.00$ | $9.20 \pm 0.60$ | $8.35 \pm 0.48$ |
| **SMART** | $9.60 \pm 0.49$ | $9.25 \pm 0.54$ | $9.45 \pm 0.50$ | $8.85 \pm 0.57$ | $8.85 \pm 0.79$ | $8.35 \pm 0.79$ |

*Deployment environment (US)*: we find that self-falsification similarly requires sufficient samples; which we note is expected behavior. Interestingly, in the deployment setting (US), $SMART_{NSF}$ generally identifies the greatest number of significant failures (*almost all possible*) across different sample sizes. This suggests that under covariate shift, using inductive knowledge (via the LLM) or domain expertise might be more useful to find meaningful model failures.

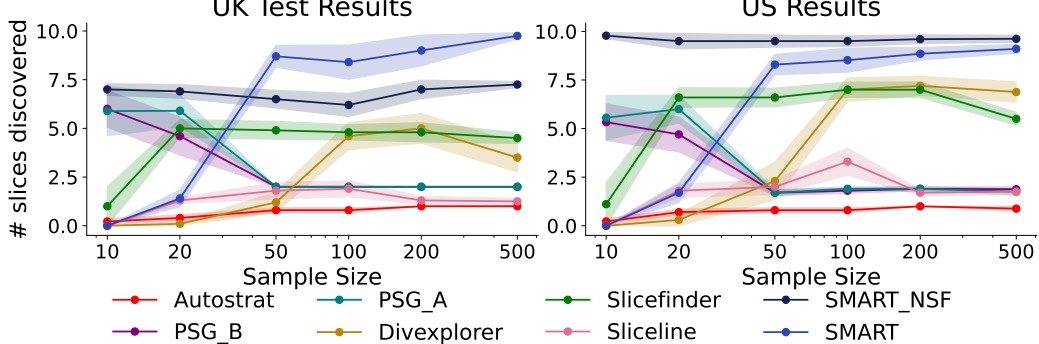

Figure 8: Number of significant groups discovered (out of a total of 10) based on the training dataset size. SMART can operate under any sample size; self-falsification mechanism requires a larger sample size to falsify hypotheses. ↑ is better.

Overall, the results highlight the flexibility of SMART to handle different scenarios and sample sizes. From a practical perspective, while both SMART variants outperform data-only methods, the implication is that there is nuance in using different SMART variants for different scenarios.

**Takeaway 4.** SMART identifies more significant divergent failure slices in a deployment setting, outperforming data-only methods across environments and sample sizes.

---

[12]$p(X)$ changes, while $p(Y|X)$ remains the same

## D.3 Effects of LLMs

In this section, we provide more experimental details which compare the effectiveness of two GPT models, GPT3.5 and GPT4. **We highlight that the goal is not to exhaustively test the framework with every LLM**. Rather, the goal is to showcase that SMART is feasible with at least the capabilities of GPT-4. We provide this section as a way to measure the sensitivity of the model's performance with lower LLMs but highlight that we *do not* recommend using it with smaller LLMs, especially LLMs with fewer than 7B parameters.

### D.3.1 Comparison over identified divergent slices

The following table reproduces the experiment from Sec. D.2 by directly comparing two models - GPT3.5 and GPT4. The setup is the same as in the original experiment.

Table 12: Number of slices identified (out of a maximum of 10) that had significantly divergent performance from average (higher is better). $S_\alpha$ counts the number of significantly divergent groups at $\alpha = 0.05$; $S_{\alpha/n}$ applies the Bonferroni correction.

| | $\mathcal{D}_{\text{train}}^{\text{UK}}$ | | $\mathcal{D}_{\text{test}}^{\text{UK}}$ | | $\mathcal{D}_{\text{test}}^{\text{US}}$ | |
| | $S_\alpha$ | $S_{\alpha/n}$ | $S_\alpha$ | $S_{\alpha/n}$ | $S_\alpha$ | $S_{\alpha/n}$ |
|---|---|---|---|---|---|---|
| Autostrat | $1.00 \pm 0.00$ | $1.00 \pm 0.00$ | $1.00 \pm 0.00$ | $1.00 \pm 0.00$ | $0.95 \pm 0.22$ | $0.95 \pm 0.22$ |
| $PSG_B$ | $2.00 \pm 0.00$ | $2.00 \pm 0.00$ | $2.00 \pm 0.00$ | $2.00 \pm 0.00$ | $1.75 \pm 0.43$ | $1.50 \pm 0.50$ |
| $PSG_A$ | $2.00 \pm 0.00$ | $2.00 \pm 0.00$ | $2.00 \pm 0.00$ | $2.00 \pm 0.00$ | $1.75 \pm 0.43$ | $1.45 \pm 0.50$ |
| Divexplorer | $1.65 \pm 1.88$ | $0.45 \pm 0.92$ | $2.00 \pm 2.28$ | $0.85 \pm 1.53$ | $3.90 \pm 2.57$ | $2.75 \pm 2.21$ |
| Slicefinder | $3.65 \pm 0.96$ | $2.75 \pm 0.62$ | $3.85 \pm 1.11$ | $2.70 \pm 0.46$ | $6.80 \pm 1.03$ | $5.95 \pm 1.02$ |
| Sliceline | $1.00 \pm 0.00$ | $1.00 \pm 0.00$ | $1.35 \pm 0.48$ | $1.35 \pm 0.48$ | $1.35 \pm 0.48$ | $1.35 \pm 0.48$ |
| $SMART\_NSF\_GPT4$ | $8.30 \pm 0.46$ | $8.00 \pm 0.00$ | $8.45 \pm 0.50$ | $8.00 \pm 0.00$ | $9.20 \pm 0.60$ | $8.35 \pm 0.48$ |
| $SMART\_GPT4$ | $9.60 \pm 0.49$ | $9.25 \pm 0.54$ | $9.45 \pm 0.50$ | $8.85 \pm 0.57$ | $8.85 \pm 0.79$ | $8.35 \pm 0.79$ |
| $SMART\_NSF\_GPT3.5$ | $8.20 \pm 0.60$ | $7.15 \pm 0.65$ | $8.20 \pm 0.75$ | $6.95 \pm 0.67$ | $9.25 \pm 0.62$ | $8.25 \pm 0.43$ |
| $SMART\_GPT3.5$ | $10.00 \pm 0.00$ | $9.85 \pm 0.36$ | $10.00 \pm 0.00$ | $9.75 \pm 0.43$ | $7.85 \pm 0.48$ | $7.15 \pm 0.57$ |

The table provides a measure of the model's performance on the training dataset from the same environment ($\mathcal{D}_{\text{train}}^{\text{UK}}$), the testing dataset from the same environment ($\mathcal{D}_{\text{test}}^{\text{UK}}$), and a different deployment environment ($\mathcal{D}_{\text{test}}^{\text{US}}$).

**Takeaway**. Both GPT3.5 and GPT4 provide strong increases over benchmark methods with little variability between the two LLMs. One of the possible reasons why is that the hypothesis space of possible model failures is somewhat limited. This can be seen by the similar hypotheses that are generated by both GPT models.

### D.3.2 Performance across different models

In this section, we vary different tabular machine learning model types and identify how well the ablated and original SMART, identified with GPT3.5 and GPT4, can identify slices with large performance discrepancies.

Table 13: The differences in accuracies between the top slice identified for each method on a testing dataset. The p-value computes the p-value associated with the difference in the accuracy. For the accuracy, higher values imply a greater ability to detect divergent slices (hence, higher is better). For the p-value, lower is better. Averages +- standard deviations are shown across 5 runs with random seeds and data splits

| | Logistic Regression | | SVM | | XGBoost | | Multi-layer Perceptron | |
| | $|\Delta Acc|$ | p-value | $|\Delta Acc|$ | p-value | $|\Delta Acc|$ | p-value | $|\Delta Acc|$ | p-value |
|---|---|---|---|---|---|---|---|---|
| SMART_NSF_GPT3.5 | $0.23 \pm 0.03$ | $0.00 \pm 0.00$ | $0.23 \pm 0.03$ | $0.00 \pm 0.00$ | $0.12 \pm 0.06$ | $0.14 \pm 0.27$ | $0.23 \pm 0.03$ | $0.00 \pm 0.00$ |
| SMART_GPT3.5 | $0.34 \pm 0.07$ | $0.00 \pm 0.00$ | $0.34 \pm 0.07$ | $0.00 \pm 0.00$ | $0.28 \pm 0.04$ | $0.00 \pm 0.00$ | $0.34 \pm 0.07$ | $0.00 \pm 0.00$ |
| SMART_NSF_GPT4 | $0.10 \pm 0.01$ | $0.00 \pm 0.00$ | $0.10 \pm 0.01$ | $0.00 \pm 0.00$ | $0.05 \pm 0.04$ | $0.25 \pm 0.42$ | $0.10 \pm 0.01$ | $0.00 \pm 0.00$ |
| **SMART_GPT4** | $\mathbf{0.40 \pm 0.02}$ | $\mathbf{0.00 \pm 0.00}$ | $\mathbf{0.40 \pm 0.02}$ | $\mathbf{0.00 \pm 0.00}$ | $\mathbf{0.29 \pm 0.06}$ | $\mathbf{0.00 \pm 0.00}$ | $\mathbf{0.40 \pm 0.02}$ | $\mathbf{0.00 \pm 0.00}$ |

**SMART with deep learning models.** SMART's targeted sampling of hypotheses, is entirely independent of the downstream model used. i.e. SMART's context-guided slice sampling mechanism is used to generate hypotheses independently of the downstream model.

We extend our analysis with Logistic Regression, SVM, XGBoost, and MLP to further include two tabular deep learning method: TabPFN and TabNet. As shown in Table 14, across all models SMART is the best at finding subgroups where the models are least reliable.

Table 14: Identifying slices with the highest performance discrepancies. We show differences in accuracies ($|\Delta Acc|$) between the top identified divergent slice and average performance across two state-of-the-art deep learning classifiers (over 5 runs) on the SEER dataset. $\uparrow$ is better. 0.00 implies the evaluation method does not support the model.

| Classifier | Evaluation Method | | | | | | |
|---|---|---|---|---|---|---|---|
| | Autostrat | PSG_B | PSG_A | Divexplorer | Slicefinder | Sliceline | SMART |
| TabPFNClassifier | $0.20 \pm 0.10$ | $0.19 \pm 0.05$ | $0.18 \pm 0.05$ | $0.00 \pm 0.00$ | $0.00 \pm 0.00$ | $0.23 \pm 0.05$ | $\mathbf{0.28 \pm 0.17}$ |
| TabNet | $0.10 \pm 0.09$ | $0.10 \pm 0.04$ | $0.10 \pm 0.04$ | $0.02 \pm 0.04$ | $0.00 \pm 0.00$ | $0.17 \pm 0.08$ | $\mathbf{0.17 \pm 0.12}$ |

**Takeaway.** GPT4 adds additional value when it comes to identifying slices with performance discrepancies. However, GPT3.5 is still able to find significant performance discrepancies across divergent slices, competitive and in many cases exceeding the performance of other methods (refer to Sec. D.4).

### D.3.3 Hypothesis generation

A key component which determines the quality of the SMART method is the ability to navigate the search space by generating hypotheses. Therefore, we overview the kinds of hypotheses which are generated and how they differ based on the type of LLM used.

The table below provides an example of the top 10 hypotheses and justifications generated by GPT3.5 and GPT4 for the prostate cancer example in the UK, where the goal is to find specific slices where the model is likely to fail, and propose justifications for them.

**Key takeaway**. Many of the hypotheses for both models are similar. The underlying difference is in the ordering and justification of the hypotheses.

Table 15: Comparison of Hypotheses and Justifications generated by GPT4 and GPT3.5 for a breast cancer prediction task. The hypothesis target likely model failures.

| | GPT4 | | GPT3.5 | |
|---|---|---|---|---|
| | Hypothesis | Justification | Hypothesis | Justification |
| 1 | Patients with lower age | The model might perform worse for younger patients as prostate cancer is more common in older men, and there might be insufficient data for younger patients. | Subgroup of patients with older age (above 80) will have worse predictive performance due to higher likelihood of comorbidities and weaker immune systems, leading to higher mortality rates. | Older age is often associated with increased risk of comorbidities and weaker immune systems, which can negatively impact the prognosis of prostate cancer. |
| 2 | Patients with higher comorbidities | The model might perform worse for patients with multiple comorbidities as the complexity of their health condition might not be fully captured by the model. | Higher Gleason scores indicate more aggressive and advanced prostate cancer, which is generally associated with poorer prognosis and higher mortality rates. | Subgroup of patients with higher Gleason scores (4 or 5) will have worse predictive performance due to more aggressive and advanced cancer, leading to higher mortality rates. |
| 3 | Patients who underwent conservative management | The model might perform worse for these patients as the effectiveness of conservative management might vary greatly among individuals, and this variability might not be well captured by the model. | Subgroup of patients with stage 4 cancer will have worse predictive performance due to the advanced stage of the disease, indicating metastasis and limited treatment options. | Stage 4 prostate cancer is characterized by the spread of cancer to distant organs, indicating a more advanced and aggressive disease. The limited treatment options available for stage 4 cancer can result in worse predictive performance. |
| 4 | Patients with lower prostate-specific antigen levels | The model might perform worse for these patients as lower PSA levels might be associated with early-stage prostate cancer, which might be harder to predict due to lack of symptoms. | Patients with multiple comorbidities often have complex health conditions that can impact the effectiveness of prostate cancer treatment and management, leading to worse predictive performance. | Subgroup of patients with multiple comorbidities (e.g., 3 or 4) will have worse predictive performance due to the presence of other health conditions that may complicate the treatment and management of prostate cancer. |
| 5 | Patients with higher Gleason scores | The model might perform worse for these patients as higher Gleason scores indicate more aggressive cancer, which might progress unpredictably. | Subgroup of patients who received conservative management as their primary treatment option will have worse predictive performance due to the potential delay in receiving more aggressive treatments. | Conservative management may involve watchful waiting or active surveillance, which can result in delayed initiation of more aggressive treatments. This delay in treatment may lead to worse predictive performance. |
| 6 | Patients in stage 4 | The model might perform worse for these patients as stage 4 prostate cancer is advanced and might progress unpredictably. | Primary hormone therapy is often used to control the growth of prostate cancer by reducing the levels of male hormones (androgens). Patients who do not receive this therapy may have uncontrolled cancer growth, resulting in worse predictive performance. | Subgroup of patients who did not receive primary hormone therapy will have worse predictive performance due to the potential lack of hormonal control over the cancer growth. |
| 7 | Patients who underwent radical prostatectomy | The model might perform worse for these patients as the outcomes of radical prostatectomy might vary greatly among individuals, and this variability might not be well captured by the model. | Subgroup of patients who did not undergo radical radiotherapy will have worse predictive performance due to the potential lack of localized cancer control. | Radical radiotherapy is a treatment option that aims to destroy cancer cells in the prostate gland. Patients who do not undergo this treatment may have a higher likelihood of localized cancer progression, leading to worse predictive performance. |
| 8 | Patients with lower Gleason scores | The model might perform worse for these patients as lower Gleason scores indicate less aggressive cancer, which might progress slower and be harder to predict. | Radical prostatectomy involves the surgical removal of the prostate gland. Patients who do not undergo this procedure may have residual cancer cells in the prostate, which can contribute to worse predictive performance. | Subgroup of patients who did not undergo radical prostatectomy will have worse predictive performance due to the potential presence of residual cancer cells in the prostate gland. |
| 9 | Patients in stage 1 | The model might perform worse for these patients as stage 1 prostate cancer is early-stage and might be harder to predict due to lack of symptoms. | Subgroup of patients with higher prostate-specific antigen (PSA) levels (above 50) will have worse predictive performance due to the indication of more advanced cancer. | Higher PSA levels are often associated with more advanced prostate cancer. Patients with higher PSA levels may have a worse prognosis and higher mortality rates, leading to worse predictive performance. |
| 10 | Patients who underwent primary hormone therapy | The model might perform worse for these patients as the outcomes of primary hormone therapy might vary greatly among individuals, and this variability might not be well captured by the model. | The combination of older age, higher Gleason scores, and stage 4 cancer represents a subgroup with multiple negative prognostic factors. The cumulative effect of these factors is likely to result in the worst predictive performance. | Subgroup of patients with a combination of older age (above 70), higher Gleason scores (4 or 5), and stage 4 cancer will have the worst predictive performance due to the cumulative effect of advanced age, aggressive cancer, and metastasis. |

## D.4  Effects of different tabular machine learning models

The primary task of the SMART method is to evaluate a given, trained machine learning model. Thus far, we have been using a logistic regression model as the basis for evaluation in the main experiments. However, the results are *not* sensitive to the type of the tabular model. Therefore, in this section, we provide additional experiments where we vary the tabular model for the task. Specifically, we consider the following models initialized with their default hyperparameters for evaluation: Logistic Regression, Support Vector Machines, Boosting (implemented with XGBoost) and a multi-layer perceptron with 2 hidden layers and RELU activation functions in the hidden layers.

**Goal**. The primary goal is to understand whether the framework generalizes to other models which operate under different mapping mechanisms (e.g. a logistic regression, which is a linear model, compared to a tree-based model).

**Setup**. Given that not all the slice discovery or model evaluation algorithms output multiple slices, we constrain the evaluation to only focus on a single slice which might have discrepant performance. We use the UK prostate cancer dataset and evaluate the discrepancy of the top identified slice relative to the average across all identified model types. The discrepancy is calculated as the absolute differences of the average performance between the two groups, as well as the p-value associated with the difference. The results show the average performance +- standard deviation of 5 random splits. A higher absolute difference indicates that the model fails on one of the slices more than average.

**Discretizing the inputs**. Many of the discovery methods, however, operate only on categorical data. The previously used dataset, however, has three continuous variables: age, prostate-specific antigen, and comorbidities. We therefore also assess the quality of these methods to discover slices on the testing dataset when these three variables are discretized into 10 bins each. The following

Table 16: The differences in accuracies between the top slice identified for each method on a testing dataset. The p-value computes the p-value associated with the difference in the accuracy. For the accuracy, higher values imply a greater ability to detect divergent slices (hence, higher is better). For the p-value, lower is better. Averages +- standard deviations are shown across 5 runs with random seeds and data splits.

| | Logistic Regression | | SVM | | XGBoost | | Multi-layer Perceptron | |
|---|---|---|---|---|---|---|---|---|
| | $|\Delta Acc|$ | p-value | $|\Delta Acc|$ | p-value | $|\Delta Acc|$ | p-value | $|\Delta Acc|$ | p-value |
| Autostrat | $0.24 \pm 0.02$ | $0.00 \pm 0.00$ | $0.24 \pm 0.02$ | $0.00 \pm 0.00$ | $0.09 \pm 0.09$ | $0.33 \pm 0.46$ | $0.24 \pm 0.02$ | $0.00 \pm 0.00$ |
| *pysubgroup_beam* | $0.23 \pm 0.01$ | $0.00 \pm 0.00$ | $0.23 \pm 0.01$ | $0.00 \pm 0.00$ | $0.11 \pm 0.07$ | $0.18 \pm 0.40$ | $0.23 \pm 0.01$ | $0.00 \pm 0.00$ |
| *pysubgroup_apriori* | $0.23 \pm 0.01$ | $0.00 \pm 0.00$ | $0.23 \pm 0.01$ | $0.00 \pm 0.00$ | $0.11 \pm 0.07$ | $0.19 \pm 0.42$ | $0.23 \pm 0.01$ | $0.00 \pm 0.00$ |
| Divexplorer | $0.05 \pm 0.11$ | $0.81 \pm 0.43$ | $0.09 \pm 0.13$ | $0.61 \pm 0.53$ | $0.14 \pm 0.15$ | $0.47 \pm 0.49$ | $0.02 \pm 0.05$ | $0.86 \pm 0.32$ |
| Slicefinder | $0.01 \pm 0.00$ | $0.00 \pm 0.00$ | $0.01 \pm 0.00$ | $0.00 \pm 0.00$ | $0.00 \pm 0.00$ | $0.00 \pm 0.00$ | $0.01 \pm 0.01$ | $0.00 \pm 0.00$ |
| Sliceline | $0.26 \pm 0.06$ | $0.00 \pm 0.00$ | $0.26 \pm 0.06$ | $0.00 \pm 0.00$ | $0.18 \pm 0.09$ | $0.00 \pm 0.01$ | $0.26 \pm 0.06$ | $0.00 \pm 0.00$ |
| *SMART_NSF* | $0.17 \pm 0.01$ | $0.00 \pm 0.00$ | $0.17 \pm 0.01$ | $0.00 \pm 0.00$ | $0.09 \pm 0.05$ | $0.07 \pm 0.16$ | $0.17 \pm 0.01$ | $0.00 \pm 0.00$ |
| SMART | $0.37 \pm 0.03$ | $0.00 \pm 0.00$ | $0.37 \pm 0.03$ | $0.00 \pm 0.00$ | $0.26 \pm 0.06$ | $0.00 \pm 0.00$ | $0.37 \pm 0.03$ | $0.00 \pm 0.00$ |

table reports the performance of all the methods (note: we did not discretize the dataset for SMART because it can work natively on continuous data).

Table 17: The differences in accuracies between the top slice identified for each method on a testing dataset when the datasets continuous features are discretized.

| | Logistic Regression | | SVM | | XGBoost | | Multi-layer Perceptron | |
|---|---|---|---|---|---|---|---|---|
| | $|\Delta Acc|$ | p-value | $|\Delta Acc|$ | p-value | $|\Delta Acc|$ | p-value | $|\Delta Acc|$ | p-value |
| Autostrat | $0.02 \pm 0.02$ | $0.38 \pm 0.27$ | $0.02 \pm 0.02$ | $0.45 \pm 0.37$ | $0.02 \pm 0.02$ | $0.52 \pm 0.32$ | $0.02 \pm 0.02$ | $0.43 \pm 0.33$ |
| *PSG_B* | $0.01 \pm 0.01$ | $0.56 \pm 0.29$ | $0.01 \pm 0.01$ | $0.62 \pm 0.38$ | $0.01 \pm 0.01$ | $0.55 \pm 0.25$ | $0.01 \pm 0.01$ | $0.57 \pm 0.23$ |
| *PSG_A* | $0.01 \pm 0.01$ | $0.55 \pm 0.29$ | $0.01 \pm 0.01$ | $0.61 \pm 0.38$ | $0.01 \pm 0.01$ | $0.55 \pm 0.26$ | $0.01 \pm 0.01$ | $0.59 \pm 0.26$ |
| Divexplorer | $0.10 \pm 0.08$ | $0.47 \pm 0.28$ | $0.11 \pm 0.10$ | $0.41 \pm 0.36$ | $0.13 \pm 0.11$ | $0.38 \pm 0.35$ | $0.13 \pm 0.10$ | $0.38 \pm 0.37$ |
| Slicefinder | $0.00 \pm 0.00$ | $0.00 \pm 0.00$ | $0.00 \pm 0.00$ | $0.00 \pm 0.00$ | $0.00 \pm 0.00$ | $0.00 \pm 0.00$ | $0.00 \pm 0.00$ | $0.00 \pm 0.00$ |
| Sliceline | $0.09 \pm 0.01$ | $0.05 \pm 0.06$ | $0.08 \pm 0.02$ | $0.14 \pm 0.09$ | $0.11 \pm 0.07$ | $0.25 \pm 0.24$ | $0.09 \pm 0.01$ | $0.06 \pm 0.08$ |
| *SMART_NSF* | $0.17 \pm 0.01$ | $0.00 \pm 0.00$ | $0.17 \pm 0.01$ | $0.00 \pm 0.00$ | $0.09 \pm 0.05$ | $0.07 \pm 0.16$ | $0.17 \pm 0.01$ | $0.00 \pm 0.00$ |
| SMART | $0.37 \pm 0.03$ | $0.00 \pm 0.00$ | $0.37 \pm 0.03$ | $0.00 \pm 0.00$ | $0.26 \pm 0.06$ | $0.00 \pm 0.00$ | $0.37 \pm 0.03$ | $0.00 \pm 0.00$ |

**Takeaway.** SMART is able to consistently identify the greatest performing slices across a number of different tabular models.

### D.5 On the inductive biases of ML model testing

**Goal.** We further observe that data-only testing methods implicitly assume the existence of slices with discrepancies in performance. While indeed ML models do fail — it is equally as problematic to highlight failures where there are none.

**Setup.** To evaluate this we propose a fully synthetic setup. Here, both the dependent variable ($Y$) and the independent variables ($\mathbf{X}$) are sampled from a predefined random distribution. Specifically, we predict loan default ($Y \in \{0, 1\}$) based on a set of independent variables $\mathbf{X} = \{N_{\text{runs}}, M_{\text{pref}}, A_{\text{rainfall}}, F_{\text{color}}, P_{\text{season}}\}$, which are conceptually and empirically independent of the outcome of interest. Ideally, if we account for context we should be able to identify these disparate features should not influence the account and hence without prior relationships we should not flag spurious slices.

We compare the data-only methods to SMART under three different data generating processes (scenarios) that capture diverse underlying dynamics denoted as $\mathcal{S}_{\text{uniform}}$, $\mathcal{S}_{\text{skewed}}$, and $\mathcal{S}_{\text{interactions}}$, where each DGP has a focus on uniform, skewed, and interactive effects, respectively.

The first scenario is given by the variables sampled from the following DGPs:

$$N_{\text{runs}} \sim \text{Uniform}\{1, 499\},$$
$$M_{\text{pref}} \sim \text{Uniform}\{1, 5\},$$
$$A_{\text{rainfall}} \sim \text{Uniform}\{20000, 99999\},$$
$$F_{\text{color}} \sim \text{Uniform}\{1, 6\},$$
$$P_{\text{season}} \sim \text{Uniform}\{0, 3\},$$
$$Y \sim \text{Uniform}\{0, 1\}.$$

The second scenario is given by the variables sampled from the following DGPs:

$$N_{\text{runs}} \sim \text{Uniform}\{1, 499\},$$
$$M_{\text{pref}} \sim \text{Binomial}(1, 0.5),$$
$$A_{\text{rainfall}} \sim \text{Categorical}(0.1, 0.3, 0.4, 0.2),$$
$$F_{\text{color}} \sim \text{Binomial}(1, 0.1),$$
$$P_{\text{season}} \sim \text{Binomial}(1, 0.05),$$
$$Y \sim \text{Uniform}\{0, 1\}.$$

The third scenario is given by the variables sampled from the following DGPs:

$$N_{\text{runs}} \sim \text{Uniform}\{1, 499\},$$
$$M_{\text{pref}} \sim \text{Binomial}(1, 0.5),$$
$$A_{\text{rainfall}} \sim \text{Categorical}(0.1, 0.3, 0.4, 0.2),$$
$$A_{\text{music\_hap}} = M_{\text{pref}} \times A_{\text{rainfall}},$$
$$A_{\text{run\_hap}} = N_{\text{runs}} \times M_{\text{pref}},$$
$$Y \sim \text{Uniform}\{0, 1\}.$$

**Analysis.** Table 18 shows the number of slices spuriously discovered, while Table 19 outlines the number of conditions within the slices. We can clearly see the pitfalls of data-only approaches which detect slices which in reality have no relation to one another — often surfacing few conditions per group which suggests they arise by chance. The rationale for this failure is simply because data-only approaches do not and cannot reason about the features and/or understand context and simply aim to find slices with discrepancies in performance — which of course could arise by chance. In contrast, we see that SMART by virtue of context-awareness can avoid surfacing groups — which in reality have no relationships.

Table 18: Number of discovered slices on a synthetic dataset with no prior relationships in three data generating process scenarios. slices capped at most 20. Average of 50 runs $\pm$ standard deviations is shown.

| Method | $\mathcal{S}_{\text{uniform}}$ | $\mathcal{S}_{\text{skewed}}$ | $\mathcal{S}_{\text{interactions}}$ |
|---|---|---|---|
| Autostrat | $1.00 \pm .0$ | $1.00 \pm .0$ | $1.00 \pm .0$ |
| $PSG\_B$ | $20.00 \pm .0$ | $20.00 \pm .0$ | $20.00 \pm .0$ |
| $PSG\_A$ | $20.00 \pm .0$ | $20.00 \pm .0$ | $20.00 \pm .0$ |
| divexplorer | $20.00 \pm .0$ | $20.00 \pm .0$ | $20.00 \pm .0$ |
| slicefinder | $20.00 \pm .0$ | $20.00 \pm .0$ | $20.00 \pm .0$ |
| SMART | $\mathbf{0.00 \pm .0}$ | $\mathbf{0.00 \pm .0}$ | $\mathbf{0.00 \pm .0}$ |

### D.6 Context aware sensitivity

We provide an additional experiment where we vary the sample size in the training dataset and observe how that affects the number of slices discovered for each method. We show that the SMART is not affected by irrelevant features regardless of the sample size of the training dataset. The result is shown in Figure 9.

Table 19: Number of conditions per discovered slice (false positives) in three data generating process scenarios. Average of 50 runs +- standard deviations is shown. Lower is better.

| Method | $\mathcal{S}_{\text{uniform}}$ | $\mathcal{S}_{\text{skewed}}$ | $\mathcal{S}_{\text{interactions}}$ |
|---|---|---|---|
| Autostrat | $2.17 \pm 0.46$ | $1.73 \pm 1.01$ | $1.13 \pm 0.35$ |
| $PSG\_B$ | $1.03 \pm 0.18$ | $1.90 \pm 0.71$ | $1.27 \pm 0.45$ |
| $PSG\_A$ | $1.03 \pm 0.18$ | $1.90 \pm 0.71$ | $1.27 \pm 0.45$ |
| divexplorer | $2.10 \pm 0.31$ | $2.87 \pm 0.68$ | $2.40 \pm 0.50$ |
| slicefinder | $1.40 \pm 0.50$ | $1.50 \pm 0.57$ | $1.00 \pm 0.00$ |
| SMART | $\mathbf{0.00 \pm 0.00}$ | $\mathbf{0.00 \pm 0.00}$ | $\mathbf{0.00 \pm 0.00}$ |

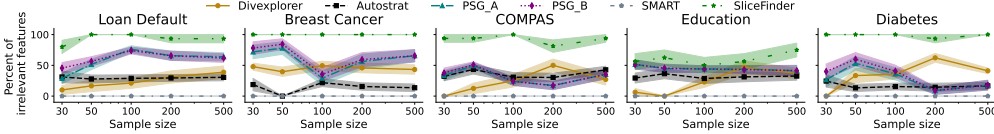

Figure 9: Proportion of irrelevant features (y) for each slice discovery method, based on the sample size. Lower is better.

## D.7    Cost of SMART

We assess the cost of LLM hypothesis generation and scalability to larger datasets. Specifically, we demonstrate not only that SMART is cheap but also easily scalable to large datasets

- Scalability: SMART's scalability depends on the number of hypotheses generated, not dataset size (unlike data-only methods). This allows SMART to easily scale to arbitrarily large datasets.

- Cost Analysis: In practical terms, cost then also scales primarily with the number of hypotheses generated, not dataset size. We provide a rough estimate based on token counts of input and outputs for 2 datasets (SEER and OULAD) in Table 20. This would be less than 0.1 USD for 5 hypotheses and less than 0.5 USD for 100 hypotheses for state-of-the-art models.

Table 20: Cost of SMART (USD) for different GPT LLMs and different numbers of hypotheses generated. The cost is estimated based on token counts. Note: GPT-4o models are post our paper and are even cheaper.

| Model | Cost SEER 5 Hypothesis (USD) | Cost SEER 100 Hypothesis (USD) | Cost OULAD 5 Hypothesis (USD) | Cost SEER 100 Hypothesis (USD) |
|---|---|---|---|---|
| GPT-4 | 0.017 | 0.249 | 0.022 | 0.316 |
| GPT-3.5 | 0.004 | 0.050 | 0.005 | 0.064 |
| GPT-4o Mini | 0.0003 | 0.005 | 0.0005 | 0.006 |
| GPT-4o | 0.008 | 0.125 | 0.011 | 0.158 |

## D.8    SMART with open-weight models

SMART ideally should be used with the most capable LLM possible.That said, we assess the differences in hypotheses between open-weight models and GPT-4.

We assess Mistral-7b, Qwen-1.5-7b, Llama-3-8b, Llama-70b, where for the OULAD and SEER datasets we generate 5 hypotheses and assess overlap to the hypotheses generated by GPT-4. This is presented in Table 21.

To summarize, the overlap between open-source models and GPT-4 is between 60-80%. We find that open-source models propose similar hypotheses, but they are not replacements for more capable models. This highlights that less capable models might propose similar hypotheses, yet they still catch fewer model failures.

## D.9    Understanding the importance of feature names

SMART uses the implicit context encoded in the interpretable feature names as a source of contextual information to guide hypothesis generation. For instance, in a medical dataset, features with names

Table 21: Comparison hypotheses by GPT-4 and overlap w/ open-weight models

| Dataset | Factors (GPT Hypotheses) | Mistral-7b | Llama 3-8b | Qwen 1.5-7b | Llama 70b |
|---|---|---|---|---|---|
| **Oulad Dataset** | Disability | ✓ | ✓ | ✓ | ✓ |
| | IMD band | ✓ | ✓ | | ✓ |
| | Age | | ✓ | ✓ | ✓ |
| | Number of previous attempts | ✓ | ✓ | ✓ | |
| | Test (boolean) | | | | |
| | **Oulad Overlap Percentage** | 60% | 70% | 60% | 60% |
| **SEER Dataset** | Age | | ✓ | ✓ | ✓ |
| | Prostate-specific antigen (PSA) | | | | |
| | Comorbidities | ✓ | ✓ | ✓ | ✓ |
| | Treatment (conservative management) | ✓ | ✓ | ✓ | ✓ |
| | Cancer stage | | | ✓ | ✓ |
| | **SEER Overlap Percentage** | 40% | 60% | 80% | 80% |

like age, sex, or patient covariate features provide context to guide LLM hypothesis generation. This contrasts with data-only approaches which only use the numerical data values alone and ignore the context surrounding the feature names.

We aim to assess the sensitivity to interpretable feature names to provide guidance on the use of SMART. First, we perform a qualitative study where we limit the data schema by hiding the feature names (such that they become uninformative) and inspect the hypotheses and justifications generated. We find that in the limited-schema case, SMART generates hypotheses based on inferences about the feature information (e.g. "the model might fail on feature_4 if feature_4 represents gender"). In contrast, informative names guide meaningful hypothesis generation. Such hypotheses and justifications are illustrated in Table 22.

Second, we evaluate whether limiting the data schema by hiding some feature names and leaving minimal external context affects detection rates of model failures. We compare two versions of SMART, original and with corrupted feature labels, in identifying data slices with high performance discrepancies from average (Fig. 10. We find that across two real-world private datasets, hiding the feature names hinders model evaluation. This highlights that feature names play an important role in finding model failures.

These results highlight while SMART does not rely on any additional feature descriptions, feature names play an important role in finding model failures, just as any human requires interpretable feature names to understand the data . That said, feature names (e.g. column labels such as sex, age, race etc) are present in almost all tabular datasets both in the research field and in the real world where data is stored in SQL tables with column names.

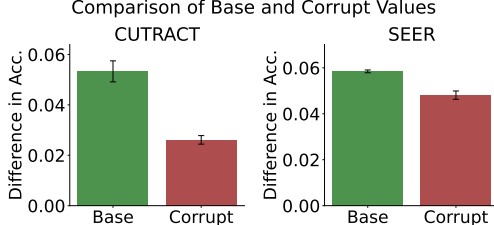

Figure 10: Identifying the importance of feature names as a source of information for context-aware testing. We compare two versions of SMART on CUTRACT and SEER datasets. The first version, "base", is the normal SMART method. The second version, "corrupt", has feature names changed to uninformative ones. The goal is to identify slices with the highest performance discrepancies between the average & subgroup prediction, calculated as the difference in their respective accuracies (y-axis). We see showing feature names enables to find subgroups that have a higher performance discrepancy from baseline performance than without feature names. ↑ is better.

Table 22: Example hypotheses and justifications when dataset column names are hidden. The hypotheses and justifications for features that do not encode meaningful information lose practical meaning. Context-aware testing benefits from rich feature names that can be used to guide meaningful hypothesis generation.

| Model type | Hypothesis | Justification |
|---|---|---|
| Base | The model will perform worse on predicting the risk of disease for older individuals. | Age is a significant factor in many diseases, and older individuals often have more complex health situations with multiple comorbidities. This complexity could make it more difficult for the model to accurately predict disease risk. |
| Base | The model will perform worse on predicting the risk of disease for individuals with lower Gleason scores. | The Gleason score is a grading system used to determine the aggressiveness of prostate cancer. Lower scores indicate less aggressive cancer, which may be more difficult to predict due to its slower progression and less obvious symptoms. |
| Corrupted | {'1': ['feature_0']} | If feature_0 represents a characteristic such as race or ethnicity, the model might perform worse for certain subgroups due to societal biases. For instance, if the dataset is biased towards a particular race or ethnicity, the model's predictions for other races or ethnicities might be less accurate. |
| Corrupted | {'2': ['feature_4']} | If feature_4 represents a characteristic such as gender, the model might perform worse for certain subgroups due to societal biases. For example, if the dataset is biased towards one gender, the model's predictions for the other gender might be less accurate. |

## D.10   Example of a model report

We show an example of a model report which is automatically generated by SMART. A model report includes a summary of key hypotheses, justifications, sources, and summary findings as a written report. A model report fully satisfies the requirements of the user.

In this case, the model is generated to directly match latex output requirements. It is provided in the shaded box below as a part of the US prostate cancer (SEER) [11] evaluation.

A report on the performance of the model has been concluded. The following are the hypotheses, their justifications tested on the model with their conclusions on whether the hypothesis was supported.

| Hypothesis | Justification | Operationalization | Hypothesis Supported |
|---|---|---|---|
| The model may perform worse for older patients | Older patients may have more co-morbidities and complex health situations that are not fully captured by the dataset. Additionally, societal biases may lead to less aggressive treatment options being pursued for older patients, which could affect the model's predictions. | $age > 75$ | Yes |
| The model may perform worse for patients with lower prostate-specific antigen levels | Lower levels of prostate-specific antigen may be associated with earlier stages of prostate cancer, which may be harder to predict due to less data and less obvious symptoms. | $prostate\_specific\_antigen < 10$ | Yes |
| The model may perform worse for patients who have undergone conservative management | Conservative management is a less aggressive form of treatment, which may be chosen due to a variety of factors not captured in the dataset, such as patient preference or other health considerations. This could introduce additional complexity into the model's predictions. | $treatment\_conservative\_management == 1$ | Yes |
| The model may perform worse for patients with a higher number of comorbidities | Patients with more comorbidities may have more complex health situations that are not fully captured by the dataset. Additionally, these patients may be more likely to die from causes other than prostate cancer, which could confuse the model's predictions. | $comorbidities > 2$ | No |
| The model may perform worse for patients with a higher Gleason score | A higher Gleason score indicates more aggressive cancer, which may be harder to predict due to its rapid progression and the potential for other health factors to influence outcomes. | $gleason\_score == 4$ | No |

A further analysis was run on the training and testing datasets to understand the effects of each identified slice. The following are two tables which summarize a part of this information:

| | group_size | support | p_value_bootstrap | num_criteria | outcome_diff | accuracy_diff |
|---|---|---|---|---|---|---|
| H0 | 4915.00 | 0.31 | 0.00 | 1.00 | 0.23 | 0.05 |
| H1 | 7749.00 | 0.48 | 0.00 | 1.00 | 0.26 | 0.01 |
| H2 | 1456.00 | 0.09 | 0.00 | 1.00 | 0.06 | 0.09 |
| H3 | 574.00 | 0.04 | 0.94 | 1.00 | 0.08 | 0.00 |
| H4 | 94.00 | 0.01 | 0.14 | 1.00 | 0.36 | 0.09 |

| | odds_ratio_outcome | odds_ratio_acc | lift_outcome | lift_acc | weighted_relative_y | weighted_relative_acc |
|---|---|---|---|---|---|---|
| H0 | 0.82 | 1.39 | 1.46 | 0.94 | 0.07 | -0.02 |
| H1 | 0.96 | 1.12 | 0.48 | 0.99 | -0.13 | -0.01 |
| H2 | 0.99 | 1.41 | 1.11 | 0.89 | 0.01 | -0.01 |
| H3 | 0.98 | 1.01 | 0.85 | 1.00 | -0.00 | -0.00 |
| H4 | 0.48 | 0.55 | 1.72 | 1.10 | 0.00 | 0.00 |

Recommendations:

1. The model appears to be less reliable for older patients ($age > 75$), patients with lower levels of prostate-specific antigen ($< 10$), and those who have undergone conservative management treatment. It also shows decreased performance for patients with more than two comorbidities and those with a Gleason score of 4. However, the model is more reliable when these conditions are not met.

2. Before deploying the model, the end user should be aware of the following: - The model's performance may be compromised for older patients and those with multiple comorbidities. Consider additional validation or alternative models for these groups. - Patients with lower prostate-specific antigen levels and those who have undergone conservative management treatment may also experience less accurate predictions. Additional clinical insights may be needed for these cases. - Although the model shows decreased performance for patients with a Gleason score of 4, this group is relatively small, so the impact on overall model performance may be limited. However, caution should be exercised when interpreting results for these patients. Remember, these recommendations are based on the training and test datasets from the UK. If deploying in a different geographical context, consider revalidating the model with local data.

—

Definitions of the metrics:

- **Group Size**:
$$\text{group\_size} = |\text{slice}|$$

- **Support**:
$$\text{support} = \frac{|\text{slice}|}{|\text{dataset}|}$$

- **Number of Criteria**:
$$\text{num\_criteria} = \text{Count("and")} + 1$$

- **Outcome Difference**:
$$\text{outcome\_diff} = |\text{avg\_outcome\_dataset} - \text{avg\_outcome\_slice}|$$

- **Accuracy Difference**:
$$\text{accuracy\_diff} = |\text{accuracy\_dataset} - \text{accuracy\_slice}|$$

- **Odds Ratio (Outcome)**:
$$\text{odds\_ratio\_outcome} = \frac{p_1(1 - p_1)}{p_0(1 - p_0)}$$

- **Odds Ratio (Accuracy)**:
$$\text{odds\_ratio\_acc} = \frac{p_1(1 - p_1)}{p_0(1 - p_0)}$$

*(where $p_1$ and $p_0$ are accuracies in the slice and the rest of the dataset, respectively)*

- **Lift (Outcome)**:
$$\text{lift\_outcome} = \frac{p_1}{p}$$

- **Lift (Accuracy)**:
$$\text{lift\_acc} = \frac{p_1}{p}$$

*(where $p_1$ is accuracy in the slice and $p$ is accuracy in the entire dataset)*

- **Weighted Relative Outcome**:
$$\text{weighted\_relative\_outcome} = \text{support} \times \text{diff\_outcomes}$$

- **Weighted Relative Accuracy**:
$$\text{weighted\_relative\_accuracy} = \text{support} \times \text{diff\_accuracy}$$

