# OpenReview forum: "Context-Aware Testing: A New Paradigm for Model Testing with Large Language Models"
_NeurIPS.cc/2024/Conference — NeurIPS 2024 poster_

### Official Review · Reviewer_ctad · 2024-06-25

**Soundness:** 2
**Presentation:** 2
**Contribution:** 1
**Rating:** 3
**Confidence:** 3

**Summary:**

This paper introduces Context-Aware Testing (CAT), a novel approach that uses context as an inductive bias to guide the search for meaningful model failures. Unlike previous methods that rely solely on data to find slices where a model's predictions underperform compared to average performance, CAT addresses the limitations of these data-only approaches. Traditional methods often fail to consider the problem context and can lead to multiple testing problems. To overcome these issues, the authors propose loosening the restrictive assumption of relying solely on data. They introduce CAT as a new testing paradigm that incorporates context to provide a more comprehensive evaluation of model performance. Additionally, the paper presents SMART Testing, which employs large language models (LLMs) to hypothesize relevant and likely failures. These hypotheses are then evaluated on data using a self-falsification mechanism, allowing for the automatic identification of more relevant and impactful failures compared to traditional methods.

**Strengths:**

The topic is highly interesting, and I agree with the authors' motivation that diverse perspective testing is necessary for practical deployment in real-world cases. These dimensions can improve the practical utility of tabular prediction models. In particular, the core limitation of data-only testing—the lack of a principled failure mode discovery procedure that can incorporate prior knowledge to guide the search for meaningful errors—is also persuasive.

**Weaknesses:**

Despite the importance of the topic and motivation, it is difficult to agree that the suggested method fully addresses all the issues. Most importantly, in the first step of hypothesis generation, we need to define the contextual information and dataset contextualization. If my understanding is correct, I am quite confused about how this contextual information differs from rule-based methods defined by human experts to use contextual information to find specific data slices. For example, in Table 1, if we have contextual information about age or comorbidities, we can directly utilize that information to find data slices without using LLMs or the operationalization step. How are these different, and how are steps 1 and 2 beneficial? I feel that utilizing LLMs is quite redundant for practical use. Additionally, too much prior knowledge is needed to utilize SMART testing, raising concerns about its practicality.

**Questions:**

- Can you provide examples for step 1 like in Table 1? What specifically constitutes contextual information and dataset contextualization?
- In Section 5.1, how are the descriptions for synthetic features provided in SMART testing? If the context corresponds to "it is synthetic" or "it is independent from the other," I think the results are too trivial.
- If there are no accurate descriptions related to the features or datasets, how is the method applicable?

**Limitations:**

- As mentioned in the weaknesses, the reason why LLM is needed should be elaborated. As the authors mentioned in the manuscript, using LLMs could even hinder the total procedure of SMART testing if there is no clear reason for their use.
- While the motivation is quite practical, rather than research-friendly, which is valuable, it would be beneficial to provide "practical" examples to illustrate the application of the proposed methods.

---

> ### Author Rebuttal · Authors · 2024-08-06
>
> Dear R-ctad,
>
> Thank you for the feedback which helped to clarify our paper. We'll address misunderstandings about contextual information and LLM necessity.
>
> We provide grouped answers A-D & highlight paper updates.
>
> ---
> # (A) Clarification on contextual information
>
> We apologize for any confusion caused by our use of the term 'context'. To clarify, we use 'context' in two distinct ways: (a) external input by a human (not strictly necessary for the framework) and (b) external knowledge (such as relevant background knowledge, domain understanding) within an external system (LLM). While we made a small footnote about this on page 5, we agree this is not enough and could have resulted in confusion about the nature of our work.
>
> | Aspect | Context as input (external input) | Context as external knowledge |
> -|-|-
> | Definition | Additional problem/dataset information in the form of a textual string | Relevant background knowledge of a system (e.g. an LLM) that informs the generation of targeted hypotheses |
> | Source | User input (optional) | An external system (e.g., GPT-4) which includes an understanding of the relationships between dataset features and their meaning|
> | Example | `info = "this is a dataset on heart disease prediction"` | Use of a language model (e.g., GPT-4) to generate hypotheses or justifications by understanding of the data |
> | Required? | Not necessary (can be left blank) | Required for CAT to work |
>
> "Context-aware testing" refers to external knowledge use, not user input. SMART uses tabular context (which is inherently encoded in the feature names) for hypothesis generation without needing human input. For example, **Table 1** in the paper showcases the hypotheses that were generated by SMART leveraging the feature names without any external human effort.
>
> ## Need for prior knowledge & practical example.
>
> Further, based on your suggestion to include a 'practical' example to illustrate the application of the proposed method, we thought it would be useful to show how easy it is to use SMART as well as the minimal contextual information and dataset contextualization needed.
>
> As shown below, users do not need any prior knowledge to use SMART. Rather, we make use of the context inherently encoded in the dataset, feature names and task.
>
> ```python
> import SMART
> model_tester = SMART('gpt-4')
>
> # Optional external input (can be left blank)
> context = "Find where a model fails. This is a prostate cancer prediction task."
>
> model = XGBoost()
> description_dataset = X_train.describe()
>
> model_tester.fit(X_train, context, description_dataset, model)
>
> print(model_tester.subgroups)
> ```
>
> This implements SMART with just a few lines, requiring no additional human effort beyond providing the dataset and model.  In our implementations, we typically provide generic, easily accessible context that does not vary across tasks (see **Appendix C.3**. for examples).
>
> **UPDATE**: Add table to Sec. 3.3 to clarify the two uses of context and rename the *context input* as *external input* which we hope will disambiguate the two meanings.
>
> ---
> # (B) Difference from Rule-Based Methods
>
> CAT and SMART fundamentally differ from rule-based methods:
>
> - No human input required: Unlike rule-based methods, CAT does not need experts to define rules based on domain knowledge.
> - Automated hypothesis generation: SMART uses feature names as implicit context to guide hypothesis generation automatically.
> - Scalability: This approach can be applied to any tabular dataset without human intervention.
>
> **UPDATE**: Add a contrast to rule-based in Sec. 3.2
>
> ---
> # \(C) Why we need an LLM in SMART
>
> We wish to avoid using humans for large-scale testing because human intervention is expensive, time-consuming and often unavailable. Instead, we desire an *automated* approach to testing ML models. For this, we suggest to employ LLMs. The use of an LLM is crucial because (as per L177-186):
>
> - Contextual understanding: LLMs can interpret feature names and generate relevant hypotheses without human input
> - Targeted sampling: LLMs limit the number of tests, focusing on relevant model failures without testing all possible combinations
>
> **Benefits**. We show that by using LLMs to search for model failures, we are able to test ML models better. SMART finds data slices where models are unreliable much more often than data-only methods (see Sec. 5). Therefore, SMART offers a solution when human experts are unavailable or too expensive.
>
> Ultimately, by using LLMs to *automatically* generate context-aware hypotheses, we can test models for model failures without succumbing to the issues that data-only testing methods face (see Sec. 3.2).
>
> **UPDATE**: Enhance current description around L174-186.
>
> ---
> # (D) Questions
> Q1: What constitutes contextual info?
>
> A1: See response (A) clarifying contextual information.
>
> Q2: How are the synthetic features provided in SMART testing in Sec. 5.1?
>
> A2: We add new random features to the dataset which have uninformative names, such as "feature1", "feature2". We show that data-only methods will test for *everything* which results in systematic errors. SMART avoids this by selectively (and automatically) testing for *important* data slices.
>
> Q3: How is the method applicable if there are no descriptions of features?
>
> A3: See response \(C): (a) feature names provide sufficient context for hypothesis generation; (b) generic prompts are used that do not change across tasks (see **Appendix C.3** for examples). There is no need to provide additional descriptions.
>
> Q4: How are steps (1) and (2) in SMART beneficial if we already have contextual information about age and comorbidities?
>
> A4: They are beneficial because we do not assume prior access to any knowledge (see response (A)). These hypotheses are automatically generated by the system purely from tabular feature names & generic external input about the task.
>
> ---
> # Thank you
>
> Given our clarifications, we hope you consider revising the paper's evaluation ☺️

---

> ### Author Response · Authors · 2024-08-11
>
> Dear reviewer ctad,
>
> We are sincerely grateful for your time and energy in the review process.
>
> We hope that our responses have been helpful. Given the limited time left in the discussion period, please let us know if there are any leftover concerns and if there was anything else we could do to address any further questions or comments :)
>
> Thank you!
>
> Paper authors

---

> > ### Comment · Reviewer_ctad · 2024-08-12
> >
> > Thank you for your detailed rebuttal. After careful consideration, I still believe that my initial assessment is accurate. The primary concern is that the proposed method seems more suited for research purposes rather than practical use, which is crucial for testing methodologies.
> >
> > - Limited Applicability Without Feature Descriptions: The effectiveness of your approach is significantly limited when feature descriptions are unavailable. This restricts its utility in practical scenarios where such descriptions might not be provided.
> >
> > - Cost of Utilizing LLMs: While LLMs are powerful, their use can be costly, both in terms of computation and resources. This raises concerns about the practicality of applying your method in real-world settings.
> >
> > Moreover, from a research perspective, the novelty of the approach seems constrained. The example provided in the rebuttal merely demonstrates using an LLM to identify failure cases by inputting a dataset description. This does not appear to offer substantial insights for practitioners.
> >
> > For instance, in the OpenML benchmark, datasets with the same data but different IDs can have vastly different descriptions. While the model fitting results would be identical, the CAT outcomes could vary significantly based on the description. Additionally, simply removing feature names could lead to different results despite the data being the same. Can we truly trust these outcomes?
> >
> > I was hoping to see a response addressing these specific concerns, but I did not find a satisfactory explanation in the rebuttal.

---

> > > ### Author Response · Authors · 2024-08-12
> > >
> > > Dear Reviewer ctad
> > >
> > > Thank you for your response. We feel there has been a major misunderstanding. Please find answers to your questions below, many of which have _already_ been answered in our earlier response or in the paper itself, which we re-clarify below.
> > >
> > > ---
> > > **Comment 1**: Limited Applicability Without Feature Descriptions: The effectiveness of your approach is significantly limited when feature descriptions are unavailable.
> > >
> > > **Answer 1**. There seems to be a misunderstanding, as this is *not true*. We have answered this question in **Response (D) A3** -- we do not require feature descriptions. Our inputs, apart from a one-sentence string describing the nature of the task, are equivalent to the inputs of data-only testing (despite showing superior performance).
> > >
> > > To put this misclarification to an end: *Our method does not rely on any additional feature descriptions*. We *do* require interpretable feature names, just as any human requires interpretable feature names to understand what they are working with. Feature names (e.g. column labels such as sex, age, race etc) are present in almost all tabular datasets both in the research field and in the real world where data is stored in SQL tables with column names.
> > >
> > > *This means that the utility of this method directly extends to all the users who work with such data*.
> > >
> > > ---
> > >
> > > **Comment 2**: Cost of Utilizing LLMs: While LLMs are powerful, their use can be costly, both in terms of computation and resources.
> > >
> > > **Answer 2**. There seems to be a misunderstanding about the costs associated with LLMs and the computational requirements. It is *not true* that our model is costly of computationally expensive. We refer the reviewer to our **global response (and our response pdf)** where we address the exact point on cost showing SMART is cheap to use and hence very practical. For your convenience, see the snippet below
> > >
> > > *Table 1. Cost of SMART (USD). We quantify the usage of smart in USD for different closed-source GPTs. SMART is extremely accessible and cheap to use. Generating 100 hypotheses with GPT-4o Mini costs less than 1 cent.* (GPT-4 < 0.3).
> > >
> > > Finally, given the stark differences in the effectiveness SMART vs data-only, we think it is vital for the community (both research and industry)  to have access to improved testing methods for more reliable ML testing.
> > >
> > > ---
> > >
> > > **Comment 3**. From a research perspective, the novelty of the approach seems constrained. The example provided in the rebuttal merely demonstrates using an LLM to identify failure cases by inputting a dataset description.
> > >
> > > **Answer 3**. We'd like to address the misunderstanding you got from our rebuttal example. The code snippet we provided demonstrates how easy SMART is to use and that it requires minimal prior knowledge from the user. **This is a benefit --- we want research code to be usable by practitioners**. The ease of use does not negate the technical novelty of our approach.
> > >
> > > To re-clarify, the core technical novelties of SMART from our paper:
> > >
> > > - Reframing model testing as a frequentist hypothesis testing problem and giving a statistical explanation of the mechanisms why current methods fail at testing ML models
> > > - New paradigm for ML testing: using context as an inductive bias to guide the search for meaningful model failures.
> > > - Showing that with context-aware testing, we can define and target task-relevant subgroups, limiting the number of tests conducted with better false positive control and greater statistical power.
> > >
> > > It seems you have not acknowledged the reframing part of model testing as our contribution, even though it forms a significant part of our paper.
> > >
> > > Additionally, our model reports provide insights to practitioners about the tests.
> > >
> > > ---
> > > **Comment 4**:  For instance, in the OpenML benchmark, datasets with the same data but different IDs can have vastly different descriptions. Can we truly trust these outcomes?
> > >
> > > **Answer 4**. You're correct that different datasets might have different feature names. That's okay.
> > > - SMART doesn't rely solely on feature names. It also leverages feature names and basic statistics of the data, which are consistent across different descriptions of the same dataset.
> > > - Self-falsification: While SMART uses descriptions to generate hypotheses, they are validated with actual data via our self-falsification mechanism. This ensures that regardless of the description, only hypotheses supported by the data are retained.
> > >
> > > We also ensure trustworthiness via the transparency and auditability provided by the model reports.
> > >
> > > **Comment 5**: Removing feature names hurts performance.
> > >
> > > We agree that removing feature names would affect SMART — as we’ve shown in **Fig 1 (Response pdf)**.  This highlights that feature names play an important role in finding model failures. Hence, SMART should not be used in settings with no feature names (i.e. we provide guidance to practitioners).
> > >
> > > ---
> > > *We hope these help clarify things.  Please let us know.*

---

> > > > ### Author Response · Authors · 2024-08-13
> > > >
> > > > Dear reviewer ctad,
> > > >
> > > > We thank you once more for the exchange we had. We hope it has been clarified that there is no need to provide feature descriptions, that our method is practically easy to use, and that SMART is cheap to use (testing a thousand hypotheses is cheaper than a morning espresso cup ☺️).
> > > >
> > > > Have our responses and clarifications addressed your main questions and concerns? If so, we kindly ask that you consider increasing your score from to better reflect this. If not, please let us know what specific unadressed issues remain, and we'll promptly provide additional information to resolve them.
> > > >
> > > > We appreciate your reconsideration and look forward to your feedback.

---

> > > > > ### Comment · Reviewer_ctad · 2024-08-13
> > > > >
> > > > > Thank you for the reply. As the authors replied, my main point is that "SMART should not be used in settings with no feature names".
> > > > > This implies that results could vary significantly when feature names are corrupted or deleted, even if the dataset and the fitted model remain exactly the same. Contrary to the authors' claim that "Feature names (e.g. column labels such as sex, age, race etc) are present in almost all tabular datasets both in the research field and in the real world where data is stored in SQL tables with column names", this is not an uncommon scenario in practical use cases. For example, in one of the largest tabular benchmarks, OpenML, it is easy to find multiple versions of the exact same datasets with different feature names.
> > > > >
> > > > > I would like to emphasize that this is a critical limitation when introducing LLMs in tabular deep learning problems. The authors should address this issue prominently in the manuscript, and the ablation study (as conducted in the rebuttal) should be included in the paper.
> > > > >
> > > > > Even if this critical aspect is resolved, I still have concerns about the novelty of this study, as it merely introduces an LLM to identify model failures without sophisticated design or sufficient insights.

---

> ### Author Response · Authors · 2024-08-14
>
> Thanks for your continued engagement. We're glad we've addressed prior concerns and reached a common understanding on some points. We clarify the remaining comments, which we hope persuades you to increase your score.
>
> **a) Should we be using SMART without clear feature names?**
>
> We're glad that we're in agreement that we also do not recommend using context-aware testing, and SMART specifically without feature names. While we see this reliance on the contextual nature of feature names as a benefit (by the nature of context-aware testing), we acknowledge we can be more explicit about this. We will expand the discussion section that data-only methods should still be the go-to testing approach in the _rare cases in practice_ where feature names are uninformative.
>
> **b) What about when feature names change?**
>
> Regarding your OpenML benchmark example, SMART can handle variations in feature names as long as they remain informative (e.g., "age" vs "patient_age"). We have consistently demonstrated in the paper that SMART can generate meaningful hypotheses with diverse column names. Thus, changing the name would not affect the reliability. That said, we have another safety mechanism in place --- our *self-falsification mechanism* (L209-224) which uses empirical data to evaluate each hypothesis. *Finally, to clarify, in real applications in both research and industry, we find that the models are tested against a single dataset version containing informative feature names.*
>
> **c) Is this a critical limitation in deep learning tabular problems?**
>
> You also mention that this is a limitation with "LLMs in tabular deep learning problems". This is the first time you have raised this concern. However, this critique is simply not true. As discussed, both our framework context-aware testing and our method SMART specifically are model-agnostic (discussed in L140-162). We give both theoretical and empirical reasons for this.
>
> - **We show theoretically why our framework is model agnostic**. We develop context-aware testing because we show that the alternative --- data-only testing--- implictly falls into the 'multiple testing problem' (Sec. 3.1.). This issue of ML testing does not depend on the underlying model. We address these theoretical issues of data-only testing methods (Sec. 3.2.) *even in deep learning models*.
> - **We provide empirical results to support this claim with deep learning methods**. In addition to our original evaluation which included MLPs (**Table 12 - main paper**), we have updated with two new deep learning tabular models (**Table 4 - response pdf**).
>
> **d) What is the novelty of our study?**
>
> We're quite surprised to see your comment that our novelty is limited.
>
> In your 'summary' section of the original review, you summarized the following (taken as quotes):
> - Unlike previous methods that rely solely on data to find slices where a model's predictions underperform compared to average performance, CAT addresses the limitations of these data-only approaches.
> - Traditional methods often fail to consider the problem context and can lead to multiple testing problems.
> - To overcome these issues, the authors propose loosening the restrictive assumption of relying solely on data
>
> In addition, you highlighted our contributions in the strengths:
> - 'the core limitation of data-only testing—the lack of a principled failure mode discovery procedure that can incorporate prior knowledge to guide the search for meaningful errors—is also persuasive.'
>
> We are concerned that there may be some **misalignment between your original review and your current comments**.
>
> To wrap up the discussion around our contributions, it seems that you acknowledge that we:
> - **Identify a fundamental limitation in testing methods** which rely only on data (Sec. 3.1 in the paper).
> - **Provide a theoretical mechanism explaining why data-only testing falls short**, relating it to the foundational view in statistics of multiple model testing in frequentist statistics. (Sec. 3.2)
> - **Proposing a new paradigm which avoids the theoretical issues**: selectively testing the most relevant tests instead of testing for everything --- and falling into the multiple hypothesis testing problem again; (Sec. 3.3)
> - **Building a concrete system - SMART** which is an auditable, easy-to-use, automated system as an alternative to data-only testing (Sec. 4)
> - **Showing that this works empirically** on over 12 quantitative & 5 qualitative evaluations. (Sec. 5)
>
> Therefore, we respectfully disagree with your most recent comment that our work "merely introduces an LLM to identify model failures" --- which also overlooks our clarification in our previous **Answer 3**
>
> **e) Finding consensus**.
>
> We hope our responses and clarifications addressed your main questions and concerns? If so, we kindly ask that you consider increasing your score to better reflect this. If not, please let us know what specific unadressed issues remain, and we'll do our utmost to resolve them.

---

### Official Review · Reviewer_H6db · 2024-07-11

**Soundness:** 3
**Presentation:** 3
**Contribution:** 3
**Rating:** 6
**Confidence:** 4

**Summary:**

The paper introduces Context-Aware Testing (CAT), a new method for testing machine learning (ML) models. Current ML testing methods rely only on data, which often leads to high false positive and false negative rates and misses meaningful failures. CAT improves this by adding external knowledge or context to the testing process. The paper presents SMART Testing, the first version of CAT, which uses large language models (LLMs) to identify potential failures and a self-falsification mechanism to test them. The results show that SMART Testing finds more relevant and impactful failures than traditional data-only methods.

**Strengths:**

1. The paper is well-written and easy to follow.
2. The concept of Context-Aware Testing represents a significant advancement in ML testing, offering a new perspective that goes beyond traditional data-only methods.
3. The SMART Testing framework effectively uses large language models (LLMs) and self-falsification, which demonstrates the practical application of CAT.
4. The paper presents valuable empirical evaluations that demonstrate the effectiveness of SMART Testing in identifying significant model failures.

**Weaknesses:**

1. The effectiveness of CAT depends on the quality and relevance of the contextual information. While context information may be available, effectively using it to generate meaningful hypotheses and justifications is not guaranteed, and there are cases where CAT may not be effective.
2. Using LLMs for hypothesis generation could introduce biases from the data used to train these models, potentially impacting the reliability of the testing results.

**Questions:**

1. How can we identify the most relevant contextual information for specific applications? How do we know that one context information is more valuable than another?
2. Are there alternative methods or models that could complement or substitute LLMs for hypothesis generation in CAT?
3. Are there any strategies to mitigate biases inherent in LLMs when using them for hypothesis generation in CAT?
4. Can the SMART Testing framework be adapted to handle non-tabular data such as images?

**Limitations:**

1. The paper mainly concentrates on tabular data and does not investigate how CAT could apply to other data types like images.
2. The reliance on LLMs for hypothesis generation in CAT poses a constraint, particularly in scenarios where LLMs are not accessible or applicable. This dependency may restrict the method's usability and applicability across diverse settings and applications.

---

> ### Author Rebuttal · Authors · 2024-08-06
>
> Dear R-H6db,
>
> Thank you for your thoughtful comments to improve the paper. We provide answers (A)-(E) & highlight updates to the paper
>
> ---
>
> # (A) Clarifying contextual information
>
> We'd like to clarify that when we refer to "context", we mean the *relevant background knowledge of a system (e.g. an LLM) that informs the generation of targeted hypotheses*.
>
> (i) Dependence on context: To clarify, we don't _require_ external contextual information or human expert input. For tabular data, context is inherently encoded in the feature names. Our focus on tabular data leverages this built-in context. For instance, in a medical dataset, features like age, sex, or patient features provide context to guide LLM hypothesis generation. This contrasts with data-only approaches which only use the numerical data values alone and ignore the context surrounding the feature names.
>
> (ii) Identifying relevant context: We _do not_ require manual selection of contextual information. The relevance or value of the contextual information is automatically determined by the LLM: (a) when proposing hypotheses based on tabular data's inherent context and (b) via our self-falsification mechanism which empirically evaluates hypotheses and prunes those not supported by data.
>
> **UPDATE**: We will update Sec. 4 using the extra camera ready page to clarify this concept of context and how it is used.
>
> Finally, to highlight the minimal input needed, we thought it would be useful to show SMART's ease of use. i.e. additional contextual info input is not necessary but possible. If not provided the LLM leverages the context inherently encoded in the dataset `X`.
>
>
> ```python
> import SMART
>
> # Instantiate SMART
> model_tester = SMART('gpt-4')
>
> # Give desired context (this can be left as an empty string)
> context = """Find where a model fails. This is a prostate cancer prediction task."""
>
> # Load ML model
> model = XGBoost()
> dataset_description = X.describe()
>
> # Test the model
> model_tester.fit(X, context, dataset_description, model)
> ```
>
> **UPDATE**: include the code snippet in the revision to show usage of SMART.
>
> ---
> # \(B) Alternatives to the LLM for hypothesis generation
>
> To the best of our knowledge, the current main alternative for generating targeted hypotheses would be human input. However, this lacks SMART's automation and scalability. This underscores both the novelty of our LLM-based approach for targeted model testing. Additionally, this highlights exciting research opportunities to develop alternative targeted samplers for hypothesis generation within the proposed CAT framework.
>
> --
> # (C) LLM bias & mitigation strategies
>
> We agree with you that is important to look at directions for removing biases from SMART. Our discussion on precisely this issue can be found in **Sec 5.4**. We realize that the title of the section might not fully convey this and hence will rename it (see update below).
>
> To summarize, in **Sec 5.4** we have outlined several strategies to mitigate biases inherent in LLMs.
>
> - (a) Using the training dataset to guide hypothesis generation.
> - (b) Use our self-falsification mechanism to empirically test hypotheses against the actual data, thereby filtering out spurious hypotheses.
> - \(c) Transparent Reporting: SMART generates comprehensive model reports (example in Appendix D.7) that provide clear justifications for each hypothesis, allowing for human oversight and intervention if needed.
>
> Empirically, Sec. 5.4 (Table 4) demonstrates SMART's ability to correctly identify underperforming subgroups, even in scenarios where LLMs might have prior biases (e.g., ethnicity-related biases).
>
> **UPDATE:** Rename section 5.4 as “Assessing and mitigating potential LLM challenges and biases” to better flag this.
>
> ---
> # (D) CAT beyond tabular data - images
>
> Our focus on tabular data is intentional due to its importance in many high-stakes domains such as healthcare, finance, and criminal justice, making ML testing particularly impactful for these critical applications by addressing a significant gap in current data-only ML testing methodologies.
>
> Additionally, tabular data inherently includes context through feature names and metadata. This context is not naturally present in images (tensor of pixel intensities).  As discussed in Appendix A, extending CAT to other domains like images would require incorporating metadata to provide necessary context — which is often unavailable.
>
> While extending CAT to other domains is an interesting future direction, it's non-trivial. We hope future work can build on our work and adapt the context-aware paradigm to images.
>
> **UPDATE:** add this discussion to our discussion in Appendix A.
>
> ---
> # (E) Constraint on LLM: might not be accessible or applicable in all scenarios
>
> We wish to clarify the value of SMART even with possible constraints. First, LLMs are becoming increasingly accessible, especially from a cost perspective. As shown in **Table 1 (Response pdf)** it costs less than 0.10 USD for 5 hypotheses and less than 0.50 USD for 100 hypotheses for state-of-the-art LLMs.
>
> Second, we show that SMART outperforms data-only methods even when using cheaper, less capable LLMs like GPT-3.5 (**Appendix D.3**) which are widely available to the public.
>
> Third, research done within the ML community is not always accessible or applicable (as illustrated by the vast research agenda of deep learning which presupposes access to expensive GPUs). Similarly, we see our research as being important in pushing the frontier of ML testing despite presupossing access to an LLM.
>
> Finally, given the stark differences in the effectiveness of data-only and context-aware methods (refer to the global response table), we think it is extremely important for the research community to have access to improved testing methods for more reliable ML testing in high-stakes domains.
>
> ---
> # Thank you
>
> Thank you for your engagement. Given these changes, we hope you might consider revising your assessment of the paper's impact.

---

> ### Author Response · Authors · 2024-08-11
>
> Dear reviewer H6db
>
> We are sincerely grateful for your time and energy in the review process.
>
> We hope that our responses have been helpful. Given the limited time left in the discussion period, please let us know if there are any leftover concerns and if there was anything else we could do to address any further questions or comments :)
>
> Thank you!
>
> Paper authors

---

> ### Comment · Reviewer_H6db · 2024-08-12
>
> Thank you for your clarification. After reviewing your answer (A), I remain unclear about how the context information is generated. You mentioned that 'context is inherently encoded in the feature names' and that no external contextual information or human input is required. Does this imply that the LLM can automatically infer context purely from feature names?
>
> Consider two identical tabular datasets with the same feature names—such as name, age, gender, occupation, and income—but with different prediction objectives: one for deciding bank loans and the other for predicting job promotions within five years. Both have binary outputs (yes or no). How would the LLM infer different context in these two different scenarios?

---

> > ### Author Response · Authors · 2024-08-12
> >
> > Dear reviewer H6db,
> >
> > Thank you for engaging with our work.
> >
> > To clarify, we use 'context' in two distinct ways: (a) external input by a human (not strictly necessary but usually helpful for the framework for specifying the overall task) and (b) external knowledge (such as relevant background knowledge, domain understanding) within an external system (LLM). While we made a small footnote about this on page 5, we agree this is not enough and could have resulted in confusion about the nature of our work and will clarify this confusion.
> >
> > SMART uses an LLM to generate hypotheses about what to test for, *assuming that the feature names indeed encode clear information*, such as age.
> >
> > In your illustrative example where we have the same covariates but different meanings of the target label, it would be necessary to add external information/context about the nature of the task as a short input. The short description of the task is always easily available info when testing a model.
> >
> > **Example when we are predicting whether to give bank loans**
> >
> > ```python
> > import SMART
> >
> > # Instantiate SMART
> > model_tester = SMART('gpt-4')
> >
> > # Give desired context <--- need to specify this in your example
> > context = """Find where a model fails. This is a prediction task about whether to give bank clients bank loan or not."""
> >
> > # Load ML model
> > model = XGBoost()
> > dataset_description = X.describe()
> >
> > # Test the model
> > model_tester.fit(X, context, dataset_description, model)
> > ```
> >
> > Notice the variable ``context`` is used as an input to SMART which provides required information about the nature of the task. In case this task is about job promotion within the upcoming five years, this would have to be changed as follows:
> >
> > ```context = """Find where a model fails. This is a dataset about whether people get promoted within a 5-year period. """```
> >
> > You are correct that it would not be possible to identify what the task is if the target label (`y`) is not labeled in a clear manner (e.g. `5_year_job_promotion`) or if this is not given any additional context, such as the example perovided in the variable `context`.
> >
> >
> > When we say that there is no need for manual human testing, we mean that there is no need to provide explicit information about *what to test for* and the only human input is the variable `context` which provides required information. Within SMART, we use a prompt template (provided in **Appendix C.3.**) that passes information about the context task, description of covariates, and feature names, which generates hypotheses about where the model is likely to fail which are then evaluated.
> >
> > **UPDATE**: To address this confusion, we will rename the context input as external input which we hope will disambiguate the two meanings.
> >
> > Does this help to clarify your question?

---

> > > ### Comment · Reviewer_H6db · 2024-08-12
> > >
> > > Regarding my question about how we determine that one piece of contextual information is more valuable than another: Is it important to identify and prioritize more valuable context, or does it not matter how precisely the context information is defined for the LLM? Could you comment on whether the quality of the context information affects the effectiveness of your context-aware testing approach?

---

> > > > ### Author Response · Authors · 2024-08-12
> > > >
> > > > Thank you for your engagement again.
> > > >
> > > > We attempted to answer this in our response.
> > > >
> > > > *Identifying relevant context: We do not require manual selection of contextual information. The relevance or value of the contextual information is automatically determined by the LLM: (a) when proposing hypotheses based on tabular data's inherent context and (b) via our self-falsification mechanism which empirically evaluates hypotheses and prunes those not supported by data.*
> > > >
> > > > To address your follow-up, we answer each question in turn. We hope this helps clarify the use of contextual information.
> > > >
> > > > **Q1: How do we determine if one piece of contextual information is more valuable than another?**
> > > >
> > > > **A1**: In practice, there is no need to compare or determine different pieces of contextual information. We consistently find that any description of the task in free-word form is sufficient to outperform all current ML model testing practices. This is true across the seven datasets (Appendix D.2, D6), five ML models (Appendix D.4) and five different LLMs tested (Table 2 in response pdf), as well as consistent across different sample sizes (Fig 9). However, feature names that are informative are required for this (we do not recommend using SMART in case feature names are uninformative). We have shown that if the feature names are not informative (e.g. `feature_1` instead of `age`), SMART does not consider these features. We consider this a benefit by providing robustness to irrelevant features (Fig 4.)
> > > >
> > > > **Q2: Is it important to identify and prioritize more valuable context, or does it not matter how precisely the context information is defined for the LLM?**
> > > >
> > > > **A2**: To echo our previous response (A1), the specific wording of the task does not matter. In fact, in many cases it can be left empty if the target value is clearly encoded (e.g. ``default`` instead of``y`` for a loan default prediction task) in the name. Why is this the case? The primary benefit from our method comes from testing only a few, selected subgroups instead of exhaustively testing for everything. This requires generating a few hypotheses which we find are not sensitive to the exact wording of the ``context`` variable. For instance, changing the word from ``context = this is a loan prediction task`` to ``context = this task is about loan prediction`` does not affect its output. As per before, no other context is strictly required for our method, and the vast majority of our 12 quantitative experiments (refer to global response) do not use any more context than in the example provided (except when this is required to show some property or feature of SMART).
> > > >
> > > > **Q3: Could you comment on whether the quality of the context information affects the effectiveness of your context-aware testing approach?**
> > > >
> > > > **A3:** That largely depends on what is meant by "quality of the context". In case you are talking about the context variable description (e.g. ``context = 'this is a loan prediction task')`` being actively misleading, then we do not expect SMART to perform well. However, we do not recommend using this method with actively misleading prompts. If by "quality of context" you mean whether the feature names are interpretable, then we show in **Table 3 and Figure 1 in the response pdf** that SMART cannot generate meaningful hypotheses if the feature names are not interpretable. This is expected --- as the generation of meaningful hypotheses relies on having access to feature names. To give more intuition as to why the specific context does not matter, we provide concrete model reports that SMART generates (**Appendix D.7.**) which can showcases what tests were performed. Lastly, this can be used to inspect which piece of context was and was not meaningful in generating the hypotheses.
> > > >
> > > > ---
> > > > Fundamentally, we find that adding only minimal context in any free-form text, as long as the task description is clear, outperforms any existing ML testing method to-date. This is consistent across 12 quantitative experiments (refer to global response).
> > > >
> > > > We apologize this was not clear at the beginning.
> > > >
> > > > **Actions taken**: In response to your questions, we will add a broader discussion of the role context plays in the introduction and method sections.
> > > >
> > > > Does this better address your questions and help to clarity the paper's contribution?

---

> > > > > ### Comment · Reviewer_H6db · 2024-08-12
> > > > >
> > > > > Thank you for your efforts in clarifying your work. From what I understand, the context information in SMART is either provided through a context variable (e.g., "context = this is a loan prediction task") or implicitly inferred by the LLM from feature names. The approach using the context variable only requires providing the necessary background information of the task to the "context" variable, without the need for skillful or careful wording. I hope I have correctly understood the concept of context information in your framework.
> > > > >
> > > > > Based on our discussion, I notice a gap between the description of context-aware testing in Section 3.3 and what is actually implemented in SMART. From line 143, it appears that context information c = {c1,…, ck} must consist of k carefully chosen context elements, and that the sampling mechanism \pi(c, D, m) (line 149) would be sensitive to the choice of c. However, it seems that c is either a simple sentence providing background information or something inferred by the LLM itself. I’m concerned that readers may not clearly grasp this distinction when reading Section 3.3.
> > > > >
> > > > > Another gap I see between your explanation of the SMART methodology and its actual implementation is in Step 1 of Section 4. Line 197 states, “we sample the N most likely hypotheses of failures, H = {H1,…, HN}.” My concern here is how you determine that the N hypotheses selected are indeed the N most likely hypotheses of failures. Since all N hypotheses are generated by the LLM, do you require the LLM to generate more than N hypotheses and then select the most likely ones? What criteria are used to evaluate which hypotheses are more likely to lead to failures than others? In my view, the N hypotheses generated by the LLM are merely proposed hypotheses, and it’s unclear whether better ones exist in terms of their likelihood to trigger model failures.

---

> > > > > > ### Author Response · Authors · 2024-08-12
> > > > > >
> > > > > > We're happy we've clarified the word context in our conversation. Your understanding about "context" is exactly correct, and we'll change our writeup to better mimic our discussion here (thank you for this valuable discussion).
> > > > > >
> > > > > > ---
> > > > > > **Question 1. Gap between description and SMART implementation**. Great catch --- you're entirely right and we missed this detail ourselves. We now see where the confusion comes from.
> > > > > >
> > > > > > To clarify (what you've already point out), the context **c** should not be a vector of contexts but rather a single piece of contextual information.
> > > > > >
> > > > > > While the definition of the input space for $\pi$ in line 148 is correct in that $\pi$ takes a single piece of context (a text string), this is not obvious from the description and writeup. Furthermore, we agree with you that it should be clearer that the sampling mechanism $\pi$ is a function (e.g. an LLM) that is using its inductive bias to map inputs to outputs.
> > > > > >
> > > > > > **Actions taken**: (1) As stated before, we will change the word `context` to *external information* where we indeed mean external information, such as the nature of the task; (2) We will adjust the definition to not be a vector of contexts. (3) We will give more focus on the fact that the sampling mechanism $\pi$ itself is a context-aware mechanism, i.e. uses its inductive bias to generate meaningful model failures.
> > > > > >
> > > > > > This discussion has been extremely fruitful.
> > > > > >
> > > > > > ---
> > > > > > **Question 2. Clarification on hypothesis evaluation**.
> > > > > >
> > > > > > When we say we sample *n* most likely hypotheses, we mean that we generate *n* hypotheses within the LLM by explicitly stating how many hypotheses and justifications to generate within the input prompt.
> > > > > >
> > > > > > However, as you correctly pointed out, this does not necessarily imply that the first hypothesis is more likely to find a model failure than the second one. At this point, the *n* generated hypotheses are just proposals. Therefore, it's not clear whether they actually contain subsets of data where the model fails the most or not.
> > > > > >
> > > > > > This is why we also have steps 2-3 of our SMART method.
> > > > > >
> > > > > > Step 2: we operationalize each hypothesis (convert a textual description to a clear criterion, e.g. "elder people" -> `age > 70` (lines 200-208 in the paper).
> > > > > >
> > > > > > Step 3: self-falsification. Here, we perform hypothesis testing on a validation dataset to evaluate each hypothesis on actual data and compute the model's performance on that specific subset of the data which has been hypothesized. Therefore, for each hypothesis, we compute the difference between the aggregate performance and the performance on that subset of data.
> > > > > >
> > > > > > With this, we can answer your question:
> > > > > >
> > > > > > **Question:** Which criteria do we use to evaluate which hypotheses are more likely to lead to model failure than others? **Answer**: We use the performance discrepancy (e.g. difference between accuracies) between the model overall and that hypothesized slice of data on a validation dataset. Then, SMART returns the slices of data which have largest performance discrepancies. This is explained in lines 221-224.
> > > > > >
> > > > > > ---
> > > > > > **Worked out example for Question 2.**
> > > > > > We thought it might be useful to give a simple toy example of SMART.
> > > > > >
> > > > > > **Step 1.** Suppose SMART generates *n=3* hypotheses (adapted from Appendix D.3.):
> > > > > >
> > > > > > - H1: The model will perform worse for younger people as there is insufficient data for young people
> > > > > > - H2: The model will perform worse for patients with multiple comorbidities due to the complexity of the disease
> > > > > > - H3: The model will perform worse for patients with stage 4 cancer because it might progress unpredictibly.
> > > > > >
> > > > > > **Step 2.** We can operationalize each hypothesis (done automatically):
> > > > > > - H1: `'age < 35'`
> > > > > > - H2: `'comorbidities >= 2'`
> > > > > > - H3: `'cancer_stage == 4'`
> > > > > >
> > > > > > **Step 3.** We then use these definitions to test the model on a validation dataset (done automatically). Suppose the overall accuracy on a validation dataset is 90\%. Then suppose for these subgroups, we obtain:
> > > > > >
> > > > > > - H1: `'age < 35'`. Accuracy: 70%. Lower by: 20%. p-value: 0.02
> > > > > > - H2: `'comorbidities >= 2'`. Accuracy: 90%. Lower by: 0%. p-value: 1
> > > > > > - H3: `'cancer_stage == 4'`. Accuracy: 50%. Lower by: 40%. p-value: 0.00
> > > > > >
> > > > > > Then, we find that people with ``cancer_stage == 4`` is the data slice with the highest performance discrepancy, followed by ``age < 35``.
> > > > > >
> > > > > > We note that this is done automatically as a part of SMART.
> > > > > >
> > > > > > ---
> > > > > > We genuinely appreciate the time you put into understanding our work and helping us improve. We hope we have addressed your concerns.

---

> > > > > > > ### Comment · Reviewer_H6db · 2024-08-13
> > > > > > >
> > > > > > > Thank you very much for your rebuttal. Your answers have adequately addressed my concerns.

---

> > > > > > > > ### Author Response · Authors · 2024-08-13
> > > > > > > >
> > > > > > > > Dear reviewer H6db,
> > > > > > > >
> > > > > > > > We're happy we've addressed your concerns --- thank you for helping us improve the paper. We believe our framework as well as our practical method (SMART) will be used by researchers and practitioners who regularly test ML models.
> > > > > > > >
> > > > > > > > Given these changes, we hope you consider revising your assessment of the paper's impact and the corresponding evaluation for NeurIPS. ☺️

---

> ### Author Response · Authors · 2024-08-13
>
> Dear reviewer H6db,
>
> Thank you for the engaging exchange we had that helped to clarify our work. As the discussion period is coming to a close, we'd like to follow up on your prior concerns.
>
> Have our responses and clarifications addressed your main questions and concerns? If so, we kindly ask that you consider increasing your score from a 5 to better reflect this. If not, please let us know what specific issues remain, and we'll promptly provide additional information to resolve them.
>
> We appreciate your reconsideration and look forward to your feedback.

---

> > ### Comment · Reviewer_H6db · 2024-08-13
> >
> > Thank you for your efforts in developing SMART and for your thoughtful engagement in the rebuttal process.
> >
> > Based on my latest understanding of the context, I believe it’s crucial to reconsider how the role of context in SMART is communicated. On one hand, overestimating the role of context could be misleading, particularly given the black-box nature of LLMs, where it’s not clear how they actually utilize contextual information. On the other hand, simply stating that "LLMs + context" magically works might understate the significance of your contributions, potentially giving the impression that the work lacks strong scientific insights.
> >
> > I feel that the current version of the paper doesn’t fully address these concerns. Could you find a more balanced way to present this aspect of your work before the end of the discussion period? I would be inclined to raise my score if these issues are addressed effectively.

---

> > > ### Author Response · Authors · 2024-08-13
> > >
> > > Many thanks for the continued engagement to improve the paper. We'll put everything we've talked above into a single answer which can hopefully address this concern (such that no one is left with the impression that our paper claims "LLMs + context" magically works" ☺).
> > >
> > > **Q1: How can the role of context in SMART be more accurately communicated?**
> > >
> > > **A1**: To summarize and wrap up the changes we're making on context (valuable discussion -- thanks again):
> > > - Emphasize that context-aware testing is a general framework and SMART is one implementation.
> > > -  **Change the way we use the word context**. Clarify that "context" in CAT refers to two aspects: (i) Minimal task description (to be renamed to external input); (ii) The LLM's inductive biases ('knowledge') used for hypothesis generation
> > > - **We rely on both context and data**. Emphasize that SMART *does not* rely solely on context, but on a combination of hypothesis generation and data-driven evaluation (which forms a significant part ouf our evaluation framework).
> > > - **Context-aware testing avoids the multiple testing problem prevalent in data-only testing**. Link clearly the context-aware mechanism with the problem with data-only testing to better motivate this design choice, explaining that context helps to avoid the multiple hypothesis testing problem implicit in current data-only ML testing (Sec. 3.1.).
> > > - **Highlight ablation study**. Show with an ablation study that not using data results in much worse performance --- a context-guided sampling mechanism (LLM) is not sufficient (ablation study already performed -- **Appendix D.3.**).
> > >
> > > **Q2: How do we address the black-box nature of LLMs?**
> > >
> > > **A2**: While we answer this concern throughout the paper in separate sections, we'll use the additional paper in the camera-ready version to summarize everything into a single paragraph:
> > >
> > > - Using LLMs offers concrete insights into which hypotheses are tested and why (Table 1 in our paper). This is not the case for data-only methods which test for everything, resulting in the issues described in  Sec 3.1-3.2.
> > > - The testing procedure is fully auditable via model reports (unlike data-only methods), described in Appendix D.7.
> > > - Our *self-falsification* module uses empirical data to validate the hypotheses on actual data.
> > >
> > >
> > > **Q3: Can we strengthen scientific insights?**
> > >
> > > **A3**: We firmly believe that strong foundational statistical intuition which explains complicated phenomena are extremely valuable as scientific insights. That's why we employ a foundational view in statistics --- multiple hypothesis testing --- to explain for the first time: _why_ and _when_ data-only methods fail in finding ML model failures (Sections 3.1-3.2).
> > >
> > > We'll add more statistical intuition why using a context-aware sampling mechanism (e.g. an LLM) resolves the problem of multiple hypothesis testing in ML evaluation (to add after line 165). We'll further link it to our experimental confirmation of these insights, where we indeed show that our theoretical analysis conforms with our experiments on false positives in Sec. 5.1. or false negatives in Sec. 5.2.
> > >
> > >
> > > ---
> > > These should fully clarify the (i) meaning of context; (ii) value of context; (iii) mechanism why it works, in addition to the practical benefits.
> > >
> > > Given the above outlined, we trust that future readers will find our work's contributions and scientific rigor to be more apparent. Thank you for engagegment and with the above in mind we hope you reassess the paper's impact on the field of ML testing---and the associated score for NeurIPS. ☺

---

> > > > ### Comment · Reviewer_H6db · 2024-08-13
> > > >
> > > > Thank you for your response. Your answers have addressed my concerns, and I would like to raise my score from 5 to 6.

---

> > > > > ### Author Response · Authors · 2024-08-13
> > > > > **Thanks for the score increase & the engagement**
> > > > >
> > > > > Dear reviewer H6db,
> > > > >
> > > > > We are glad our response was helpful and would like to thank you for raising your score!
> > > > >
> > > > > We really appreciate your feedback and the time you put into engaging with our work, which has helped us improve the clarity of the paper ☺ We will make changes to the revised paper to reflect our discussion. Thanks again!
> > > > >
> > > > > Regards
> > > > >
> > > > > Paper Authors

---

### Official Review · Reviewer_bfyv · 2024-07-12

**Soundness:** 3
**Presentation:** 4
**Contribution:** 3
**Rating:** 6
**Confidence:** 3

**Summary:**

The paper offers a multiple hypothesis testing view of ML evaluation (in regard to test slice
finding). Authors identify problems with data-only approaches (like high amounts of false positive
and false negative model failure triggers). The paper proposes SMART, a context-aware LLM based ML
model testing methodology to mitigate said problems. SMART uses LLMs for hypothesis generation (data
slices where model could potentially underperform). The method is empirically evaluated and compared
to a set of data-only baselines, showing less false-positives and recovering more model failure
modes.

**Strengths:**

The paper is well structured and well written, it is easy to follow and is enjoyable to read. It
tackles an important problem in model testing at an interesting angle, identifies an issue with
existing (and common) methodology. The paper presents a novel, clever and well motivated solution
that incorporates strong sides of LLMs (contextualization, hypothesis generation) with strong sides
of prior data-based solutions. The discussion on mitigating potential LLM challenges is important
and well considered.

**Weaknesses:**

1. *It is unclear how the method degrades with less informative context*.
   - Most experiments validate the method in conditions well suited for it. e.g. easily
     distinguishible irrelevant features (could you clarify what are the feature names in the
     experiment, is it truly the case that the LLM could easily differentiate them from potentially
     useful features?)
3. *Some details of the method are missing from the main text*
   - What are the prompts (better to include some information on this in the main text, I had questions when exploring the experimental results)
   - Not sure from which subset of the data the slices are selected (it is intuitive that slices are
     subsets of the test data, but if I'm not mistaken it is not clearly defined in the text).
4. *Some related work may be missing*. I think that the subject of ML model testing in industry and production scenarios might be
   related, while it is not discussed in the paper [1, 2].

**Questions:**

- Is it possible to run a method when the data schema is limited (e.g. some feature names are unknown, and the context is rather shallow in general)?
- Could you provide experiments on more real world datasets with real known model failure cases? The
  TableShift [3] benchmark could help with such an experiment as it already has subsets (OOD eval sets)
  that could serve as ground-truth problematic data slices.
- In what form the code would be available? I suggest making a package with said methodology to increase adoption.
- Are open-weights free LLMs able to generate sufficiently high quality hypothesis for the method to work?

References:
- [1] Shankar, Shreya, et al. "Automatic and precise data validation for machine learning." Proceedings of the 32nd ACM International Conference on Information and Knowledge Management. 2023.
- [2] Polyzotis N. et al. Data validation for machine learning //Proceedings of machine learning and systems. – 2019. – Т. 1. – С. 334-347.
- [3] Gardner, Josh, Zoran Popovic, and Ludwig Schmidt. "Benchmarking distribution shift in tabular data with tableshift." Advances in Neural Information Processing Systems 36 (2024).

**Limitations:**

Limitations are adequately addressed

---

> ### Author Rebuttal · Authors · 2024-08-06
>
> Dear reviewer bfyv,
>
> Thank you for taking the time to carefully review our work. We provide answers below (A-F) and highlight our paper updates.
>
> ---
> ## (A) Method degradation with less informative context
>
> We appreciate your interest in SMART's performance under varying context quality. We've developed the context-aware framework primarily for the tabular domain because feature names encode useful information for reasoning about potential failures. Therefore, we *do not* recommend using CAT when feature names do not encode useful information.
>
> We address your concern in two ways. First, we perform a qualitative study where we limit the data schema by hiding the feature names and inspect the hypotheses and justifications generated. We find that in the limited-schema case, SMART generates hypotheses based on inferences about the feature information (e.g. "the model might fail on feature_4 if feature_4 represents gender"). In contrast, informative names guide meaningful hypothesis generation. Such hypotheses and justifications are illustrated in **Table 3 in the response pdf**.
>
> Second, we evaluate whether limiting the data schema by hiding some feature names and leaving minimal external context affects detection rates of model failures. We compare two versions of SMART, original and with corrupted feature labels, in identifying data slices with high performance discrepancies from average (**Fig. 1 in the response pdf**). We find that across two real-world private datasets, hiding the feature names hinders model evaluation. This highlights that feature names play an important role in finding model failures.
>
> **UPDATE**: We'll discuss SMART's sensitivity to feature names in the appendix.
>
> ---
> ## (B) Moving content from the appendix to the main text
>
> We agree with you that highlighting the specific prompt structure used would improve readability and we'll add more discussion in the camera-ready version.
>
> We will also clarify our data subset selection methods (slices are always selected based on train data and evaluated on test data).
>
> ---
> ## \(C) Related work
>
> We appreciate you bringing attention to relevant related work. Both works [1] and [2] differ from our approach in that they focus on data validation techniques for general ML pipelines, while our context-aware testing framework specifically aims to use context to guide the search for model failures.
>
> **UPDATE**: We will cite these works in the revision and provide a discussion of how context-aware testing differs from existing data validation techniques used in industry.
>
> ---
> ## (D) Comparison on TableShift benchmark
>
> While TableShift [3] is a useful benchmark, it's not directly applicable to SMART. TableShift primarily focuses on OOD-detection. In contrast, SMART searches for model failures within the existing data distribution (recall that the context-guided sampling mechanism in Def. 1 samples from the existing data distribution).
>
> That said, we agree that addressing dataset shift would be a valuable extension of context-aware testing. We'll mention this as future work, highlighting TableShift as a potential benchmark for extending context-aware testing to the OOD domain.
>
> ---
> ## (E) Code release
>
> We expect to release the code in the form of a package mirroring your suggestion. We demonstrate the usage capabilities here and will expand it in the appendix.
>
> ```python
> import SMART
>
> # Load SMART
> model_tester = SMART('gpt-4') # Instantiates smart
>
> # Give desired context
> context = """Find where a model fails. This is a prostate cancer prediction task."""
>
> # Load a desired ML model
> model = XGBoost()
>
> # Test the model
> model_tester.fit(X_train, context, model)
> ```
>
> Once fitted, many attributes can be accessed, such as found subgroups ```model_tester.subgroups```, generated hypotheses ```model_tester.hypotheses```, justifications ```model_tester.justifications```, model report ```model_tester.model_report```, and more. We can also evaluate it: ```print(model_tester.top_underperforming_subgroup)```
>
> ```python=
> Overall accuracy of the model: 85.1%
> Accuracy on 'Age > 75' subgroup: 55.2%
> Discrepancy of: 29.9%
> ```
>
> **UPDATE:** Add code usage to appendix.
>
> ---
> ## (F) Open-weight models for hypothesis generation
>
> Before jumping in, we wish to highlight **Appendix D.3**, which notes our goal is not to exhaustively test the SMART framework with every LLM. Rather, the goal is to showcase that SMART is feasible with at least the capabilities of GPT-4. While we assessed SMART's performance with different LLMs, we caution against using it with less capable models.
>
> That said, based on your comment we have conducted an experiment with Mistral-7b, Qwen-1.5-7b, Llama-3-8b, Llama-70b, where for the OULAD and SEER datasets we generate 5 hypotheses and assess overlap to the hypotheses generated by GPT-4. This is presented in **Table 2 in the response pdf**.
>
> To summarize, the overlap between open-source models and GPT-4 is between 60-80%. We find that open-source models propose similar hypotheses, but they are not replacements for more capable models. This corresponds to our findings in **Appendix D.6** --- less capable models might propose similar hypotheses, yet they still catch fewer model failures.
>
> **Closed-source models are cheap.** In case your worry is that closed-source models are expensive to run, we conduct an additional evaluation calculating the cost of running SMART with closed-source models (**Table 1 in the response pdf**). The cost does not scale with dataset size but with the number of tests conducted. We find that generating 100 hypotheses with a powerful closed-source model (GPT-4o-mini) costs less than 1 cent and using state-of-the-art models (GPT-4) costs about 0.30 USD.
>
> ---
> ## Thank you
>
> Thank you for your engagement. Given these changes, we hope you might consider revising your assessment of the paper's impact.

---

> > ### Comment · Reviewer_bfyv · 2024-08-09
> >
> > Thanks for the detailed response with clarifications and additional experiments. I believe my concerns and questions are adequately addressed. I intent to keep the score, as I already recommend acceptance.

---

> > > ### Author Response · Authors · 2024-08-12
> > >
> > > Dear Reviewer bfyv,
> > >
> > > Thank you for your thoughtful consideration of our paper and your positive response to our rebuttal.
> > > We appreciate your recommendation for acceptance and, if you feel it is appropriate, would be grateful if you could consider whether our paper might align more closely with the criteria for a score of 7 rather than 6.
> > >
> > > We believe our work meets the criteria for a score of 7 for these reasons:
> > >
> > > - High Impact: Our work improves ML testing by addressing limitations found in all of the current data-only methods. More effective ML testing enabled by SMART is impactful to improve the reliability, safety, and trustworthiness of ML models across various applications and industries.
> > >
> > > - Potential for Wide Adoption: Our implementation of SMART as an accessible Python package (mirroring your suggestion) has the potential for widespread adoption and impact within the ML community.
> > >
> > > - Thorough Evaluation: Our paper presents a comprehensive evaluation, including 11 quantitative and 2 qualitative experiments. We've also conducted 5 new experiments, providing additional insights into SMART's performance.
> > >
> > > We see that these points align with the criteria for a score of 7: "Technically solid paper, with high impact on at least one sub-area of AI or moderate-to-high impact on more than one area of AI, with good-to-excellent evaluation, resources, reproducibility, and no unaddressed ethical considerations."
> > >
> > > We respectfully ask you to consider revising your assessment based on these alignments.
> > >
> > > Regardless of your decision, we're deeply appreciative of your support and recommendation for acceptance. Your feedback has really helped us to improve the paper!
> > >
> > > Thank you again for your time and consideration.

---

### Official Review · Reviewer_6t3M · 2024-07-12

**Soundness:** 3
**Presentation:** 3
**Contribution:** 3
**Rating:** 6
**Confidence:** 3

**Summary:**

This paper introduces context-aware testing (CAT), which is a novel tabular model testing paradigm that uses context as an inductive bias to guide the search for meaningful model failures, and build a CAT system named SMART Testing. Detailedly, SMART includes four steps: (1) use an LLM to generate hypotheses of failures based on a combination of contextual information and dataset contextualization, (2) transform natural-language-based hypotheses into a form that can directly operate on the training dataset, (3) discard “incorrect” hypotheses by testing out each hypothesis through a self-falsification mechanism, and (4) automatically generate a report of the overall performance of the model under varying hypotheses. Evaluation shows that SMART is more robust than data-only methods in terms of false positives and false negatives, and can target model failures more accurately.

**Strengths:**

- The paper explores an important and interesting research direction.
- The improvements achieved by SMART shown in the evaluation are significant.

**Weaknesses:**

- There is a lack of systematic comparison between the practicality of testing subgroups found by SMART and other data-only methods. While the paper includes some examples of the reports generated by SMART, it would be better for authors to provide both qualitative and quantitative comparison between SMART and data-only methods in terms of the practicality of testing subgroups.
- The tabular models included in the evaluation are outdated. In the paper, SMART testing is only applied to evaluate the performance of simple machine learning models such as Logistic Regression and SVM. It would be better to include more up-to-date deep learning models, such as transformer-based models.

**Questions:**

- What is the cost of using GPT-4 to generate hypotheses in SMART and is it scalable to larger evaluation benchmarks?
- Since the idea of SMART is general, is it generalizable to other domains apart from tabular testing, such as testing NLP models?

**Limitations:**

- It would be good to discuss the potential future directions of removing biases from SMART.

---

> ### Author Rebuttal · Authors · 2024-08-06
>
> Dear R-6t3M,
>
> Thank you for your thoughtful comments to improve the paper. We provide answers (A)-(F) & highlight updates to the paper.
>
> ---
> ## (A) Highlighting our systematic comparison of SMART vs data-only
>
> We appreciate your suggestions and believe there may be a misunderstanding regarding the extent of our comparisons. We wish to clarify we provide in total **11 quantitative** and **2 qualitative experiments** (**see global response table above**), as well as **five new additional insight experiments** in the response pdf.
>
> **UPDATE:** To help clarify, we add a **new summary table (Global response table**) with the main takeaway and references to the relevant sections, tables and figures.
>
>
> We hope this addresses your concerns and demonstrates the thoroughness of our comparisons.
>
> ---
> ## (B) Clarifying tabular models used
>
> We thank you for your comment on the choice of downstream tabular models.  We clarify that a core contribution of SMART is its targeted sampling of hypotheses, which is entirely independent of the model used. As detailed in **Section 3.3 and Section 4**, SMART's context-guided slice sampling mechanism $\pi$ is used to generate hypotheses independently of the downstream model. This is why the specific tabular model used is not the primary focus of our paper.
>
> Despite this, we demonstrate SMART's effectiveness across multiple models, including Logistic Regression, SVM, XGBoost, and MLP, showing the applicability of our findings.
>
> While we understand the reviewer's concern about "outdated" models, we argue that our chosen models remain highly relevant for tabular data:
>    - Interpretability: Models like logistic regression offer crucial interpretability in domains such as healthcare and finance and are still widely used in industry.
>    - Performance: Recent work has shown that traditional ML models like tree-based boosting methods (XGBoost) often outperform deep learning approaches on typical tabular data [Grinsztajn et al., NeurIPS 2022] and we would argue that tabular deep learning has not been universally adopted for tabular data.
>
> That said, we have also run additional results with two tabular deep learning method: **TabPFN and TabNet (Table 4 in the response pdf)**. Across all the methods, SMART is the best at finding subgroups where the models are least reliable.
>
>
> ---
> ## \(C) Cost of LLM hypothesis generation & scalability to larger datasets
>
> We clarify the cost and scalability of SMART — demonstrating not only that SMART is cheap but also easily scalable to large datasets
>
> - Scalability: SMART's scalability depends on the number of hypotheses generated, not dataset size (unlike data-only methods). This allows SMART to easily scale to arbitrarily large datasets.
> - Cost Analysis: In practical terms, cost then also scales primarily with the number of hypotheses generated, not dataset size.  We provide a rough estimate based on token counts of input and outputs for 2 datasets (SEER and OULAD). This would be $<0.10$ USD for 5 hypotheses and $<0.50$ USD for 100 hypotheses for state-of-the-art models.
>
> | Model       | Cost SEER 5 Hypothesis (USD) | Cost SEER 100 Hypothesis (USD) | Cost OULAD 5 Hypothesis (USD) | Cost SEER 100 Hypothesis (USD) |
> |----|-----|----------|------|------|
> | GPT-4       | 0.017                  | 0.249                    | 0.022                   | 0.316                    |
> | GPT-3.5     | 0.004                  | 0.050                    | 0.005                   | 0.064                    |
> | GPT-4o Mini | 0.0003                  | 0.005                    | 0.0005                   | 0.006                    |
> | GPT-4o      | 0.008                  | 0.125                    | 0.011                   | 0.158                    |
>
>
>
>
> ---
> ## (D) SMART beyond tabular data (e.g. NLP)
>
> We appreciate your interest in the generalizability of SMART. Our current focus is on tabular data due to its importance in high-stakes domains like healthcare and finance,  as well as,  context being naturally encoded in tabular data which is used to guide the targeted sampling of failures. Extending SMART to other domains like NLP requires non-trivial adaptation, warranting dedicated investigation.
>
>
> Applying SMART to text would require addressing the lack of explicitly interpretable features and developing new ways to operationalize hypotheses on unstructured data. In contrast, tabular data offers inherent context due to the interpretability of features, which is crucial for context-aware hypothesis generation and operationalization.
>
> **UPDATE:** We'll use the extra camera-ready page to highlight this as an interesting future avenue for extending the context-aware testing paradigm.
>
> ---
> ## (E) Discussion on future bias mitigation when using SMART
>
> We agree with you that is important to look at directions for removing biases from SMART. Our discussion on precisely this issue can be found in **Sec 5.4**. We realize that the title of the section might not fully convey this and hence will rename it (see update below).
>
> To summarize, in **Sec 5.4** we have outlined several strategies to mitigate biases inherent in LLMs.
>
> - (a) Using the training dataset to guide hypothesis generation.
> - (b) Use our self-falsification mechanism to empirically test hypotheses against the actual data, thereby filtering out spurious hypotheses.
> - \(c) Transparent Reporting: SMART generates comprehensive model reports (example in Appendix D.7) that provide clear justifications for each hypothesis, allowing for human oversight and intervention if needed.
>
> **UPDATE:** Rename section 5.4 as “Assessing and mitigating potential LLM challenges and biases” to better flag this.
>
> ---
> ## Thank you
>
> Thank you for your engagement --- we believe these changes should improve the paper's contribution and clarity. Given these changes, we hope you might consider revising your assessment of the paper's impact and the corresponding evaluation for NeurIPS.

---

> > ### Comment · Reviewer_6t3M · 2024-08-13
> >
> > Thank you for your detailed rebuttal. While I still think it is important to extend SMART to scenarios beyond tabular data for more comprehensive evaluation, I would like to adjust my rating from 5 to 6.

---

> > > ### Author Response · Authors · 2024-08-13
> > >
> > > Dear reviewer 6t3M,
> > >
> > > We are glad our response was helpful and would like to thank you for raising your score! Thanks again for your time and suggestions which have helped us to improve the paper.
> > >
> > > Regards
> > >
> > > Paper Authors

---

> ### Author Response · Authors · 2024-08-11
>
> Dear reviewer 6t3M
>
> We are sincerely grateful for your time and energy in the review process.
>
> We hope that our responses have been helpful. Given the limited time left in the discussion period, please let us know if there are any leftover concerns and if there was anything else we could do to address any further questions or comments.
>
> Thank you!
>
> Paper authors

---

### Author Rebuttal · Authors · 2024-08-06

We thank the reviewers for their insightful and positive feedback!

We are encouraged that the reviewers found our work on model testing "important" (**R-6t3M, R-bfyv**) and "interesting" (**R-6t3M, R-bfyv, R-ctad**). They agreed our motivation for diverse perspective testing is "necessary for practical deployment" (**R-ctad**). Our Context-Aware Testing concept, which incorporates prior knowledge to guide the search for meaningful errors, was deemed "persuasive" and recognized as "a significant advancement in ML testing" (**R-H6db**) by "identifying an issue with existing (and common) methodology" (**R-bfyv**) and addressing the "core limitation of data-only testing" (**R-ctad**). The reviewers appreciated our "novel, clever and well motivated solution" (**R-bfyv**), which is the SMART testing LLM instantiation deemed to effectively demonstrate "the practical application of CAT" (**R-H6db**). Our "valuable" (**R-H6db**) empirical evaluations showed the effectiveness of SMART Testing in identifying significant model failures (**R-H6db**), demonstrating "significant" improvements (**R-6t3M**), complemented by our "important and well considered" discussion on mitigating potential LLM challenges (**R-bfyv**).


---
# Summary of experiments and findings
We'd like to highlight the experiments we've conducted as a summary. We hope these will clearly showcase the value of context-aware testing.


| Type | Figure/Table | Purpose | Finding |
|------|--------------|---------|---------|
| Quantitative | Figure 4 | Assess robustness to irrelevant features | SMART consistently avoids irrelevant features, outperforming data-only methods. |
| Quantitative | Table 9 | Evaluate ability to satisfy testing requirements | SMART satisfies most requirements while maintaining statistical significance. |
| Quantitative | Table 2 | Assess ability to identify significant model failures | SMART discovers slices with larger performance discrepancies across models. |
| Quantitative | Table 3 | Measure false negative rates in identifying underperforming subgroups | SMART achieves the lowest false negative rates in all settings. |
| Quantitative | Table 4 | Assess robustness to potential LLM biases | SMART effectively mitigates biases in identifying underperforming subgroups. |
| Quantitative | Figure 8 | Assess how sample size affects irrelevant feature detection | SMART consistently avoids irrelevant features regardless of sample size. |
| Quantitative | Table 10 | Evaluate performance in different deployment environments | SMART identifies more significant failure slices in new environments. |
| Quantitative | Figure 9 | Assess impact of sample size on performance | SMART consistently outperforms data-only methods across all sample sizes. |
| Quantitative | Table 11 | Evaluate tendency to flag non-existent failures | SMART avoids spurious failures, unlike data-only methods. |
| Quantitative | Table 6 | Compare performance of GPT-3.5 and GPT-4 in SMART | Both GPT versions in SMART outperform benchmark methods. |
| Quantitative | Table 12 | Evaluate SMART across different tabular ML models | SMART identifies larger performance discrepancies across various models. |
| Qualitative | Table 7 | Compare hypotheses generated by GPT-3.5 and GPT-4 | Both GPT versions generate similar, relevant failure hypotheses. |
| Qualitative | Appendix D.7 | Showcase SMART's practical output | SMART generates comprehensive model reports, providing clear justifications for each hypothesis. |

---
# Response pdf
In addition to the above experiments, we provide new insights based on our discussions. In the response pdf, we provide the following:

- **Table 1. Cost of SMART (USD)**. We quantify the usage of smart in USD for different closed-source GPTs. SMART is extremely accessible and cheap to use. Generating 100 hypotheses with GPT-4o Mini costs less than 1 cent.

- **Table 2. Comparison hypotheses between GPT-4 and open-weight models**. We provide insights into whether smaller open-weight models can be used as an alternative to more capable closed-source models. While we do not recommend this, we find significant overlap between smaller and larger models.

- **Table 3. Example hypotheses and justifications when dataset column names are hidden**. This table showcases that SMART cannot generate meaningful hypotheses if the feature names are not interpretable.

- **Table 4. Identifying slices with the highest performance discrepancies across two deep learning tabular classifiers**. We additionally run how SMART compares to other approaches for two state-of-the-art deep learning classifiers.

- **Figure 1. Identifying the importance of feature names as a source of information**. We show that the ability to detect model failures is smaller if column names are hidden.

---
# Thank you
The review has been extremely productive. We thank everyone for their help in shaping the paper to be in better form.

---

### Decision · Program_Chairs · 2024-09-25

**Decision:**

Accept (poster)

**Comment:**

This work introduces a method for formulating and testing hypotheses around when and why a statistical model performs poorly on a given dataset. An LLM is employed to both formulate plausible hypotheses around potential failure modes and select datapoints that are relevant to these hypotheses, followed by an optional set of statistical tests to confirm or reject generated hypotheses (referred to as self-falsification) (Sec. 4). The resulting method outperforms various prior methods that do not incorporate a broader semantic context on discovering impactful features (Sec. 5.2) and is more robust (Sec. 5.1 & 5.3).

The authors communicated extensively with the reviewers to clarify concerns, which led to most reviewers agreeing to accept the work.  Specifically, most reviewers agreed that this work provides a novel method that offers significant improvements. The main concern from the dissenting reviewer is that the proposed method requires meaningful feature names to be present in the tabular data that it operates on. While this is a valid limitation that should be noted in the work, many tabular datasets, including the six real-world datasets considered in this work (five in Sec. 5.1, one in 5.2) do have descriptive feature names. The results generally demonstrate that this technique can yield nontrivial gains in those settings. Additionally, concerns around the level of technical contribution were surfaced by the dissenting reviewer. However, the majority of reviewers express positive views on the level of technical novelty. The combination of LLM-based hypothesis generation and statistical verification is particularly effective (Tab. 2) and novel in this context.

Given this, the recommendation is to accept this work on the ground of its strong results and nontrivial technical novelty. The authors are encouraged to address all the noted concerns, including the method's reliance on meaningful feature names noted by reviewer ctad.